**Technical Report**

# Massively parallel immunopeptidome by DNA sequencing provides insights into cancer antigen presentation

Quanming Shi[1,2], Elana P. Simon [1,2,3], Cansu Cimen Bozkus [4], Anna Kaminska[4], Leandra Velazquez[4], Mansi Saxena[4], Zilin Zhang[5], Julia A. Belk [1,2], Shuo Wang[5], Nuoya Yang [5], Yaowen Zhang[1,2], Ashley Kwong[1,2], Yonglu Che[1,2], Robert R. Stickels [6], Charles R. Crain[7], Laura Schmidt-Hong [7,8], Cheryl F. Lichti[9,10], Gaurav D. Gaiha [7,11,12], Theodore L. Roth [6], Nina Bhardwaj [4], Ansuman T. Satpathy [6], Bingfei Yu [5,13,14] ✉ & Howard Y. Chang [1,2,15] ✉

Human leukocyte antigens (HLAs) are encoded by the most polymorphic genes in the human genome. HLA class I alleles control antigen presentation for T cell recognition, which is pivotal for autoimmunity, infectious diseases and cancer. Current knowledge of HLA-bound peptides is limited, skewed and falls short of population-wide HLA binding profiles for high-value targets. Here we present ESCAPE-seq (enhanced single-chain antigen presentation sequencing), a massively parallel platform for comprehensive screening of class I HLA–peptide combinations for antigen presentation via deep DNA sequencing. ESCAPE-seq demonstrates programmability, high throughput, sensitivity and nominated viral and cancer epitopes. We simultaneously assessed over 75,000 peptide–HLA combinations, revealing broadly presented epitopes from oncogenic driver mutations and fusions across diverse *HLA-A*, *HLA-B* and *HLA-C* alleles that cover 90% of the human population. We further identified epitopes that are differentially presented, comparing oncogenic hotspot mutations versus wild type. ESCAPE-seq enables one-shot population-wide antigen presentation discovery, offering insights into HLA specificity and immune recognition of genomic mutations.

The major histocompatibility complex (MHC) is the most polymorphic region of the human genome[1]. MHC underlies the central event in adaptive immunity termed antigen presentation[2]. Every cell in the body presents bits of its proteome as short peptides in the cleft of MHC proteins on the cell surface, which allows T cells to determine whether a cell is endogenous ('self') or foreign, has been infected or became cancerous. In humans, MHC class I (MHC-1) is known as HLA class I (HLA-I), encoded by *HLA-A*, *HLA-B* and *HLA-C* genes, which present antigen epitopes to CD8+ T cells to eradicate pathogen-infected cells and cancer cells[2]. HLA loci are the most prominent signal in any genome-wide association

study of autoimmune and infectious diseases[3]. Acquisition of HLA alleles has been used to track ancient human population migration and the selection pressures in different continents[4]. A decades-long goal of the entire field has been to achieve a population-wide understanding of the functional HLA binding preferences across diverse HLAs[4].

In HLA-I antigen presentation, peptide fragments are sampled from the cellular proteome through proteolysis, transported to the endoplasmic reticulum (ER) via the TAP complex, trimmed into 8–10 amino acids when necessary and loaded onto HLA-I molecules upon high-affinity binding. The stable peptide–HLA (pHLA) complex is then

transported to the cell surface for subsequent T cell recognition[5]. The set of peptides presented in HLA-I is termed the immunopeptidome. Mass spectrometry (MS) is instrumental to determine the immunopeptidome and to define fundamental rules of antigen presentation, with recent advances in HLA monoallelic MS approaches enabling systematic characterization of the allele-specific peptide repertoire. However, current immunopeptidome data remain highly skewed[6], as MS-based approaches still face major challenges including requirement for large sample input, difficulty in deconvoluting multi-allele HLAs and limited sensitivity in detecting low-abundance but clinically relevant peptides such as those from pathogens and mutated oncoproteins[7,8]. Computational prediction algorithms such as NetMHC offer rapid and scalable in silico antigen identification but show decreased reliability for HLA alleles with insufficient training datasets, especially for *HLA-C* alleles with lower surface expression compared with *HLA-A* or *HLA-B* alleles[9,10].

Existing HLA-I immunopeptidome data are highly biased toward certain well-studied alleles such as *HLA-A2*, while many HLA alleles associated with disease but more frequent in non-European ancestry groups have scant biochemical characterization (Supplementary Note 1). Such biases also lead to substantial inequities in clinical trial eligibility[11]. In particular, targeting HLA-C may enable population coverage with a smaller number of alleles, but the paucity of antigen binding data has limited this strategy. Other methods, such as in vitro peptide binding and T cell stimulation assays, have limited throughput due to the requirement and cost of individual peptide synthesis[12–14]. Considerable efforts have been invested over time in enhancing throughput, as evidenced by the recent advancements in methodologies such as EpiScan and time-resolved fluorescence resonance energy transfer (TR-FRET) with parallel reading, but scalability remains limited[15,16]. For instance, EpiScan does not work for *HLA-C* alleles[12], which precludes comprehensive and unbiased HLA profiling.

Here we developed ESCAPE-seq, a high-throughput and combinatorial platform, enabling the simultaneous assessment of around 75,000 pHLA combinations (1,500 peptides across 50 HLA alleles) in a single screen. By applying ESCAPE-seq to SARS-CoV-2 viral variants, recurrent tumor driver mutations and aberrant tumor fusion variants, we successfully identified viral antigens and tumor neoantigens that are missed from computational prediction. Importantly, ESCAPE-seq revealed public shared tumor neoantigens presented across multiple HLA alleles, providing a comprehensive catalog of tumor neoantigens with potential therapeutic implications across diverse human populations.

## Results

### Single-chain pMHC trafficking to the cell surface depends on specific peptide–HLA binding

The peptide–MHC (pMHC) single-chain trimer (SCT) has been extensively utilized to explore pMHC–T cell receptor (TCR) interactions. The pMHC consists of an 8–10-amino-acid antigen peptide, $\beta_2$-microglobulin (B2M) and an MHC allele arranged in tandem and linked by flexible glycine- and serine-rich linkers[17,18]. Typically, a high-affinity peptide preceded by a signal peptide traffics the trimeric complex to the cell surface, endowing high surface expression in HLA-knockout cells (Extended Data Fig. 1a,b).

Using HLA-A2 SCT, we observed that substituting the high-affinity peptide with one lacking HLA-A2 binding affinity (no affinity) led to a striking reduction of cell-surface staining. Neither B2M nor HLA-A2 antibodies showed positive staining of the no affinity peptide construct (Fig. 1a,b and Extended Data Fig. 1a,b). To exclude misfolding, we fused enhanced green fluorescent protein (eGFP) directly to the cytoplasmic tail of the HLA-A2 allele in both high-affinity peptide–SCT and no affinity peptide–SCT. We observed robust membrane localization only with the high-affinity pp65 peptide, whereas the nonbinding peptide displayed diffusive intracellular eGFP distribution (Fig. 1c), indicating that stable HLA–peptide binding is required for efficient surface trafficking. Structural integrity was confirmed via two

approaches: HLA-A2:NY-ESO-1 SCT-expressing cells activated cognate TCR-expressing Jurkat cells via NFAT reporter induction (Extended Data Fig. 1c), and H2-Kb:SIINFEKL antibody showed concordant staining with total surface H2-Kb (Extended Data Fig. 1d). Together, these results showed that only properly folded SCTs with high-affinity peptides can traffic to and maintain structural integrity at the cell surface.

To assess the dependence of SCT surface localization on peptide binding across multiple MHC alleles, we generated SCTs with representative alleles from HLA-A, HLA-B and HLA-C groups (A*0101, B*0702 and C*0401). Leveraging the Immune Epitope Database (IEDB), we systematically fused peptides with high or no binding affinity into SCTs for each HLA-I allele (Supplementary Table 1). Consistently, trimers with nonbinding peptides exhibited markedly reduced surface expression across examined HLA alleles (Fig. 1d), including mouse MHC H2-Kb allele (Fig. 1d). The geometric mean of fluorescence intensity (gMFI) for high-affinity peptides was around 1,000-fold higher than for nonbinding peptides, offering a broad dynamic range (Extended Data Fig. 1e). We observed a loose sigmoidal relationship between gMFI and peptide HLA-A2 affinities, plateauing at higher affinities (Fig. 1e and Supplementary Table 2). The Y84C mutation conserved across *HLA-A*, *HLA-B* and *HLA-C* alleles and G2C in linker generate disulfide bridges, enhancing pHLA stability and surface display[19,20] (Extended Data Fig. 1f). Comparing surface display with or without cysteine mutation revealed noisier staining without mutation (Fig. 1e and Methods), suggesting the cysteine mutation enhances sensitivity and robustness for detecting HLA-restricted peptides[21].

HLA proteins are highly unstable without bound peptides, with the peptide-loading complex (PLC) stabilizing MHC before suitable peptide loading[22,23]. We hypothesized that SCTs with nonpresentable peptides are destabilized, impeding ER exit and membrane trafficking. *TAP1/2* knockout minimally affected SCT presentation (Extended Data Fig. 1g). Truncated SCTs without fusing peptides showed cytoplasmic diffusion, whereas HLA-A2 alone localized to the cell surface in HLA-knockout cells (Fig. 1f and Extended Data Fig. 1h,i), supporting that SCT antigen presentation bypasses PLC. When truncated SCTs were co-expressed with a presentable HLA peptide in the ER, surface presentation was partially restored (Fig. 1f and Extended Data Fig. 1i), albeit at least tenfold less efficiently than for the SCT. To a certain extent, this mirrors a previous method that used the co-expression of separate HLA and peptide transgenes to score antigen presentation[16]. These results suggest that cell-surface pHLA expression serves as a proxy for peptide binding strength, with SCTs offering higher dynamic range and sensitivity.

### ESCAPE-seq leverages HLA escape to cell surface to quantify peptide–HLA presentation

We next exploited the system's potential for high-throughput HLA peptide screening (Fig. 2a). Starting with known-affinity peptides from IEDB, we selected thousands of peptides with different affinities for four representative HLA alleles (A*0101, A*0201, B*0702 and C*0401), cloning them into pooled lentiviral vectors per HLA allele and introducing them into cells at a low multiplicity of infection (Methods). Cells were sorted into four intensity-based bins in log-scale reflecting their programmed SCT's ability to escape to the cell surface (Fig. 2b). We assigned ESCAPE-seq scores (*E*-scores) to each peptide (Methods), with higher *E*-score representing higher cell-surface staining and presentation. Biological replicates demonstrated high correlation (Extended Data Fig. 2a,b) for all alleles studied. We compared *E*-score correlation with IEDB affinity using HLA-A*02 SCT with or without Y84C mutation. ESCAPE-seq with Y84C mutation showed better correlation (Pearson correlation 0.77 versus 0.49 without the mutation) (Fig. 2c and Extended Data Fig. 2c). Therefore, we used the Y84C mutation in all subsequent ESCAPE-seq experiments (Extended Data Fig. 1f).

By binning IEDB affinity of defined presentable peptides, we measured recall and precision rates (Fig. 2d and Methods). Comparison with NetMHC4, a widely used pMHC prediction tool trained on IEDB[24],

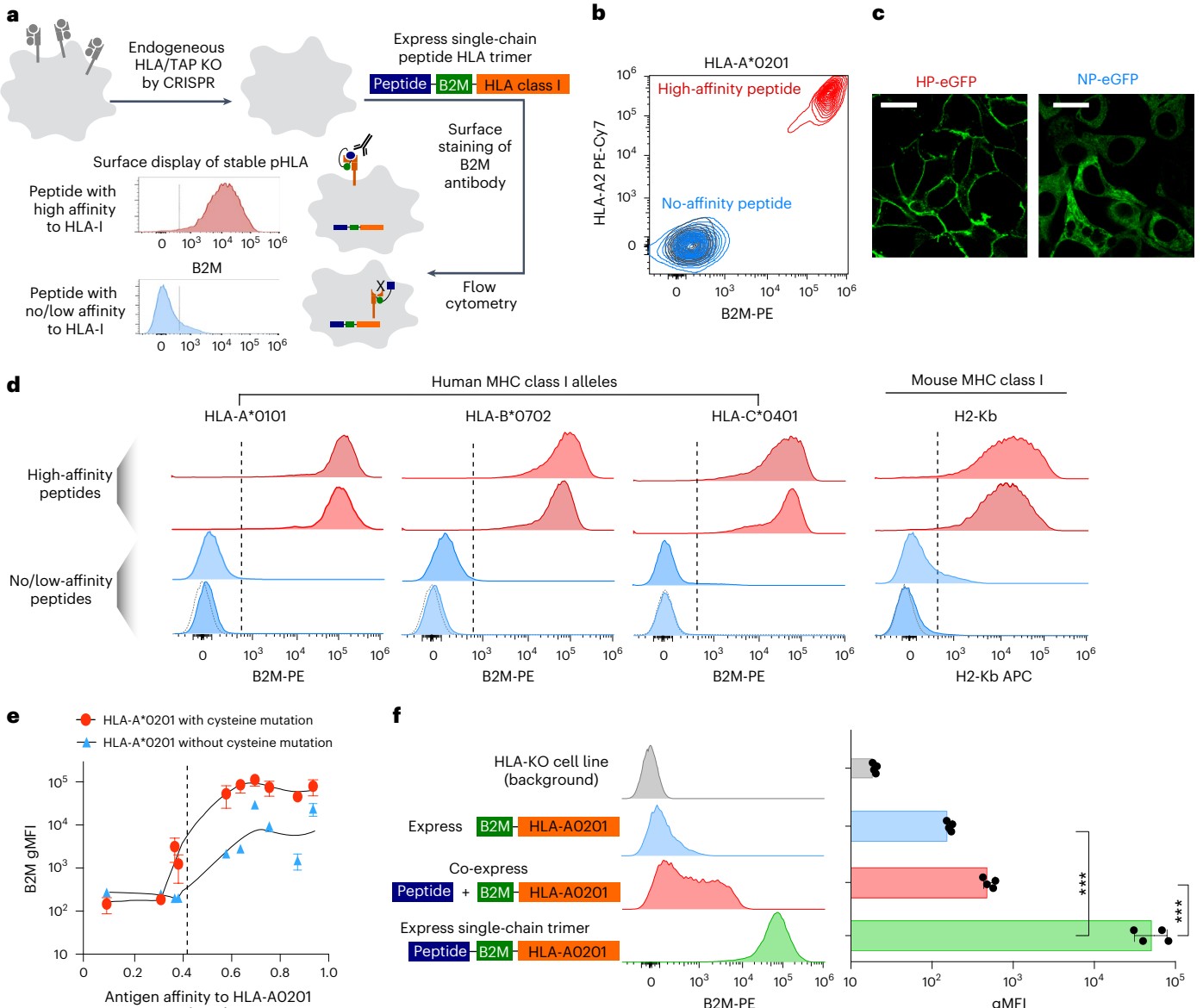

**Fig. 1 | SCTs of pMHC class I differentiate presentable peptides from those with no or low affinities. a**, Schematic of assay, where SCTs with high-affinity peptides were displayed on cell surface when transfected, while SCTs with no peptides binding to MHC alleles failed to escape to cell surface. KO, knockout. **b**, Flow cytometry measurement of surface pHLA-A*0201 trimers with high-affinity (red) and low-affinity peptides (blue) in SCTs. Antibodies against either B2M or HLA-A2 alleles were used to stain the cells. Blank cells are shown in gray as background signals. **c**, Imaging of pHLA SCT fused with eGFP directly at the cytoplasmic tail of HLA-A2. Repeated experiments show consistent results. Representative images are shown. Scale bar, 40 μm. **d**, Representative histograms of cell-surface SCTs for human

HLA-A*0101, B*0702 and C*0401, and mouse H2-Kb allele. Two examples of high-affinity peptides (red) and two of negative peptides (blue) were shown per allele. The gray line shows the background from blank cells. **e**, Plot of gMFI versus IEDB measured binding affinity for SCTs containing Y84C mutation (red) or of wild-type (WT) allele (blue). $n = 2–4$. **f**, Histogram of B2M staining for cell-surface pHLA-A2 presentation. Three conditions were measured, respectively, B2M fused with HLA-A2 only, co-expression of B2M–HLA-A2 with a high-affinity peptide pp65 preceded with a signal peptide and, last, a normal SCT for pp65 peptide. Right: the bar graph showing the quantification of signals. $n = 4$; ***$P < 0.001$ by two-sided $t$-test. Error bars denote s.e.m.

revealed the reliable performance of ESCAPE-seq. ESCAPE-seq yielded >90% recall for high-affinity peptides, aligning closely with NetMHC4 predictions (Fig. 2d). The receiver operating characteristic (ROC) curve indicated an area under the curve (AUC) of 0.919 when evaluating ESCAPE-seq against IEDB measured affinity, comparable to NetMHC4 in both binding affinity (NetMHC4-BA) mode and eluted ligand modes (NetMHC-EL) (Fig. 2e). The precision–recall curve (PRC), which offers more robust metrics when positive rates are low (typical for antigen presentation datasets where most peptides will not bind the HLAs), showed comparable performance with ESCAPE-seq, obtaining an AUC of 0.91 (Fig. 2f).

Systematic comparisons across HLA-A*0101, B*0702 and C*0401 showed ESCAPE-seq achieving similar performance to NetMHC4 (Extended Data Fig. 2d–k), evident in both correlation coefficient (Extended Data Fig. 2l,m) and AUC-ROC (Fig. 2g). PRC analysis mirrored ROC findings (Fig. 2h). Overall, our results indicate ESCAPE-seq yields comparable metrics to NetMHC on the model's training data.

### ESCAPE-seq reveals presented epitopes from SARS-CoV-2 variants
Utilizing ESCAPE-seq, we screened for presentable peptides in SARS-CoV-2 spike and nucleocapsid proteins. All peptides were

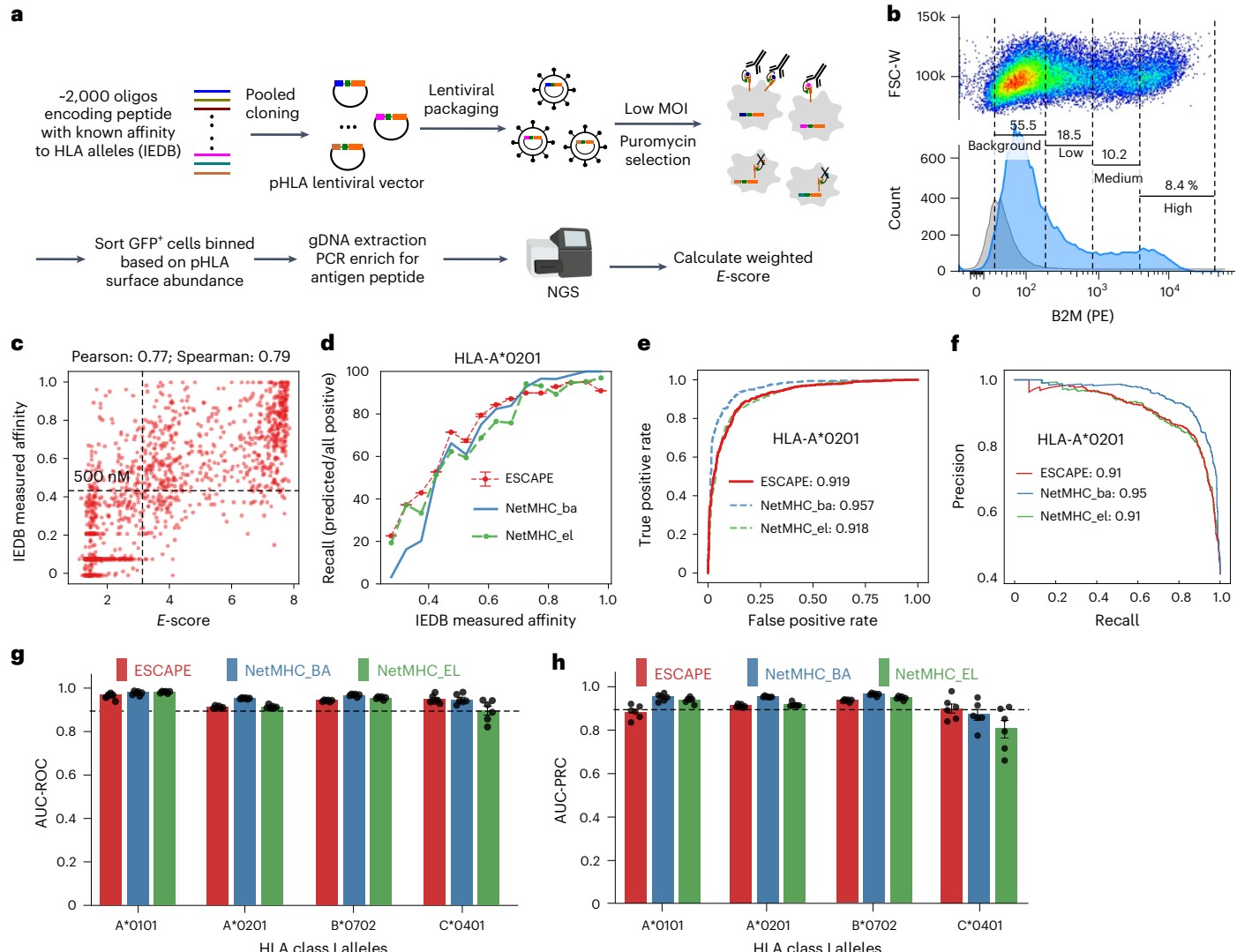

**Fig. 2 | ESCAPE-seq demonstrates good performance and benchmarks across all HLA-I subtypes. a**, Schematic of ESCAPE-seq. A pool of peptides across a wide range of affinities was selected from IEDB, synthesized and cloned to build the SCT library. Cells were transduced with the pooled virus and were sorted and sequenced. **b**, Representative histogram of cell-surface HLA-A*02 trimer expression on cells transduced with pooled virus. The whole cell population was sorted into four evenly divided bins on log-scale as indicated on the histogram. Gray line marks the signal from cells without pHLA transduced. **c**, Scatter plot of IEDB affinity versus $E$-score calculated for each peptide. A line at 500 nM of affinity and a cutoff line for $E$-score were drawn as indicated. **d**, Plots of recall rate

(predicted positive/all positive) versus IEDB-defined affinity bins. ESCAPE-seq performance was compared with NetMHC4 in either binding affinity (ba) or eluted ligand (el) mode. $n = 2$ biological replicates. **e**, Plot of ROC curve of HLA-A2 allele for evaluating ESCAPE-seq in comparison with NetMHC. The AUCs were noted. **f**, PRC plot for HLA-A2 allele in comparison with NetMHC. **g**, Bar plots of AUC-ROC for all four HLA-I alleles to compare the performance of ESCAPE-seq with NetMHC. **h**, Bar plots of AUC-PRC for all four HLA-I alleles to compare the performance of ESCAPE-seq with NetMHC. $n = 4$. Error bars denote s.e.m. FSC-W, forward scatter width; HP, high-affinity peptide; MOI, multiplicity of infection; NGS, next-generation sequencing; NP, peptide with no binding affinity.

standardized to nine amino acids, comprising approximately 1,500 peptides tiling across spike and nucleocapsid proteins, plus 900 viral mutation peptides from 17 SARS-CoV-2 variant strains including alpha, beta, delta and omicron (Fig. 3a, Methods and Supplementary Table 3).

ESCAPE-seq screens across three common HLA-I alleles (A*0101, A*0201, B*0702) showed high replication consistency (Extended Data Fig. 3a,b,). Comparing $E$-scores against NetMHC predictions, we observed 90% of peptides were nonpresentable by both methods whereas 2% of peptides were discordant (Fig. 3b and Extended Data Fig. 3c–e). ESCAPE-positive peptides' motif patterns closely aligned with known HLA-I allele patterns (Fig. 3c), and most literature-reported SARS-CoV-2 peptides appeared positive by ESCAPE-seq[25,26] (Fig. 3d,e). Presentable peptides were dispersed across viral proteins without forming clusters. Cross-allele clustering indicated few peptides were presented by two HLA alleles, and rarely by all three (Fig. 3f,g). Generally,

5–10% of peptides are presentable across three alleles depending on $E$-score cutoff (Extended Data Fig. 3f). Interestingly, ROC-AUC between ESCAPE-seq and NetMHC was 0.95 for IEDB training peptides but decreased significantly for SARS-CoV-2 peptides (Extended Data Fig. 3g,h). Since ESCAPE-seq performance is input-independent, this decrease likely reflected reduced NetMHC prediction power on new peptides, suggesting NetMHC performs less effectively on novel peptides compared with familiar training data—an intuitively expected outcome.

We investigated peptide presentation changes across mutations in spike and nucleocapsid proteins from SARS-CoV-2 variant strains. Notably, numerous nonpresentable wild-type peptides became presentable after mutations, and vice versa (Fig. 3h,i and Extended Data Fig. 3i,j). Point mutations particularly altered presentation status, with notably fewer peptides retaining their presentation status after a point mutation (Fig. 3j and Extended Data Fig. 3k).

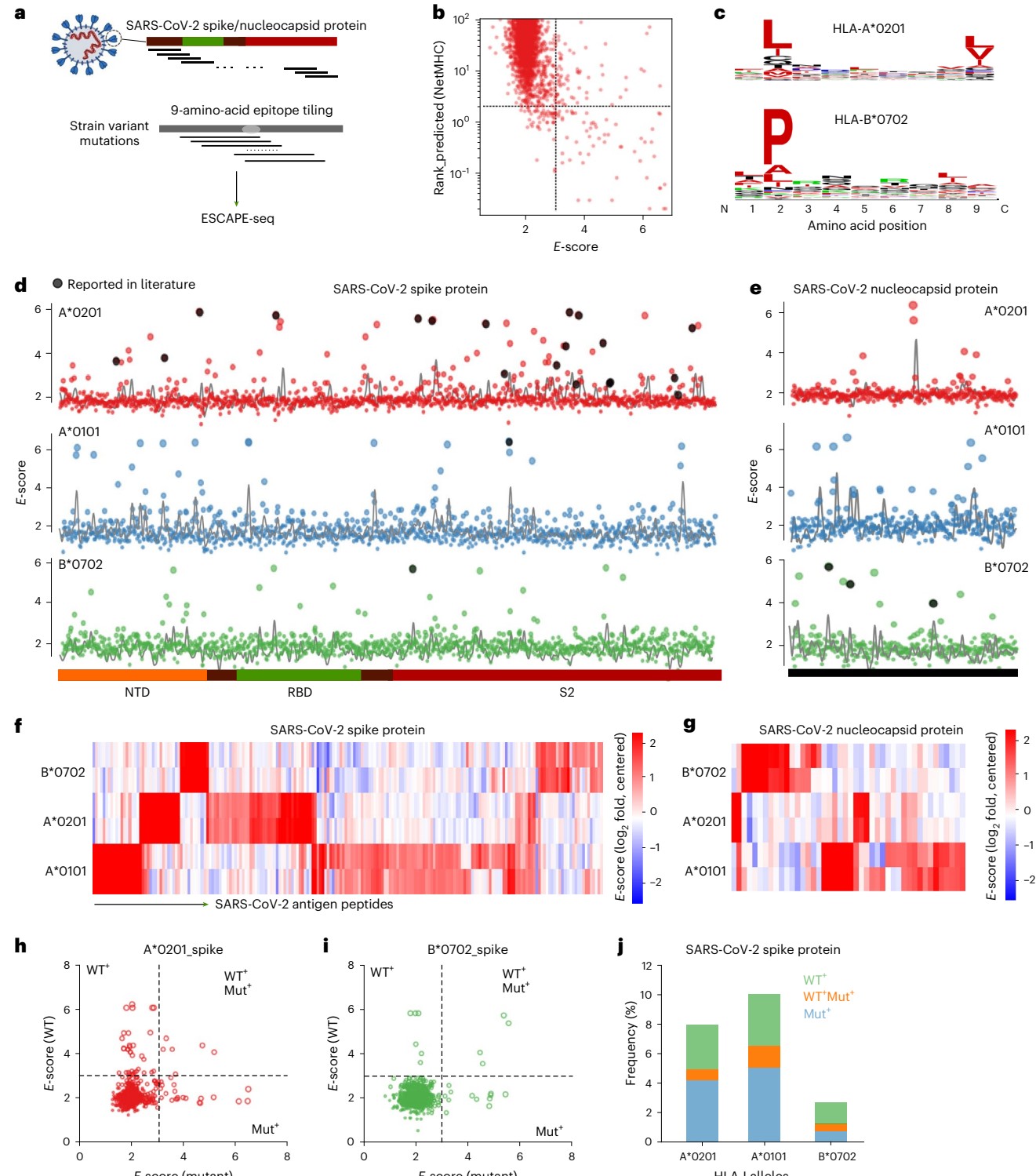

**Fig. 3 | ESCAPE-seq reveals presented peptides in SARS-CoV-2 spike and nucleocapsid proteins, and strain variants. a**, Schematic of peptide pool generation by tiling across full spike and nucleocapsid proteins, and the strain variant mutations. **b**, Scatter plot of NetMHC prediction versus *E*-score for each peptide. Lines of cutoff values were drawn at 2% rank for NetMHC_El and 3.2 for *E*-score. Here, the *E*-scores are normalized as described in the Methods. The percentage ranking score from NetMHC shows how a peptide's predicted binding affinity compares to a large set of random natural peptides, with typical cutoffs of 0.5% and 2% as strong and weak binders, correspondingly. **c**, Consensus motif for HLA-A*02 and HLA-B*07 from ESCAPE-seq positive SARS peptides. **d**, Plots of *E*-score for each peptide with three HLA alleles across the spike protein (*x* axis, schematic drawing of spike protein at the bottom). Black circles highlight the

reported presentable peptides from literature. **e**, Plots of *E*-score for peptides spanning nucleocapsid protein. Three HLA alleles were drawn in aligned position for each peptide. Black circles highlight the reported presentable peptides from literature. **f**, Hierarchical heatmap showing the cluster and shared peptides from SARS spike protein across HLA alleles of interest. **g**, As in **f** but for nucleocapsid protein. **h**, Neoantigen plot of SARS-CoV-2 spike peptides presented by HLA-A*02 allele, where the *x* axis is the *E*-score for each mutational peptide and the *y* axis is the *E*-score of its corresponding wild-type peptide. Three quadrants containing positive antigen peptides were highlighted. **i**, As in **h** but for HLA-B*07 allele. **j**, Bar graph of percentage of peptides in three quadrants for all three HLA-I alleles. NTD, N-terminal domain; RBD, receptor binding domain; S2, S2 subunit of spike protein.

## Combinatorial ESCAPE-seq achieves simultaneous profiling of peptide presentation across diverse HLA alleles

We next integrated ESCAPE-seq with a combinatorial HLA indexing strategy to screen peptide pools across multiple HLA alleles simultaneously (Fig. 4a and Methods). We integrated a barcode system with synonymous mutations in B2M gene to enable HLA allele identification via sequencing (Fig. 4a). This multiplexed screening of peptide pools with HLA allele pools can simultaneously screen over 10,000 pHLA SCT variants, with the SCT library transduced and sorted into four bins based on cell-surface B2M expression (Fig. 2b).

For large-scale validation, we selected the top 986 MS-captured peptides across 30 HLA alleles from existing monoallelic HLA MS data[27]. We performed combinatorial ESCAPE-seq screening, examining over 29,000 pHLA interactions (Fig. 4b). Quality control analysis showed high consistency between replicates, and MS-validated peptides exhibited substantially higher $E$-scores (Extended Data Fig. 4a–c).

To assess the ability of ESCAPE-seq to identify MS-validated peptides, we calculated the percentage of MS-detected peptides that were also identified by ESCAPE-seq for each HLA allele (capture percentage, Fig. 4c). Remarkably, ESCAPE-seq captured over 80% of MS-detected peptides across most HLA alleles, with some reaching 90%, demonstrating that a single ESCAPE-seq experiment can recapitulate the majority of findings from multiple MS experiments across 30 different HLA alleles. ROC analysis revealed strong performance, with AUC-ROC values between 0.8 and 1.0 for most HLA alleles, although slightly lower (~0.7) for HLA-B56:01, B52:01 and C*07:01 (Fig. 4d and Extended Data Fig. 4d). In addition, ESCAPE-seq outperformed NetMHC4.1 predictions for multiple HLA alleles (Extended Data Fig. 4e).

Next, we performed ESCAPE-seq analysis with peptides ranging from 8 to 12 amino acids, examining 10 gene loci (3 mutations, 3 wild-type counterpart genes, 3 fusions, 1 deletion) across 25 HLA alleles, designing ~45 peptides per target that tiled across mutation or junction sites (10,000 total interactions) (Extended Data Fig. 4f–h). The 10-mers showed the highest presentation frequency, including the well-established KRAS(G12D) 10-mer epitope (Extended Data Fig. 4i,j). An $E$-score heatmap of mutation position versus peptide length shows that addition of one amino acid often maintains peptide presentation ability (Extended Data Fig. 4j).

## Combinatorial ESCAPE-seq enables population-wide discovery of cancer neoantigens

To systematically identify cancer neoantigens across human populations by combinatorial ESCAPE-seq, we assembled 92 prevalent oncogenic mutations from cancer driver genes and the 31 most frequent oncogenic fusions from the COSMIC database[28] (Methods). This curated pool encompassed point mutations such as KRAS(G12D), BRAF(V600E) and TP53(R175H), and fusion junctions of oncogenic translocations such as BCR-ABL1 (Supplementary Table 4). We then generated a peptide pool tiling over these mutations and presented each peptide across the 50 most common *HLA-A*, *HLA-B* and *HLA-C* alleles (Fig. 4e), covering >90% of the global population[27].

This multiplexed screening generated over 75,000 pHLA SCT variants in a single experiment, with 100 IEDB-derived peptides spiked in as controls (Fig. 4e and Methods). The library showed high reproducibility (correlation coefficient 0.89) between replicates (Extended Data Fig. 5a,b). We optimized the protocol to minimize peptide–barcode mis-pairing from recombination events during library amplification, achieving <10% recombination-induced nonspecific reads (Methods and Extended Data Fig. 5c,d).

$E$-score histograms per HLA allele showed bimodal distributions with dominant negative peaks, and normalized $E$-scores were calculated per peptide–HLA pair (Methods and Extended Data Fig. 5e). Spike-in peptides correlated well with IEDB affinity/elution data, confirming robust single-allele performance in combinatorial settings (Fig. 4f and Supplementary Table 6).

Importantly, ESCAPE-seq uncovered previously reported tumor neoantigens paired with correct HLA alleles that were falsely predicted as nonbinders by NetMHC (Fig. 4g and Supplementary Table 7). This list is enriched for tumor neoantigens presented on *HLA-C* alleles such as KRAS(G12V) (GAVGVGKSA)/C*0304, KRAS(G12C) (GACGVGKSA)/C*0304, KRAS(G12D) (GADGVGKSA)/C*0802 and EGFR(T790M) (LTSTVQLIM)/C*0701 (refs. 15,29–32), highlighting the sensitivity of ESCAPE-seq to identify HLA-C-presented tumor neoantigens that are missed by computational prediction (Fig. 4g). Notably, the reported HLA-C*0304-restricted KRAS(G12V) (GAVGVGKSA) antigen can be presented by additional HLA-I alleles covering diverse human populations, including B*5401 (East Asia), B*5601 (Australia, Oceania) and C*0602 (North Africa)[33] (Supplementary Table 7).

The landscape of neoantigen presentation across oncogenes and HLA alleles (Fig. 5a) provided several lessons. *HLA-B* alleles displayed substantial variance in peptide presentation (1–20% presentation rates) while *HLA-C* alleles showed higher peptide presentation (Extended Data Fig. 5f). Most peptides were presented by only one or two HLA alleles (Fig. 5b), consistent with divergent allele-specific binding motifs[27,34–36]. However, a small number of neoantigen peptides were presented by ≥20 HLA alleles (that is, 'public neoantigens') (Fig. 5b); such antigen peptides hold great interest for potential utilization in immunotherapies. Using HLA-C*0304 as an example, the majority of these 'public antigens' showed intermediate and high $E$-scores. Some of those with the highest $E$-scores could be readily detected and validated in independent MS experiments (Fig. 5c, Extended Data Fig. 5g and Methods).

## ESCAPE-seq nominates high-priority cancer neoantigens from driver oncoproteins

We next investigated antigen presentation characteristics by aggregating results across HLA alleles for each oncogenic mutation and identified hot spots of presented peptides across HLAs (Fig. 5d,e). Defining presentable neoantigens as those with at least one presentable peptide, several HLA alleles presented over 50% of the 92 samples tested, with *HLA-C* alleles showing particularly high coverage (60–80%) (Extended Data Fig. 5f–h). We observed that mutations displayed varying HLA compatibility: 'public' driver neoantigens such as EGFR(T790M) and MED12(G44V) were presented by >60% of HLA alleles, while 'private' neoantigens such as KRAS(G12C) and TP53(R284W) were presented by only 2% (Fig. 5h).

For oncogenic fusion proteins including those prevalent in pediatric sarcomas[37,38], presentation patterns paralleled those of point mutations (Extended Data Fig. 5i,j). We discovered a diverse range of antigen presentation compatibility for fusion oncoproteins (Fig. 5g). For example, EWSR1–FLI fusion peptides are presented by almost 30 HLA alleles (approximately 60% of HLAs tested, a 'public neoantigen'), whereas PAX7–FOXO fusion was presented by only one allele. Of note, all presented fusion breakpoint peptides represent potential neoantigens as they are de novo peptides consisting of two gene fragments not observed in the wild-type genome.

Considering that typical diploid human cells harbor two alleles of each HLA-A, HLA-B and HLA-C, we extrapolated the distribution of presented mutation coverage within the human population through repeated sampling (Methods). Remarkably, >90% of prevalent oncogenic mutations could be presented by at least one allele, with fusion mutations showing slightly lower but still substantial coverage (>80%) (Extended Data Fig. 5k).

Finally, we systematically compared oncogenic (Mut) versus wild-type (WT) peptide presentation by including both in the same assay. Analysis revealed four distinct presentation patterns: (1) WT⁻Mut⁻ (neither presented), (2) WT⁺Mut⁺ (both presented), (3) WT⁺ only (mutation enables immune escape), and (4) Mut⁺ only (mutation confers novel HLA binding) (Fig. 6a). It is known that a single amino acid change may not change the TCR binding or specificity[39,40], and

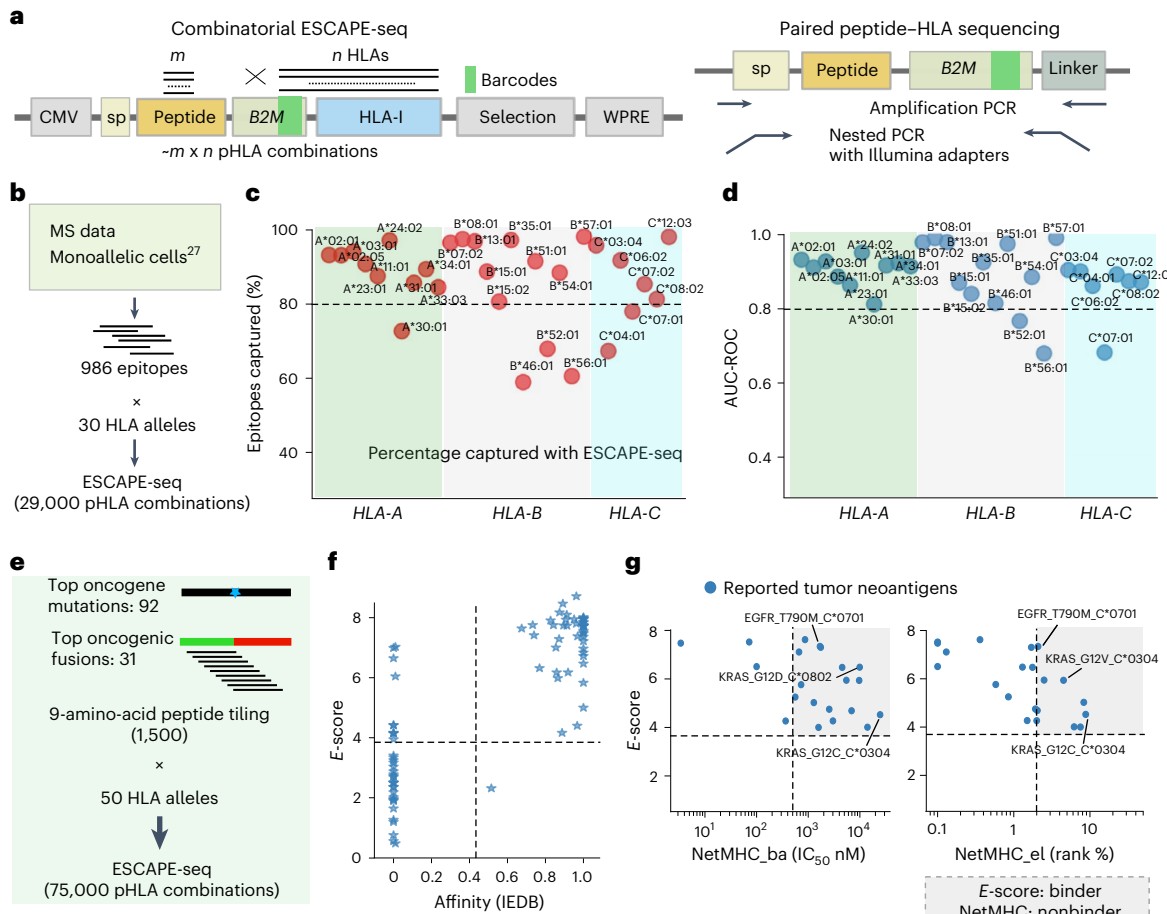

**Fig. 4 | Combinatorial ESCAPE-seq achieves simultaneous profiling of peptide presentation across diverse HLA alleles in one screen. a**, Schematic representation of generating, barcoding and sequencing for combinatorial ESCAPE-seq approach. As an example, the barcodes representing *n* HLA alleles were introduced via synonymous mutations at the 3′ end of *B2M* gene (12 amino acids). Libraries of paired peptides and HLA barcodes were then amplified by three rounds of PCR for the Illumina-based sequencing platform (Supplementary Table 5). sp, signal peptide. **b**, Schematics of combinatorial ESCAPE-seq using existing MS elution peptides. The 986 most common peptides from the MS dataset, along with 30 HLA alleles, were selected[27]. **c**, The percentage of epitopes identified by MS that show a positive *E*-score by ESCAPE-seq. **d**, AUC-ROC metrics comparing ESCAPE-seq and MS results across 30 HLA alleles. Peptides were

assigned a value of 1 if detected in MS and 0 if not. **e**, Schematics of combinatorial ESCAPE-seq used to screen cancer epitopes. A pool of 1,500 oncogene peptides, tiling across the 92 top cancer mutations and 31 oncogenic fusions, was randomly combined with 50 human HLA alleles. All peptides were 9 amino acids in length, and a pool of 100 known pHLA pairs were spiked into the experiment. This generated approximately 75,000 peptide–HLA pairs in one screening. **f**, Plot of *E*-score versus IEDB binding affinity for the spike-in pHLA results. The spike-in pool included 17 known antigens and >80 positive and negative peptides selected from IEDB. **g**, Plot of *E*-score versus NetHMC prediction for the known antigen peptides in the combinatorial pool. NetMHC4 results in both binding affinity mode (ba, top) and eluted ligand model (el, bottom) are shown. The grayed box highlights peptides for which NetMHC failed to make a prediction.

thus differential antigen presentation is one of the key determinant therapeutic indexes of immunotherapies targeting cancer neoantigens. With HLA-A*0201 as an example, we found that approximately half of the peptides containing a point mutation and their corresponding wild-type peptides are coordinately presented (Fig. 6b). Across all HLA alleles, the prevalence of this shared population where both mutant and wild-type are equally presented is dominant (Extended Data Fig. 6a), aligning with the widely accepted notion that only a few amino acids on presented peptides serve as anchor points dictating binding[41]. Notably, a substantial number of mutant peptides are well presented but the corresponding wild-type peptides are not (Fig. 6b,c and Extended Data Fig. 6a,b, Mut⁺). We reasoned that these Mut⁺ peptides constitute promising immunogenic neoantigen candidates, since immune tolerance of these antigens will not be established due to the lack of presentation of wild-type peptides. Indeed, these Mut⁺ peptides contain previously reported immunogenic tumor neoantigens including KRAS(G12D) (GADGVGKSA)/C*0802 (ref. 29), FLT3(D835Y) (YIMSDSNYV)/A*0201 (ref. 42) and EGFR(T790M) (IMQLMPFGC)/A*0201 (ref. 43) (Fig. 6b,c). Thus, ESCAPE-seq permits nomination of potential immunogenic

tumor neoantigens to prioritize candidates for antigen-directed immunotherapy.

To validate the immunogenicity of Mut⁺ only peptides, we performed an in vitro peptide-induced T cell stimulation assay using HLA-A*23:01⁺ healthy donor peripheral blood mononuclear cells (PBMCs). We compared one Mut⁺ peptide (p17: LTSTVQLIM*) against two WT⁺Mut⁺ peptides (p22: QLIMQLMPF and p23: LIMQLMPFG) from EGFR(T790M) (Fig. 6e and Extended Data Fig. 6d). Flow cytometry analysis revealed robust IFNγ⁺TNF⁺ polyfunctional T cell responses exclusively to the Mut⁺ only peptide (Fig. 6d–g and Methods). Extended analysis of 13 fusion-derived epitopes identified by ESCAPE-seq using HLA-A*03:01⁺/A*24:02⁺ healthy donor PBMCs revealed that the NPM1-ALK fusion-derived epitope was immunogenic, showing a significant increase in polyfunctional T cells after peptide stimulation (Fig. 6h,i and Extended Data Fig. 6e). Together, these results establish ESCAPE-seq as a powerful tool for identifying potentially immunogenic neoantigens, particularly Mut⁺ only peptides that bypass immune tolerance due to the lack of presentation of wild-type counterparts.

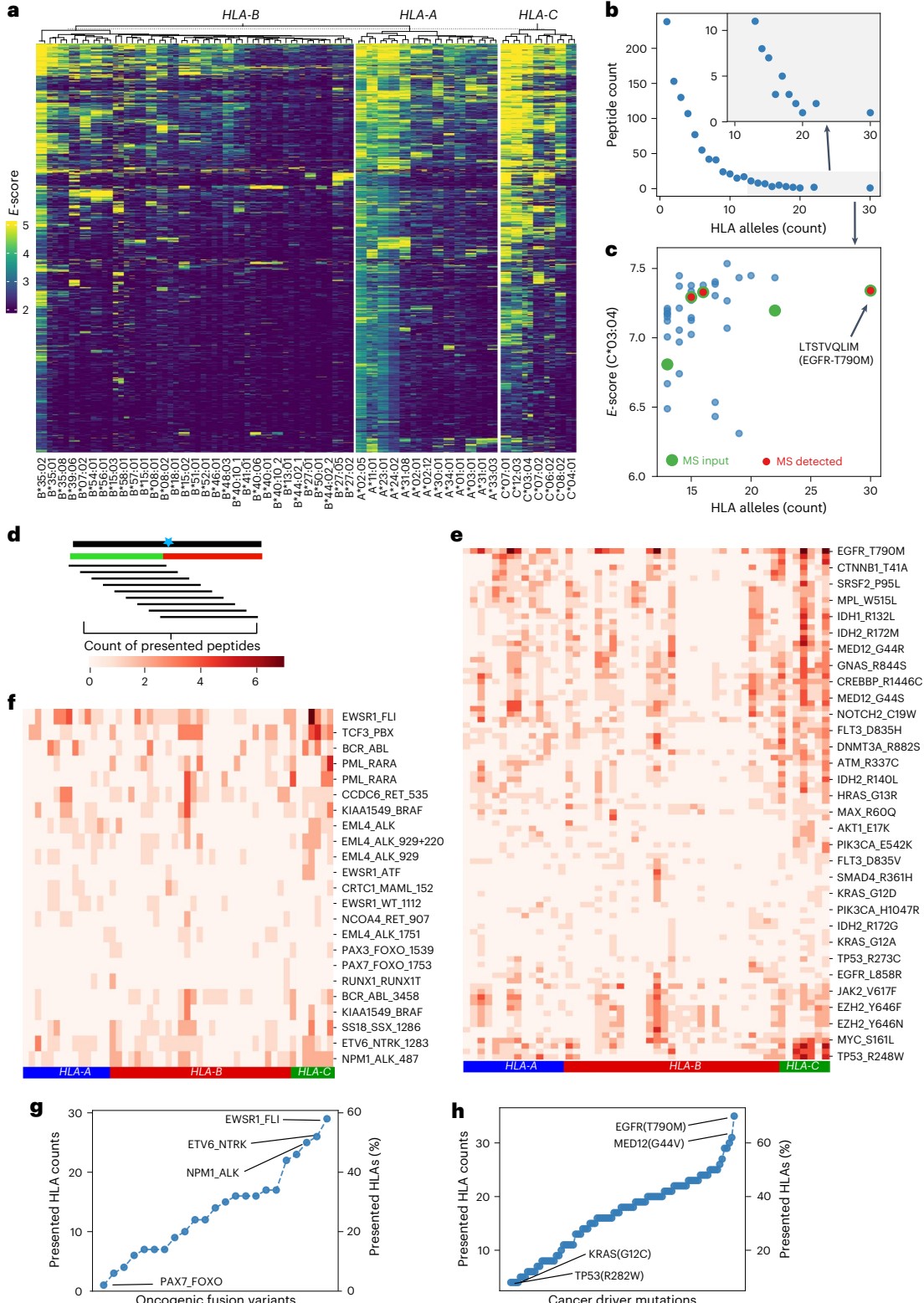

**Fig. 5 | Combinatorial ESCAPE-seq enables population-wide antigen presentation discovery of cancer neoantigens derived from driver oncogenes.** **a**, Heatmap of the *E*-score of 1,500 oncogene peptides (*y* axis) across 50 HLAs (*x* axis), showing moderate clustering and absence of commonly presented peptides. **b**, Count of peptides that were commonly presented by multiple alleles (*x* axis). Inset, magnified view of peptide count in the shaded region. **c**, Scatter plot of *E*-score versus HLA allele count for individual peptides highlighted in the gray region in **b** that are presented by multiple HLA alleles (*x* axis, *n* > 12). Green dots represent peptides included in the MS experiments with monoallelic C*0304 cells, while red dots indicate peptides detected in MS.

**d**, Aggregated analysis combining all peptides presented across the same mutation. For each point mutation or fusion breakpoint, nine candidate peptides tiling the mutation were analyzed. The color map indicates the number of presented peptides out of nine. **e**, Heatmap of number of peptides presented for each oncogenic point mutation (*y* axis) across *HLA-A*, *HLA-B* and *HLA-C* alleles (*x* axis). **f**, Heatmap similar to **e** for peptides derived from oncogenic fusions. **g**, Number of HLA alleles capable of presenting peptides containing oncogenic fusion mutation (*y* axis) plotted against mutations (*x* axis), ordered by HLA allele count. **h**, As in **g** but for oncogenic point mutations.

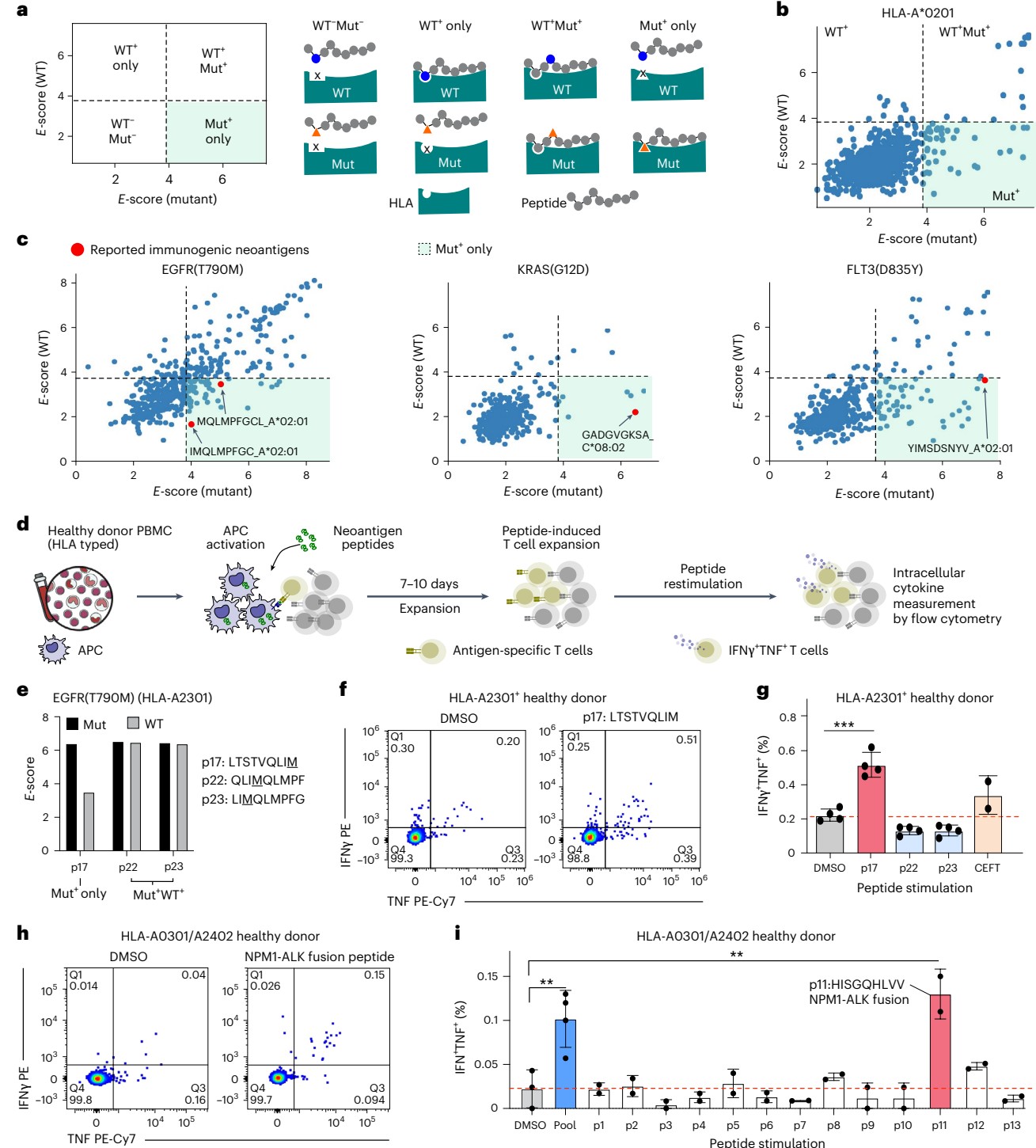

**Fig. 6 | ESCAPE-seq nominates high-priority cancer neoantigens. a,** Schematic of a scatter quadrant plot displaying $E$-scores for peptides with mutations (Mut) versus their corresponding wild-type peptides. Dashed lines indicate the $E$-score thresholds, dividing peptides into four groups. The Mut⁺ only region, where mutant peptides are presented but their corresponding wild-type peptides are not, is highlighted in green. **b,** Example scatter plot of all point mutations as described in **a** for the HLA-A0201 allele. The reported neoantigen FLT3(D835Y) is highlighted along with the Mut⁺ only region. **c,** Scatter plots of $E$-scores for mutant peptides versus their corresponding wild-type peptides for three mutations: EGFR(T790M) (left), KRAS(G12D) (middle) and FLT3(D835Y) (right). Red dots indicate known neoantigen peptides reported in the literature. **d,** Schematic of the T cell stimulation assay used to assess the immunogenicity of specific peptides (Methods). T cells isolated from healthy donors were activated and stimulated with peptides, followed by intracellular staining for IFNγ and TNF

for flow cytometric analysis. **e,** $E$-scores for three EGFR(T790M) mutant peptides (p17, p22 and p23; Supplementary Table 8) and their corresponding wild-type peptides presented by HLA-A2301, plotted and compared. **f,** Flow cytometric analysis of dual intracellular IFNγ and TNF staining in donor-derived T cells stimulated with either DMSO (control) or peptide p17 (an EGFR(T790M) mutant peptide). **g,** Bar plots showing the percentage of IFNγ and TNF double-positive T cells stimulated with different peptides or conditions. CEFT, a positive control of 27 peptides (JPT). Error bars represent s.e.m.; $n = 4$ or as plotted individually. ***$P < 0.001$. **h,** Representative flow cytometry plots of T cells, as in **f**, stimulated with DMSO (control) or fusion peptide p11. **i,** Bar plots showing the percentage of IFNγ and TNF double-positive T cells stimulated with various fusion peptides or conditions. Error bars represent s.e.m.; $n = 2$ biological replicates as plotted individually; 2–4 technical replicates each. **$P < 0.01$. All statistics used unpaired two-sided $t$-test. APC, antigen-presenting cell.

## Discussion

The HLA region comprises the most polymorphic segments within the human genome, with over 26,000 identified alleles within the current IMGT/HLA collection[44]. Developing methods to efficiently screen antigen-presenting peptides across numerous HLA alleles could significantly accelerate our understanding of pMHC–TCR biology. This includes the discovery of tumor neoantigens, the search for pathogenic antigens and the mapping of novel TCRs, essential to advancing therapeutic interventions. ESCAPE-seq has enabled population-wide antigen presentation analysis for HLA-I alleles, potentially joining genome-wide association studies as a general approach to understand human variation and disease.

MS-based HLA-I immunopeptidomes allow mapping of HLA-I eluted peptides and have been a mainstay of T cell antigen discovery. This approach is dominated by self peptides from the cellular proteome, limiting the sensitivity to detect clinically important peptides from pathogens and oncogenic proteins[45]. Further challenges for HLA-I antigen identification using MS include high cost, specialized equipment, large quantities of sample material, ambiguous assignment of peptides to specific HLA alleles and potential peptide bias due to cysteine oxidation in MS sample preparation[7,8,13,46]. To address these limitations, DNA sequencing-based HLA immunopeptidome discovery methods have emerged, leveraging large-scale, cost-effective DNA oligonucleotide synthesis for antigen screening[16]. ESCAPE-seq and related methods[16] leverage the cellular ER machinery for quality control of proper and stable pHLA folding as a prerequisite for cell-surface trafficking. Our single-chain design offers several advantages compared with previous methods that separately express HLA and peptide transgenes: (1) increased sensitivity by customizable linkers and cysteine mutations; (2) improved throughput and dynamic range through direct component fusion (Fig. 1f); and (3) simplified implementation without requiring individual HLA allele cell line generation. ESCAPE-seq allowed high-throughput profiling of *HLA-C* alleles that previously could not be studied[16]. Moreover, ESCAPE-seq facilitates combinatorial screening across multiple HLA alleles, setting it apart as an innovative and versatile tool. Unlike MS-based approaches, ESCAPE-seq is not limited by the set of antigen proteins or HLA alleles present in the sample. ESCAPE-seq demonstrated comparable performance to NetMHC on training datasets while identifying true positive neoantigens missed by computational prediction, particularly for *HLA-C* alleles with limited training data. This superior sensitivity in identifying HLA-C-presented antigens reflects insufficient computational training datasets, potentially due to lower surface expression compared with *HLA-A* or *HLA-B* alleles[9]. It was previously shown that *HLA-C* alleles can form larger clusters than the HLA-B alleles, which may compensate for their low surface expression to trigger T cell activation[47]. Thus, ESCAPE-seq could improve prediction tools by providing extensive *HLA-C* training datasets. Our pilot investigation of SARS-CoV-2 spike and nucleocapsid proteins revealed rapid alteration of antigen presentation through variant mutations, indicating a potential immune escape mechanism. Systematic application of methods such as ESCAPE-seq may better inform the immune evasion dynamics of seasonal variants and help identify conserved and broadly recognized epitopes to facilitate vaccine design.

For cancer immunotherapy, the ideal cancer antigen is uniquely present in cancer, exists in each of the cancer cells and cannot be readily lost by the cancer to evade immunity. Driver oncogenic mutations fulfill many of these criteria, and tracking or directing immune response against driver mutations, such as KRAS(G12D), has garnered high interest[48]. Oncogene-derived epitopes are a prime example of high-value HLA ligands that may be missed by current immunopeptidome methods, which require processing many irrelevant but abundant epitopes from self proteins to visualize rare disease-relevant epitopes from mutated proteins. Current knowledge and existing clinical trials are biased toward HLA-A2 alleles, limiting therapeutic accessibility. ESCAPE-seq enables systematic antigen presentation profiling to nominate neoantigen epitopes that are broadly presented by diverse HLAs and are preferentially presented over cognate wild-type sequences. We demonstrated this potential through comprehensive validation of EGFR(T790M) epitope LTSTVQLIM across multiple HLA alleles using multiple orthogonal approaches. Recent HLA monoallelic MS confirmed its presentation by HLA-C*07:01 in cells expressing the T790M minigene[49], while a neoantigen vaccine clinical trial demonstrated presentation by HLA-C*15:02 with specific T cell responses detected via pMHC tetramers[50]. Using engineered monoallelic HLA-C03:04 cell lines and HLA-eluted peptide MS, we further confirmed direct presentation by this additional allele. Moreover, our peptide stimulation assays with HLA-A*23:01+ healthy donor PBMCs revealed significant T cell responses to this epitope, characterized by IFNγ+/TNF+ polyfunctional T cells. These comprehensive validations establish the potential of ESCAPE-seq to identify therapeutically valuable targets across diverse populations.

These insights may prove fruitful for designing the next generation of cancer vaccines or engineered T cell therapies. Nonetheless, the complexity of the native antigen presentation process—encompassing peptide digestion, PLC peptide transfer and variations across cell types—may contribute to differences observed between MS findings and DNA sequencing-based screenings[7,8]. Furthermore, mutations in presented peptides may not necessarily affect TCR recognition, adding another layer of complexity. Therefore, future research efforts could be directed toward developing high-throughput assays for identifying immunogenic antigens and antigen-specific TCRs to advance antigen-directed cancer immunotherapies[51,52]. Beyond HLA-I, we predict ESCAPE-seq could be applied to HLA-II antigen screening by leveraging cellular quality control mechanisms for stable peptide–MHC-II complex selection. Such expansion to HLA-II would enable comprehensive profiling of both CD8+ and CD4+ T cell responses in cancer and infectious diseases.

In summary, ESCAPE-seq is a promising tool for discovery of HLA-presented antigens. ESCAPE-seq can complement NetMHC for initial screening and the immunopeptidome for more targeted screening. Its potential is significant in accelerating the discovery of potential immunogenic antigen peptides for diverse T cell pools, spanning both pathogenic and oncogenic neoantigens. Moreover, its capacity for combinatorial pooling across multiple HLAs in varied settings underscores its promise as an efficient and high-throughput platform for mapping the landscape of antigen presentations across the diversity of the human population.

## Online content

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

[1]Department of Dermatology, Stanford University, Stanford, CA, USA. [2]Department of Genetics, Stanford University, Stanford, CA, USA. [3]Department of Biomedical Data Science, Stanford University School of Medicine, Stanford, CA, USA. [4]Tisch Cancer Institute, Icahn School of Medicine at Mount Sinai, New York, NY, USA. [5]Department of Molecular Microbiology and Immunology, Keck School of Medicine, University of Southern California, Los Angeles, CA, USA. [6]Department of Pathology, Stanford University School of Medicine, Stanford, CA, USA. [7]Ragon Institute of Mass General, MIT and Harvard, Cambridge, MA, USA. [8]Program in Immunology, Harvard Medical School, Boston, MA, USA. [9]Department of Pathology and Immunology, Washington University, St. Louis, MO, USA. [10]Bursky Center for Human Immunology and Immunotherapy, Washington University, St. Louis, MO, USA. [11]Division of Gastroenterology, Massachusetts General Hospital, Boston, MA, USA. [12]Program in Health Sciences and Technology, Harvard Medical School, Massachusetts Institute of Technology, Cambridge, MA, USA. [13]Norris Comprehensive Cancer Center, University of Southern California, Los Angeles, CA, USA. [14]Alfred E. Mann Department of Biomedical Engineering, University of Southern California, Los Angeles, CA, USA. [15]Present address: Amgen Research, South San Francisco, CA, USA. ✉e-mail: bingfeiy@usc.edu; howchang@stanford.edu

## Methods

All research complies with Stanford University's ethical regulations and safety regulations, and was reviewed and approved by both Stanford University ethics panel and biosafety panel.

### DNA synthesis and plasmid construction

All plasmids were made with Gibson assembly (NEB) unless specified otherwise. In brief, based on a lentiviral vector of SCTs pHLA-A0201 and pHLA-A0101 (ref. 51), P2A-puromysin-2A-eGFP was inserted after the cytoplasmic tail of HLA alleles. Then, oligonucleotides encoding different peptides were inserted into the SCT between a signal peptide and B2M gene with a flexible linker. To build the SCT of other MHC alleles, we made the Y84C mutation first (Supplementary Fig. 1d), then inserted it to replace A*02 or A*01 above. For example, the HLA-A*02 was digested out and replaced with B*0702 (Addgene, cat. no. 135509), C*0401 and H2-Kb alleles (synthesized by TwistBio).

Oligonucleotide sequences encoding human HLA alleles were obtained from IEDB (HLA-A0101, HLA-A0201 and HLA-B0702). For each allele, we first made a cloning lentiviral vector with Esp3i sites in the place of peptide in the single-chain format[51], followed by a P2A sequence, eGFP, T2A and puromycin-resistance gene. Afterward, various peptides obtained from literature or the IEDB database (Supplementary Tables 1 and 2) were inserted in place. For HLA–eGFP direct fusion, the P2A sequence between HLA and eGFP above was replaced with a flexible GS-rich linker. Various point mutants and deletion mutants were generated similarly using the above vector as source. To express peptides alone in the ER, a cytomegalovirus (CMV) promoter and the signal peptide from the human growth hormone gene were utilized. The peptide expression was driven in the ER using a lentiviral vector that included a blasticidin-resistance gene. All pooled oligonucleotides encoding peptides were ordered from TwistBio. For benchmarking against the IEDB database, we retrieved the IEDB database and extracted all peptides for four common HLA alleles, A*0101, A*0201, B*0702 and C*0401. Then we randomly selected peptides across all affinities for each allele and combined to make a pool of >2,000 peptides in total. Next, the oligonucleotide pool encoding the peptides flanked by overhang sequences on both sides was inserted into the cloning vector as described above and was electroporated into competent cells (Enduro electrocompetent cells, Biosearch Technologies) using a Bio-Rad MicroPulser. For screening on SARS-CoV-2, DNA sequences encoding spike and nucleocapsid protein from strain Wuhan-Hu-1 were used. A tiling pool was made to cover the full sequencing with a 3-base-pair (bp) shift in-frame between neighbor tiles. All strain variant peptides were included in the pool based on the strain variant data (the GFF file of December 2021 from UniProt).

### Combinatorial pool generation

Fifty HLA-I alleles were selected based on their high frequencies in diverse world populations (https://www.allelefrequencies.net/hla.asp) or their association with disease. Overall, 14 HLA-A, 29 HLA-B and 7 HLA-C were obtained accordingly, which are A*01:01, A*02:01, A*02:05, A*02:12, A*03:01, A*11:01, A*23:01, A*24:02, A*30:01, A*31:01, A*31:08, A*34:01, A*33:03, A*68:01, B*07:02, B*08:01, B*08:02, B*13:01, B*15:01, B*15:02, B*15:03, B*18:01, B*27:01, B*27:05, B*27:02, B*35:01, B*35:02, B*35:08, B*39:06, B*40:01, B*40:06, B*40:10, B*41:01, B*44:0, B*46:01, B*48:03, B*50:01, B*51:01, B*52:01, B*54:01, B*56:01, B*57:01, B*58:01, C*03:04, C*04:01, C*06:02, C*07:01, C*07:02, C*08:02, C*12:03. Sequences encoding human HLA alleles were obtained from IEDB. Some restriction enzyme sites if present in HLA were mutated without changing amino acid and Y84C mutation introduced (Extended Data Fig. 1d). Then part of the B2M sequence, a unique barcode per HLA allele and a long linker containing the 10x TSO sequence were added to the HLA sequence and synthesized by TwistBio. An oligonucleotide pool encoding over 1,500 peptides (cancer driver gene peptides and oncogenic fusion gene peptides) was collected as described below. Then flanking sequences for Gibson assembly were added to either side of

each oligonucleotide and synthesized by TwistBio. The combinatorial paired peptide–HLA plasmid pool was generated by two steps. First, the 50 HLA alleles gene fragments were amplified by PCR with very low cycle number to yield about ~50 ng, as we noticed that PCR with high cycle number produces chimeric HLA alleles due to high homology among the alleles. Alternatively, HLA allele fragments can be inserted into lentiviral vector via digestion and ligation without the PCR step, and then inserted with the peptide pool with the Gibson reaction. This avoids the PCR step that potentially generates chimeric fragments. In this study, we first inserted each HLA-I allele individually to generate 50 cloning vectors. Then we pooled the plasmids to insert peptide pools as described below. This strategy offers the flexibility of customized selection of HLA alleles for different peptide pools of interest in the future.

After that, peptide oligonucleotide pools were amplified for six cycles with PCR. The cloning plasmid pool with 50 HLAs inserted (as described above) was digested with Esp3i and the peptide pool was inserted into it via Gibson to generate the final combinatorial pHLA pooled plasmids. We aimed to get over 50,000 colonies for the first Gibson assembly of 50 HLAs, while the last pHLA pool should give colonies of >100× (1,000× ideally; 7.5–75 million colonies) in terms of library complexity.

We also generated spike-in pools to mix during the experiments. To examine the chimeric reads and recombination of pHLA reads, we generated two spike-in pools separately, with each consisting of 25 HLA alleles with their corresponding peptide pools (Supplementary Table 6).

### Cell line generation and culture

HEK293T cells (ATCC) were cultured in DMEM supplemented with 10% FBS and 1% Penstrep. HLA knockout using Cas9 RNP was done as described previously. Single-guide RNAs were synthesized by Synthego, including ACUGCUACUUCUCGCCGACU (Human TAP1), CUGGUGGGGUACGGGGCUGC (human TAP2), CGGCUACUACAAC-CAGAGCG (HLA_1), AGAUCACACUGACCUGGCAG (HLA-2), AGGU-CAGUGUGAUCUCCGCA (HLA-3). For dual HLA and TAP knockout cells, HLA-knockout HEK293T cells were used to electroporate cas9 RNP mixed with TAP1/TAP2 single guide RNAs. The cells were cultured for 5 days. Then the cells were stained with PE-B2M (BioLegend), sorted into single cells and seeded into 96-well plates. The clonal cell wells were picked and expanded. For each well, its TAP1/2 were verified with genomic PCR and Sanger sequencing.

### Lentivirus production and titration

Lentiviruses were made as described previously[51]. In brief, per six-well plate, HEK293T cells were transfected with a viral expression vector (2 μg), pMD2.G (VSV-G WT) (1 μg) and psPax2 (2 μg) with Lipofectamine 3000. The medium was changed once the next day, and viral supernatant was collected twice at 48 h and 72 h, respectively. The virus was concentrated with 4× Lenti-X according to manufacturer's protocol, and stored as 20× concentrated at −80 °C. For pooled pHLA virus, we used either 6-cm or 10-cm dishes to make larger quantities of virus, with proportional scaling of DNA and reagent amount. The virus was first titrated with HEK293T cells at 25% confluence, and percentage of infection was measured by a flow cytometer (Attune, Lifetechnology).

### Transfection, cell assay and flow cytometry

First, 100,000 cells were seeded onto 24-well plates and cultured overnight. The next day, 0.75 μl of DNA with 1.5 μl of Lipofectamine 3000 (Thermo Fisher) were transfected into cells. The cells were collected 1 or 2 days after transfection and incubated in full DMEM medium for 30 min at 37 °C to recover. Afterward, the cells were pelleted and stained with 2 μl of antibodies (PE anti-B2M from BioLegend, PE-Cy7 anti human HLA-A2 from BioLegend) for 30 min on ice. Cells were washed once before being examined with a flow cytometer (Attune, Lifetechnology).

## Imaging

Cells were transfected with HLA–eGFP fusion construct as described above. The images were taken with a Zeiss LSM780 confocal microscope the next day.

## Pooled antigen presentation screening

The cells were infected with virus at a multiplicity of infection at ~0.15. Puromycin was added to cells after 2 days of infection at 2 μg ml⁻¹ final. The cells were collected after 4–6 days of drug selection. After incubating in fresh DMEM medium for 30 min at 37 °C, the cells were stained with anti-B2M-PE antibody (cat. no. 316306, clone 2M2, from BioLegend) at 2 μl (0.4 μg) per 1 million cells on ice for 30 min with intermittent mixing. Once washed, the cells were resuspended and sorted into four fractions based on PE-B2M intensity by a BD Aria cell sorter. The four bins are evenly distributed on log-scale based on PE-B2M intensity between background peak and signal's maxima. For example, typically we make the center of the negative peak (background peak) at 100, then the low fraction bin centered around 400, the medium bin at 1,600 and the high bin at around 6,400 or higher. We typically require two biological replicates per screen, and start cells about >1,500 times the peptide pool's complexity for each replicate.

## DNA extraction, library generation and sequencing

Sorted cells were spun down and genomic DNA was extracted with a Zymo column (Zymo QuickDNA) according to the manufacturer's protocol, or with lysis and precipitation. In brief, the cells were first resuspended in lysis buffer (20 mM Tris, 5 mM EDTA and 50 mM NaCl, 0.1% SDS), then RNase A and proteinase K (20 mg ml⁻¹ stock solution) were added at 5 μl per 100 μl of solution. The samples were incubated at 37 °C for 30 min then 50 °C overnight. The next day, the aqueous phase containing the DNA was obtained using phenol:chloroform:isoamyl alcohol (Invitrogen) and Maxtract High Density from Qiagen following the manufacturer's protocol. DNA was precipitated with 70% isopropanol following standard protocol.

The library that encodes HLA peptides and barcodes was generated through three rounds of PCR (primers in Supplementary Table 5). First, we enriched the pHLA fragments from genomic DNA by 15 cycles of PCR (98 °C for 3 min, then 15× of 20 s at 98 °C 20 s, 20 s at 58 °C and 60 s at 72 °C) with 0.8 μM primers of amp_GH_Fw and B2M-bc_rev. After cleanup, 5 μl of elution was used for the second round of PCR with 0.8 μM nested primer containing Illumina adapters P7_GH_HLA_fw and P5_BC-B2M_rev (98 °C for 1 min, then 6× of 20 s at 98 °C 20 s, 20 s at 59 °C and 60 s at 72 °C). The above primers were designed in a way compatible with the dual index used in Illumina sequencing platforms. The final libraries were obtained by a third round of index PCR with Illumina Truseq-based index primers (98 °C for 1 min, then 6× of 20 s at 98 °C 20 s, 20 s at 63 °C and 60 s at 72 °C). Due to low complexity of the libraries, typically 25% PhiX Control v3 libraries (Illumina) were spiked in when sequencing by NextSeq or HiSeq.

## Database retrieval and collection

We created three distinct sets of peptides for the evaluation. We selected 2,178 peptides with a broad range of measured affinity values from the IEDB for alleles HLA-A0*1:01, HLA-A*02:01, HLA-B*07:02, HLA-C*04:01. For HLA-C, due to the scarcity of strong binding measurements, we opted for a smaller selection of 150 peptides due to the lack of existing measured affinity values. The *E*-scores of these peptides were measured in four separate experiments, one for each allele.

The sequence of SARS-CoV-2 Wuhan-1 strain used was a wild type The GFF file containing different strain mutations was obtained from UniProt (accession: P0DTC2, as of 2021), from which the lists of mutations per strain were extracted to build the peptide pools. In this study, for simplicity, we considered only the point mutations and in-frame deletions/mutations in spike and nucleocapsid proteins of SARS-CoV-2

(Supplementary Table 3), which gave >2,600 9-mer antigens for alleles HLA-A*01:01, HLA-A*02:01, HLA-B*07:02. We measured their *E*-scores concurrently with the IEDB experiments.

Top cancer driver gene mutations were collected from COSMIC database. In brief, we first manually ranked oncogenes in the database based on occurrence. Then the top prevalent point mutations per gene were collected to get a candidate list. DNA fragments encoding peptides across each mutation were extracted from gene sequences from the NCBI website.

For fusion mutations, we similarly ranked COSMIC fusion genes based on mutation numbers from the COSMIC database, and picked the top ~30 fusion variants (for example, >5–10% or occurrence >50). Per variant, we retrieved the genomic coordinates for the breakpoint first. Then, we obtained fusion genes' DNA and protein sequences based on genomic coordinates using FusionGDB2 (https://compbio. uth.edu/FusionGDB2/index.html). Next, we used the UCSC genome browser to find the DNA sequences for these two genes (version hg19), and aligned DNA sequences to find the breakpoint site on the coding sequence. The breakpoint per fusion was further confirmed by UniProt (https://www.uniprot.org/uniprotkb), where protein sequences for the two genes were aligned to fusion protein sequences obtained above to retrieve peptides across breakpoints. Per mutation/fusion point, a pool of the peptides was obtained by tiling 9–11-amino-acid short peptides across the mutation or breakage point. Altogether, we tested 1,500 top oncogenes from the COSMIC database across 50 diverse alleles from HLA-A, HLA-B and HLA-C.

## Data analysis

Custom Python scripts were built to read the fastq files using Biopython functions and count the paired reads. Here we took only the antigen peptides or HLA barcodes with exact match of their flanking sequence of 6–10 bp. Then, a count table containing read counts for each peptide–HLA pair per sorted bin was built.

## *E*-score calculation and clustering

The transformation of four binned numbers into an *E*-score commences with a normalization of all the reads within each bin through dividing the counts in each bin by the average. This standard procedure in genomic studies was performed to account for any discrepancies that may arise due to variations in sequencing read depth. Differentiating from the conventional RNA sequencing read normalization, we omitted log normalization in our process. Our rationale stems from the presumption that the distributions of our counts are unlikely to conform to a log-normal distribution, and thus log normalization may not provide an accurate representation of our data. Subsequently, we normalized the total number of sequencing reads measured per trimer (pHLA pair). This step is crucial in adjusting for uneven distribution of antigen–HLA reads that were originated from various steps in the ESCAPE-seq experiments, including pooled plasmid cloning, virus production and transduction, and library construction. With the normalization complete, each allele–peptide trimer now possesses normalized counts, indicating the number of cells detected in each respective bin. To synthesize these data into a singular *E*-score, we used the formula: $E\text{-score} = \text{counts}_{bg} \times w_{bg} + \text{counts}_{low} \times w_{low} + \text{counts}_{med} \times w_{med} + \text{counts}_{high} \times w_{high}$ where weight $w$ were asigned as $w_{bg} = 0, w_{low} = 2, w_{med} = 4, w_{high} = 8$ for background (bg), low, medium (med) and high fractions, where we put weight at log-scale that matched with the binning scale during cell sorting, while assigning background bin as 0.

For combinatorial screening with multiple HLA alleles, we further normalized *E*-score for each allele to align their distributions in a more comparable manner. This normalization per allele is instrumental for the alignment of our data to a specific reference. This is to mitigate the influence of varying allele efficacies in presentation and/or their intrinsic stability to escape from ER. To facilitate this, we calculated

the mode of lower peak (Extended Data Fig. 5e) for each allele (presumably from all negative peptides). Following this, we adjusted every $E$-score for that specific allele such that these modes converge. This technique of alignment ensured that the $E$-score was not skewed due to the intrinsic variations between different alleles, thus making the comparison between different alleles more accurate and meaningful.

Heatmaps were generated using the ComplexHeatmap R package. Only peptides with at least one HLA $E$-score above threshold were included. For the heatmap in Fig. 4, the score matrix was clipped to values between 2 and 5 to enhance visualization.

### Evaluation metrics

NetMHC predictions for individual peptides in binding affinity mode (_ba) and eluted ligand mode (_el) were obtained through the NetMHC-4.1 web interface (peptide mode in https://services.health-tech.dtu.dk/services/NetMHCpan-4.1/). Predictions for pooled peptides were obtained through the command line version of NetMHC-4.1 (downloaded from https://downloads.iedb.org/tools/).

When contrasting the measured $IC_{50}$ values, $E$-scores and NetMHCpan predictions, we employed both regression metrics (Pearson's $r$ and Spearman's $r$) and classification metrics (ROC-AUC and PR-AUC), calculated separately for each allele.

The regression analyses were performed after transforming $IC_{50}$ values into a log-transformed version, following the format used in training predictive models: $1 - \log(\text{binding affinity})/\log(50,000)$, as detailed in the source. This transformation resulted in affinity values in the range 0 to 1, with an $IC_{50}$ greater than 0.426 corresponding to an $IC_{50}$ less than 500 nM. Concurrently, for classification analyses involving ROC-AUC and PR-AUC, a thresholding of the labels was implemented. For $IC_{50}$-based labels, including those from IEDB measurements and NetMHCpan predictions, binders were defined as those with $IC_{50}$ values below 500 nM. Specifically, Pearson's and Spearman's correlations were calculated using the modules spearmanr and pearsonr from scipy.stats. ROC and PRC and their AUCs were calculated using functions in the Python package sklearn.metrics. To get the error bars, we visualized the 95% confidence intervals of the metrics across all the alleles. For labels derived from ESCAPE-seq measurements in this paper, binders were classified as entities with an $E$-score greater than 3.2 for single HLA allele screening. For combinatorial ESCAPE-seq, we raised the cutoff to 3.8 due to higher observed noise, presumably from multiple HLA alleles. In general, a cutoff between 3.5 and 4 is reasonable.

Furthermore, for each metric calculated, we established a 95% confidence interval. This was achieved through the utilization of a nonparametric bootstrap method, entailing the generation of 1,000 samples from the original dataset with replacement, thereby offering an estimation of the uncertainty inherent in our metric estimates.

### Evaluating predictions of $E$-scores by NetMHCpan models

To ascertain the degree of alignment between current state-of-the-art model predictions and $E$-scores, we calculated metrics using NetMHCpan-4.1 predictions as an estimation for $E$-scores, employing a threshold of $E$-score > 3.8 for the classification metrics. NetMHCpan-4.1 includes two model variants: a binding affinity model trained to predict $IC_{50}$ values and an eluted ligand model trained to predict ligands identified on the cell surface via MS. Although the eluted ligand model is generally the standard, we observed a superior correlation with $E$-scores in the binding affinity model, which we therefore adopted for comparative metrics.

All metrics indicated a high degree of congruence between model predictions and $E$-scores when utilizing the IEDB data, which formed the training basis for these models. However, when the models were applied to the SARS and oncogene datasets, the metrics were less encouraging. Besides ROC-AUC, which is largely impervious to class imbalances characteristic of the datasets we analyzed, there was a pronounced discrepancy in performance between the IEDB metrics and those derived from the SARS and oncogene datasets.

Previous studies have demonstrated a correlation between the number of peptides in the training data for a given allele and the performance of the model on other peptides of the same allele. However, in our analysis, the number of training examples for a given allele failed to account for most of the variance observed in the model's inaccurate prediction of stability in the oncogene and SARS datasets. This was despite the model demonstrating competent performance when evaluated on the same alleles using peptides from the training set. While a slight dip in performance is predictable when evaluating peptides absent from the training data, a 30–75% performance reduction across most metrics implies an increased challenge posed by these new datasets. The smaller proportion of positive peptides per allele in these datasets could contribute to this difficulty. However, even when resampling the data to match the percentage of positives in the IEDB dataset, a significant gap in prediction ability persisted. This observed drop-off in performance underpins the necessity of the ESCAPE-seq methodology to augment current computational prediction methods and address the identified performance gaps.

### Exploring allele similarity as defined by peptide preferences

To evaluate the similarity among HLA alleles, we may consider three different parameters: sequence similarity, structural similarity or functional similarity (the similarity in their binder peptides). Although crystal structure information is not available for all HLA alleles, sequence data are available, and with the introduction of our new oncogene dataset, we can compare the binding preferences of 50 distinct alleles.

The sequences used to compare alleles are composed of 34 amino acids from each allele, often denoted as MHC pseudo-sequences. These sequences encapsulate all the amino acids within 5 Å of the peptide binding cleft that vary between alleles. We calculate sequence similarity between sequences $s_1$ and $s_2$, using the equation:

$$\text{SequenceSim}(s_1, s_2)$$
$$= 1 - \frac{\sum_{aa_i=1}^{34} \text{AAsim}(s_1[i], s_2[i])}{\sum_{aa_i=1}^{34} \text{AAsim}(s_1[i], s_1[i]) \times \sum_{aa_i=1}^{34} \text{AAsim}(s_2[i], s_2[i])}$$

where $\text{AASim}(a_1, a_2)$ represents the similarity between amino acids $a_1$ and $a_2$ as per the BLOSUM62 matrix, across 34 amino acids (aa1–34).

To quantify the similarity among alleles based on peptide preferences, we consider the list of $E$-scores for all 1,500 peptides in the oncogene pool and calculate the cosine distance between the $E$-scores for all peptides between each allele pair. These cosine distances are used for clustering the alleles and for visualizing alleles in two dimensions. Specifically, the Uniform Manifold Approximation and Projection (UMAP) was constructed with Python umap package (with parameters n_components = 2, random_state = 42, n_neighbors = 5, min_dist = 0.01, metric = 'cosine').

### T cell immunogenicity and T cell reporter assays

Antigen-specific T cell immunogenicity was evaluated by a previously published protocol[53]. In brief, HLA-typed PBMCs from four healthy donors were used. Cryopreserved PBMCs were quickly thawed in a 37 °C water bath and transferred into RPMI medium (Thermo Fisher) containing DNase I (Sigma-Aldrich) at a final concentration of 2 U ml$^{-1}$, spun down and resuspended in X-VIVO 15 medium (Lonza) supplemented with cytokines promoting dendritic cell differentiation, GM-CSF (Peprotech, 1,000 IU ml$^{-1}$), IL-4 (R&D Systems, 500 IU ml$^{-1}$) and Flt3L (Peprotech, 50 ng ml$^{-1}$). Cells were seeded at $10^5$ cells per well in U-bottomed 96-well plates and cultured for 24 h before being stimulated with control reagents or pooled test peptides (custom peptide synthesis, JPT Peptide Technologies), where each peptide

was at a final concentration of 1 µM, together with adjuvants promoting dendritic cell maturation, LPS (Invivogen, 0.1 mg ml⁻¹), R848 (Invivogen, 10 mM) and IL-1β (R&D Systems 10 ng ml⁻¹), in X-VIVO 15 medium. Starting 24 h after stimulation, cells were fed every 2–3 d with cytokines supporting T cell expansion, IL-2 (R&D Systems, 10 IU ml⁻¹), IL-7 (Peprotech, 10 ng ml⁻¹) and IL-15 (Peprotech, 10 ng ml⁻¹), in complete RPMI medium (GIBCO) containing 10% human serum (R10). After 10 d of culture, cells were collected, pooled within groups, washed, resuspended in R10 and seeded at $2 \times 10^5$ cells per well in U-bottomed 96-well plates. Expanded T cells were then re-stimulated with control reagents or 1 µM of test peptides, either pooled or individual, together with 0.5 mg ml⁻¹ of costimulatory antibodies, anti-CD28 (CD28.2, cat. no. 55726, BD Biosciences) and anti-CD49d (clone: 9F10, cat. no. 555502, BD Biosciences), and protein transport inhibitors BD GolgiStop, containing monensin, and BD GolgiPlug, containing brefeldin A, at the manufacturer's recommended concentrations. After 8 h of incubation at 37 °C, cells were processed for intracellular staining for flow cytometry using BD Cytofix/Cytoperm reagents according to the manufacturer's protocol. The following antibodies were used: for surface staining CD3 (clone: SK7, cat. no. 344804, BioLegend), CD4 (clone RPA-T4, BV785) and CD8a (clone: RPA-T8, cat. no. 301049, BioLegend), and for intracellular staining IFN-g (clone: B27, cat. no. 506507, PE) and TNF-a (clone: Mab11, cat. no. 502930, PE/Cy7). All antibodies were purchased from BioLegend. LIVE/DEAD Fixable Aqua Dead Cell Stain Kit (Thermo Fisher) was used for live and dead cell discrimination. Data were acquired using the Invitrogen Attune NxT flow cytometer and FlowJo v.10 was used for analysis. DMSO (Sigma-Aldrich) was used at an equal volume to the test peptides and served as the vehicle/negative control. Significance was evaluated by t-test comparing DMSO versus peptide-specific cytokine formation by CD8⁺ T cells. For the T cell reporter assay in Extended Data Fig. 1c, we transfected HLA/TAP-knockout HEK293T cells with SCTs containing either HLA-A2:SLLMWITQC (NY-ESO-1 epitope) or HLA-A2:NLVPMVATV (CMV epitope). After 48 h, SCT-transfected cells were cocultured with Jurkat NFAT-GFP reporter cells stably expressing NY-ESO-1-specific 1G4 TCR. TCR activation was measured by quantifying GFP⁺ Jurkat cells using a flow cytometer (BD FACSCanto) at coculture time points ranging from 5 min to 24 h.

## Mass spectrometry

MS experiments based on HLA-C*0304 allele were done as previously published[16]. We first transduced wild-type HLA-C*0304 allele with lentivirus with puromycin resistance into HEK293T cells with dual HLA and TAP knockouts as described above. Afterward, a pool of peptides with a range of low and high E-score with HLA-C*0304 from ESCAPE-seq were transduced to the cells with lentivirus carrying the blasticidin-resistance gene. Cells were selected with puromycin and blasticidin and expanded for 2 weeks. Approximately 600 million cells were collected for MS experiments. MS experiments were outsourced to MS works (MI). Peptides (100%) were desalted using solid-phase extraction (SPE) with a Waters µHLB C18 plate. Peptides were loaded directly and eluted using 30:70 acetonitrile:water (0.1% TFA). Eluted peptides were lyophilized and reconstituted in 0.1% TFA. Peptides (50%) were analyzed in analytical duplicate by nano liquid chromatography tandem MS using a Waters NanoAcquity system interfaced to a Thermo Fisher Fusion Lumos mass spectrometer. Peptides were loaded on a trapping column and eluted over a 75-µm analytical column at 350 nl min⁻¹; both columns were packed with Luna C18 resin (Phenomenex). A 2-h gradient was employed. The mass spectrometer was operated using a custom data-dependent method, with MS performed in the Orbitrap at 60,000 full-width at half-maximum (FWHM) resolution and sequential tandem MS performed using high-resolution CID and EThcD in the Orbitrap at 15,000 FWHM resolution. All MS data were acquired from m/z 300–1,600. A 3-s cycle time was employed for all steps.

## Statistics and reproducibility

We usually included two biological replicates for pooled screening and each pooled screen contained negative and positive controls. We required two replicates that were highly correlated (for example, Pearson's correlation > 0.9) for further downstream analysis. For individual experiments, we used 3–4 replicates unless samples were limited (for example, HLA-typed donor PBMCs) or measurements were highly repeatable (in which case, we selected just two samples to report in the manuscript).

## Reporting summary

Further information on research design is available in the Nature Portfolio Reporting Summary linked to this article.

## Data availability

All sequencing files have been deposited to the SRA (accession PRJNA1268025). MS data were included as Supplementary Information. Other data and materials are available upon request. Source data are provided with this paper.

## Code availability

Customized Python scripts were included as Supplementary Information.

## References

53. Cimen Bozkus, C., Blazquez, A. B., Enokida, T. & Bhardwaj, N. A T-cell-based immunogenicity protocol for evaluating human antigen-specific responses. *STAR Protoc.* **2**, 100758 (2021).

## Acknowledgements

We thank members of the Chang and Yu laboratory and R. Schreiber (Washington University) for discussion. This study was supported by National Institutes of Health grant no. U54CA260517 (H.Y.C.), the Parker Institute for Cancer Immunotherapy (B.Y., H.Y.C.), the V Foundation (B.Y.),the Donald E. & Delia B. Baxter Foundation (B.Y.), Margaret E. Early Medical Research Trust (B.Y.) and Robert E. and May R. Wright Foundation (B.Y.). J.B. was supported by an HHMI Hanna Gray Fellowship. A.T.S. is supported by a Lloyd J. Old STAR award from the Cancer Research Institute, a Career Award for Medical Scientists from the Burroughs Wellcome Fund, a Pew-Stewart Scholars for Cancer Research Award and the CRISPR Cures for Cancer Initiative Award. H.Y.C. was an Investigator of the Howard Hughes Medical Institute.

## Author contributions

Q.S., B.Y. and H.Y.C. conceived the idea. Q.S. and B.Y. designed the experiments. Q.S. performed the screening with Z.Z., S.W. and N.Y. Y.Z. and A. Kwong assisted in experiments. Q.S., B.Y. and E.P.S. analyzed the data and J.A.B. assisted with analysis. C.C.B., A. Kaminska, L.V., M.S. and N.B. contributed to the PBMC epitope validation experiment and analysis. C.R.C., L.S.-H. and G.D.G. helped with epitope discovery and prioritization. C.F.L. advised on MS experiments. A.T.S., Y.C., R.R.S. and T.L.R. helped with data interpretation for the project. Q.S., B.Y. and H.Y.C. wrote the manuscript with input from co-authors. H.Y.C. and B.Y. supervised the project.

## Competing interests

Stanford University has filed a patent based on this work in which Q.S., B.Y. and H.Y.C. are named as inventors. H.Y.C. is a co-founder of Accent Therapeutics, Boundless Bio, Cartography Biosciences and Orbital Therapeutics, and an advisor to 10x Genomics, Arsenal Biosciences, Chroma Medicine and Exai Bio until 15 December 2024. H.Y.C. is an employee and stockholder of Amgen as of 16 December 2024. T.L.R. is a co-founder and former Chief Scientific Officer of Arsenal Bioscience, and an advisor to NewLimit. A.T.S. is a co-founder of Immunai, Cartography Biosciences, Santa Ana Bio and Prox Biosciences, is an advisor to 10x Genomics and Wing Venture Capital, and receives

research funding from Astellas Pharma. N.B. is an extramural PICI member. The other authors declare no competing interests.

## Additional information

**Extended data** is available for this paper at https://doi.org/10.1038/s41588-025-02268-1.

**Correspondence and requests for materials** should be addressed to Bingfei Yu or Howard Y. Chang.

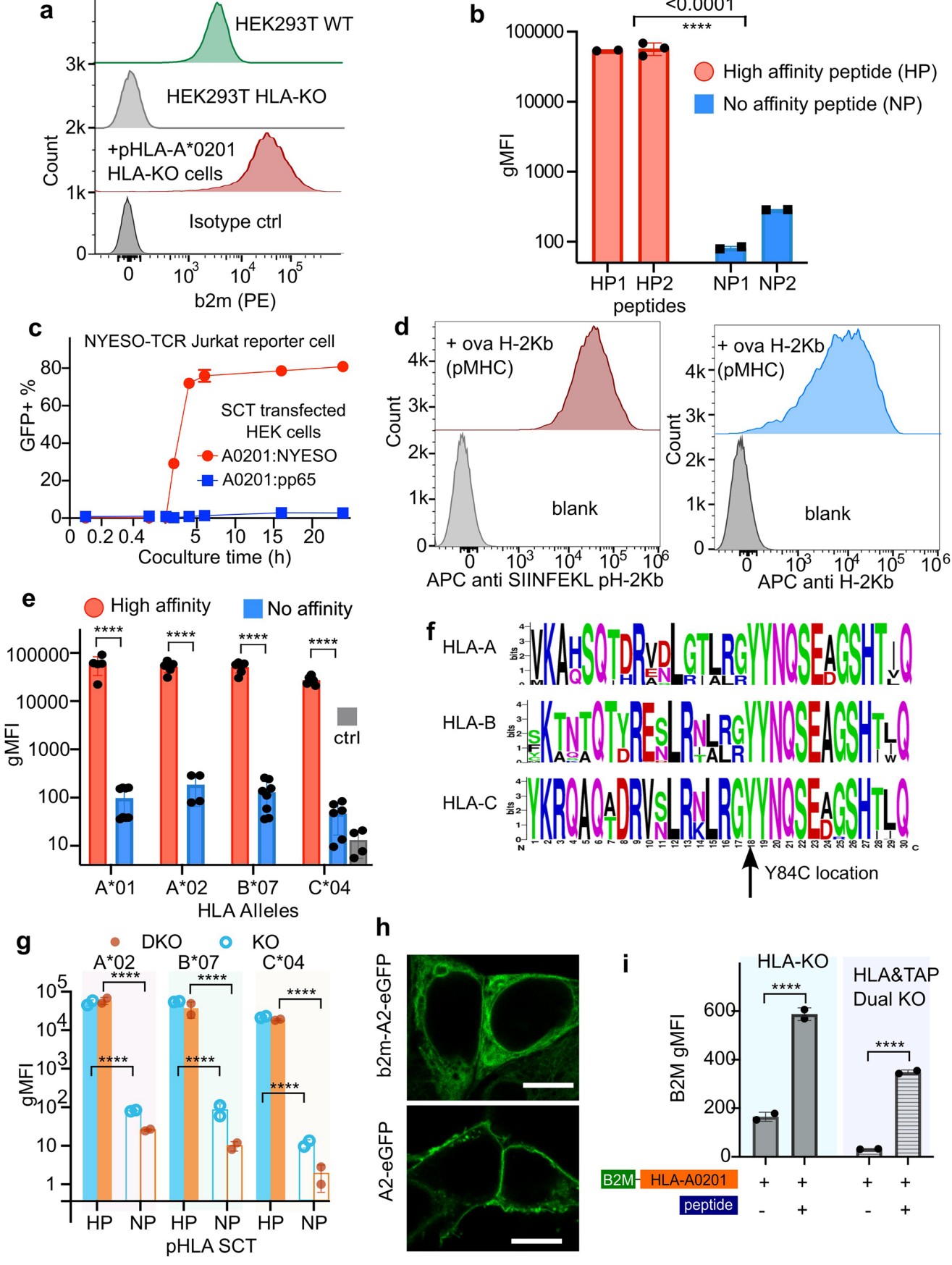

**Extended Data Fig. 1 | See next page for caption.**

**Extended Data Fig. 1 | ESCAPE provides consistent readout across HLA-A, B and C alleles. a**, Histogram of β2m staining in wild-type (WT) HEK293T cells, HEK293T cells with HLA-A, -B, and -C knocked out using CRISPR-Cas9 RNP (HLA-KO), and HLA-KO HEK293T cells expressing NYESO pHLA-A0201 single-chain trimer (SCT). Bottom: isotype antibody staining of HLA-KO HEK293T cells expressing pHLA-A0201. **b**, Bar plots quantifying the geometric mean fluorescence intensity (gMFI) of β2m staining in HEK293T cells expressing SCTs of HLA-A0201 fused to high-affinity (HP) peptides versus peptides with no binding affinity (NP) in Fig.1b. Two peptides were tested for both HP and NP groups. **c**, T-cell stimulation assay showing that NYESO pHLA-A0201-expressing cells induce eGFP expression in T cells co-expressing a cognate 1G4 TCR and an NFAT reporter. In contrast, pp65 pHLA-A0201-expressing cells do not stimulate T cells. **d**, Direct staining of OVA-pH-2Kb-expressing cells using anti-SIINFEKL pH-2Kb and anti-H-2Kb antibodies. **e**, gMFI of cells expressing SCTs fused with HP and NP peptides for HLA-A0101, HLA-A*0201, HLA-B*0702, and HLA-C*0401. **f**, Motif analysis of amino acids surrounding the conserved Y84 residue across all HLA-A, -B, and -C alleles. The arrow indicates the conserved Y84 residue, which is mutated to Y84C in SCT constructs. **g**, Effect of TAP1/2 knock-out. gMFI quantification of HP and NP with A*02, B*07 and C*04 alleles in both HLA KO cells and HLA/TAP dual KO cells. **h**, Fluorescent image of direct eGFP fusion with B2M-A*02 allele (top) and A*02 allele (bottom). **i**, Compare the effect of TAP KO in Fig. 1f. In either HLA KO cells or HLA/TAP dual KO cells, the gMFI of B2M-A2 fusion with or without co-expression of pp65 peptide were compared. *** p < 0.001; **** p < 0.0001. Number of replicates n are depicted on bar plots. Error bars represent standard error of mean (SEM).

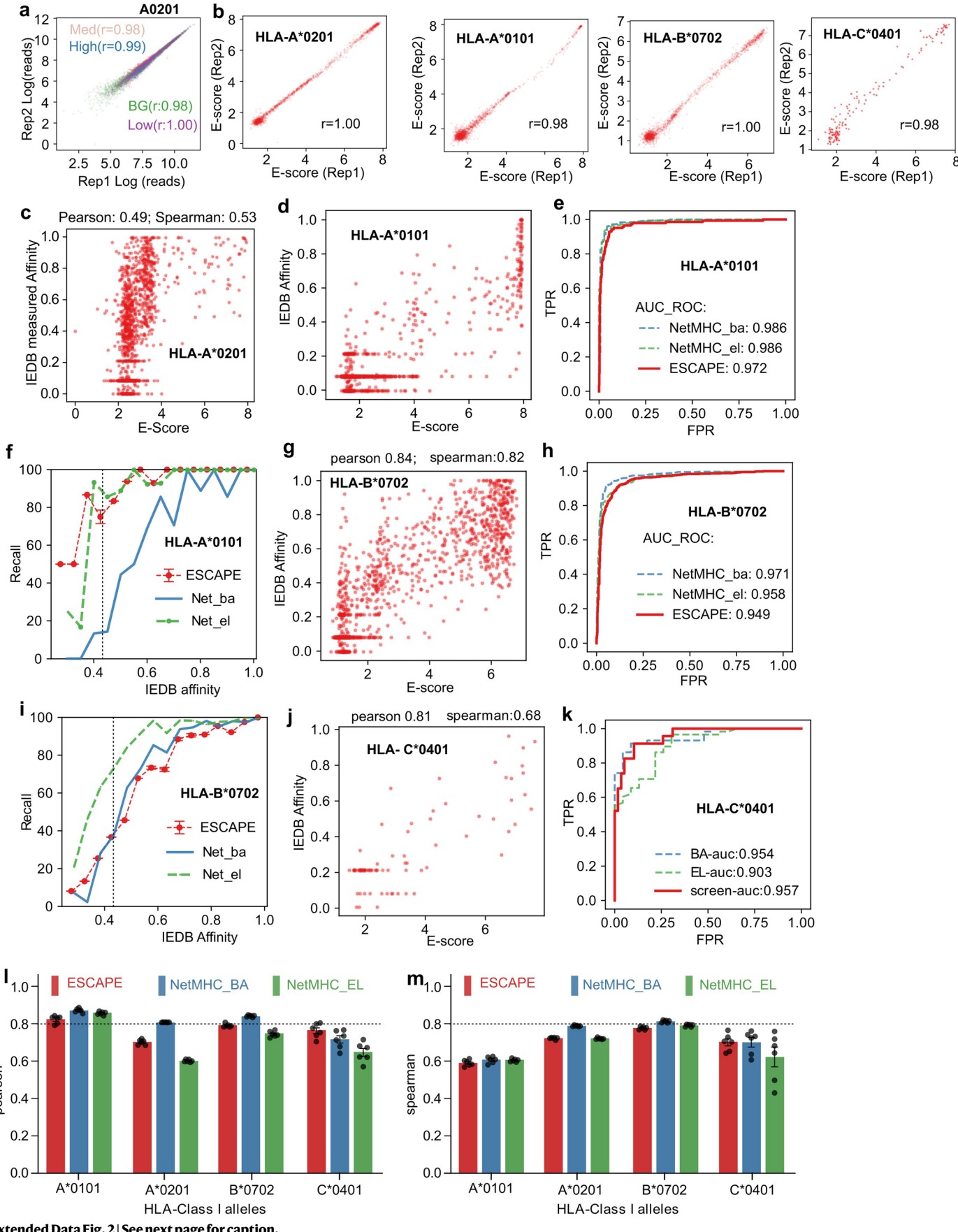

**Extended Data Fig. 2 | See next page for caption.**

**Extended Data Fig. 2 | Comparison of ESCAPE-seq and NetMHC prediction using IEDB training data. a**, Representative correlation plot of read count between replicates for ESCAPE screen. HLA-A*02 was used here an example. 4 fractions (BG/background, low, medium and high) were highlighted in different colors. Pearson correlation coefficients were shown. **b**, Correlation of E-score between replicate of ESCAPE-seq using HLA alleles including A*02, A*01, B*07 and C*04. Pearson correlation coefficients of each were shown in the plots. **c**, Scatter plot of IEDB affinities vs. ESCAPE-seq E-score for wild-type HLA-A*0201 allele. **d**, Scatter plot of IEDB affinities vs. ESCAPE-seq E-score for HLA-A*0101 allele

without Y84C mutation. **e**, ROC curves of ESCAPE, NetMHC_el and NetMHC_ba as comparison using IEDB affinity as ground truth. **f**, Plot of recall rate of ESCAPE-seq vs. IEDB affinity for HLA-A*0101 allele. NetMHC_ba and NetMHC_el prediction results were shown as a comparison. **g-i**, Similar plots as (**d-f**) for HLA-B*0702 allele. **j-k**, Similar plots as (**c,e**) for HLA-C*0401 allele. **l**, Comparison of Pearson correlation coefficient for ESCAPE-seq, NetMHC_ba and NetMHC_el predictions across 4 HLA alleles examined here. **m**, Similar to (k) for Spearman correlation coefficient. N = 6, error bars represents SEM.

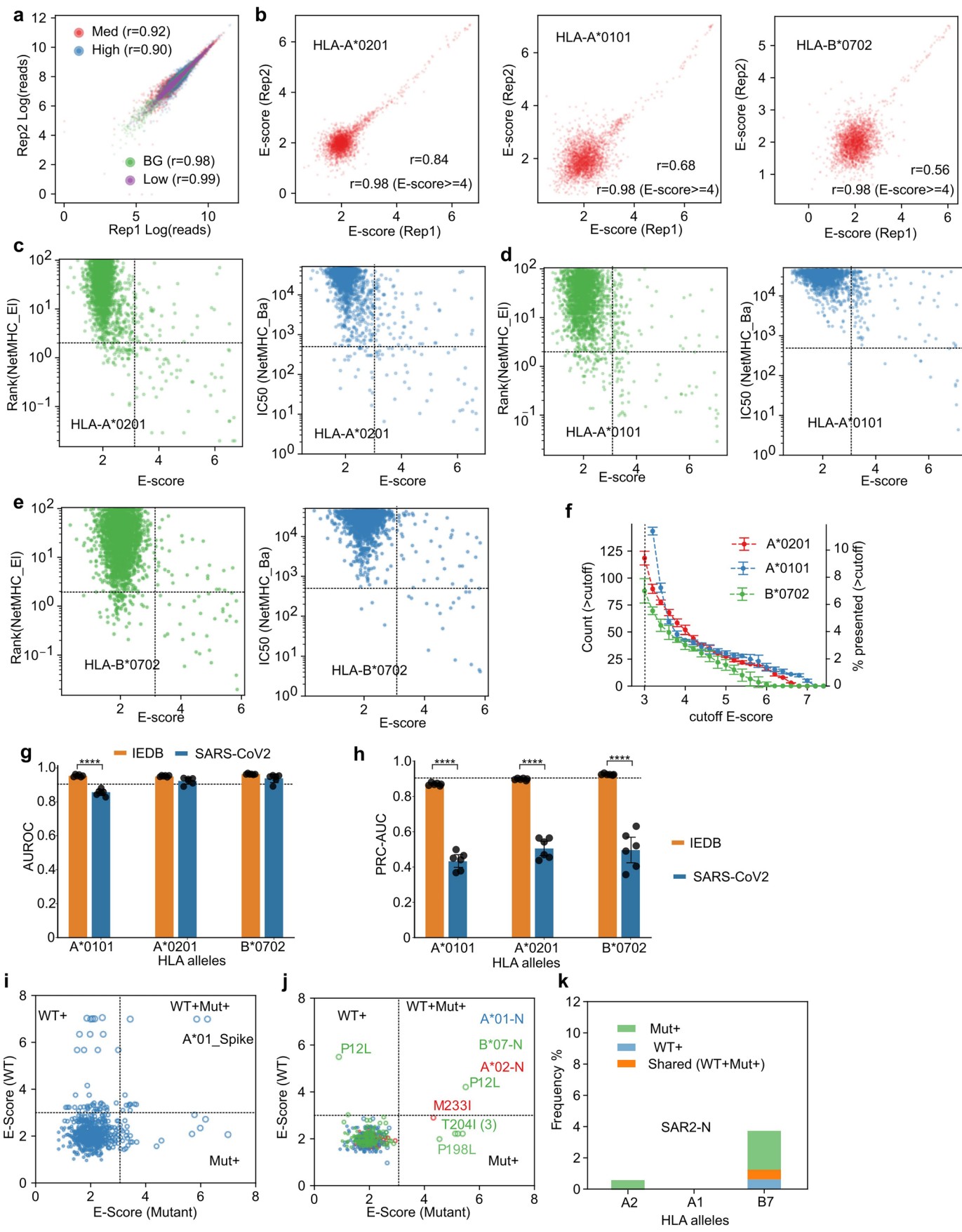

**Extended Data Fig. 3 | See next page for caption.**

**Extended Data Fig. 3 | Comparison of ESCAPE-seq and NetMHC prediction on peptides from SARS-CoV-2 virus. a**, Representative correlation plot of read count between replicates for HLA-A*0201. Pearson correlation coefficients were shown. **b**, Correlation of E-score between biological replicates for SARS antigen screening for HLA-A*0201, HLA-A*0101 and HLA-B*0702 alleles. Pearson correlation coefficients were shown in each plot. **c**, Plot of NetMHC_el rank and NetMHC_ba vs. E-score for HLA-A*0201. **d**, The same plot as (c) for HLA-A*0101. **e**, The same plots as (c) for HLA-B*0702. **f**, Plots of count and percentage of presentable peptides (y-axis) vs E-score cutoffs. Error bars are SEM from n = 2 biological replicates. 3 HLA alleles are compared. Replicate n = 2. **g**, Bar plot of ROC-AUC of NetMHC with ESCAPE E-scores of peptides from IEDB training database (IEDB) and from SARS-CoV2 proteins (SARS-CoV2) for A*01, A*02 and B*07 3 alleles in x-axis. **h**, The same plot as (g) for PRC-AUC metrics across the 2 peptide pools. **** notes p-value < 0.0001; n = 6. **i**, Scatter plots of E-score of antigen peptides containing variant mutation (Mut, x-axis) vs E-score of its corresponding wild-type peptide (WT, y-axis) for spike-HLA-A*0101 pairs. **j**, The same plots for protein N pHLA pairs for all 3 alleles. **k**, Quantification of percentage of presented peptides in each group in S3j.

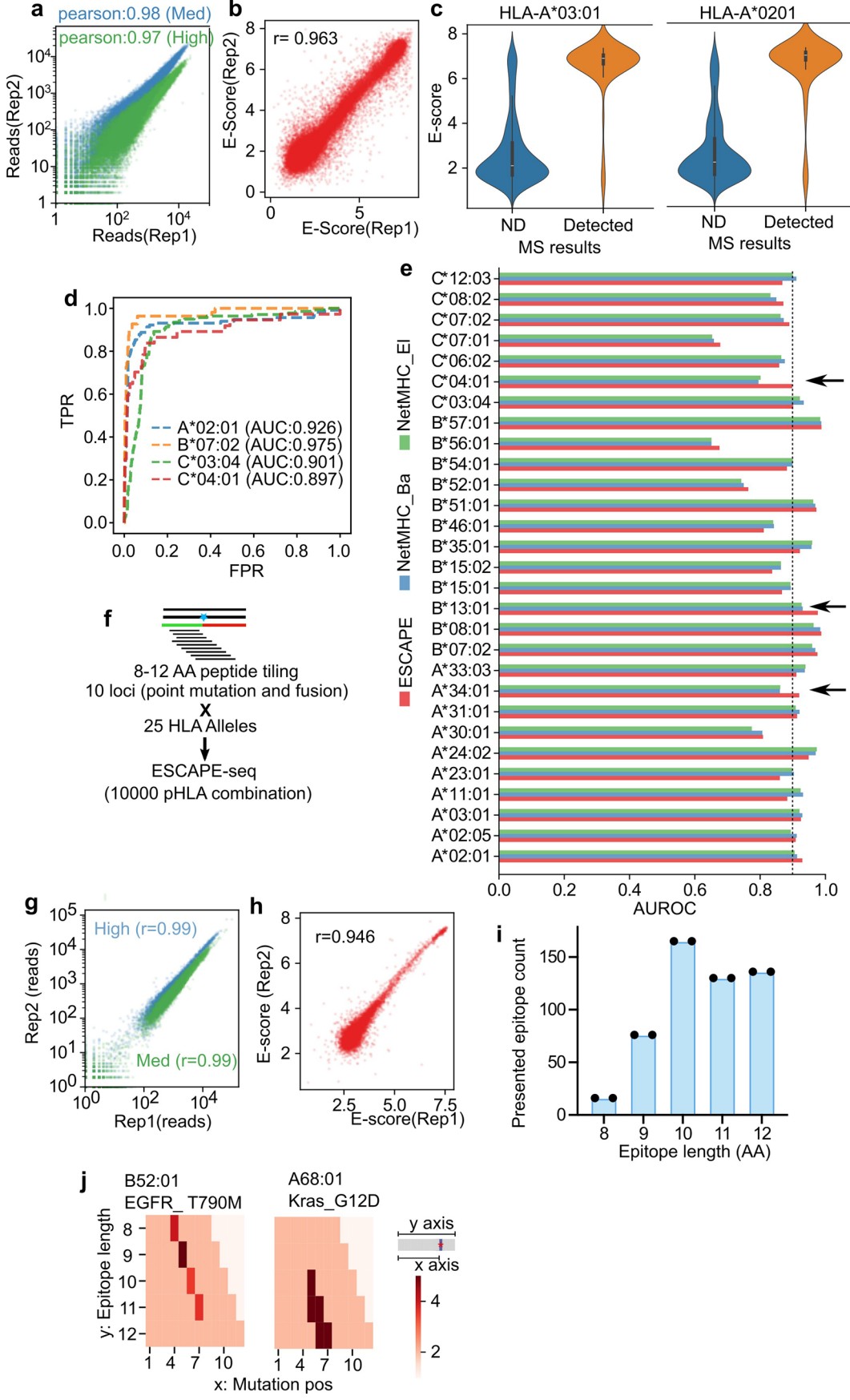

Extended Data Fig. 4 | See next page for caption.

**Extended Data Fig. 4 | Characterization of peptide length and performance of ESCAPE-seq using MS immunopeptidome data. a**, Representative correlation plot of read counts between replicates for ESCAPE-seq performed on a pool of peptides from the MS dataset. Pearson correlation coefficients are shown in the plot. **b**, Scatter plot of calculated E-scores between replicates from (**a**). **c**, E-score distributions for peptides that were not detected (ND) or detected by MS. Examples are shown for the HLA-A*03:01 and HLA-A*02:01 alleles. **d**, Representative ROC curves for 4 alleles (A*02:01, B*07:02, C*03:04, and C*04:01) out of a total of 30 alleles in the pool. **e**, Across all 30 alleles, comparison of E-scores with NetMHC predictions in both EL (eluted ligand) and BA (binding affinity) modes. Arrows highlight alleles where E-scores show better AUC-ROC metrics. **f**, Schematic of ESCAPE-seq screening for peptides of 8–12 amino acids

in length across 25 HLA alleles. **g**, Scatter plots of read counts between biological replicates, and **h**, scatter plots of calculated E-scores between biological replicates. Pearson correlation coefficients are shown in each plot. **i**, Bar plot showing the number of presented peptides identified by ESCAPE-seq at different peptide lengths. **j**, E-score heatmap for peptides generated from the same gene locus with varying lengths per allele. Examples include peptides derived from EGFR T790M presented by HLA-B*52:01 and peptides tiling across KRAS G12D presented by HLA-A*68:01. The heatmap illustrates changes in E-scores when an epitope is extended by one amino acid at either the N- or C-terminus. In some cases, peptide presentability remains consistent when an additional amino acid is added to the N-terminus (left and right panels) or the C-terminus (right panel).

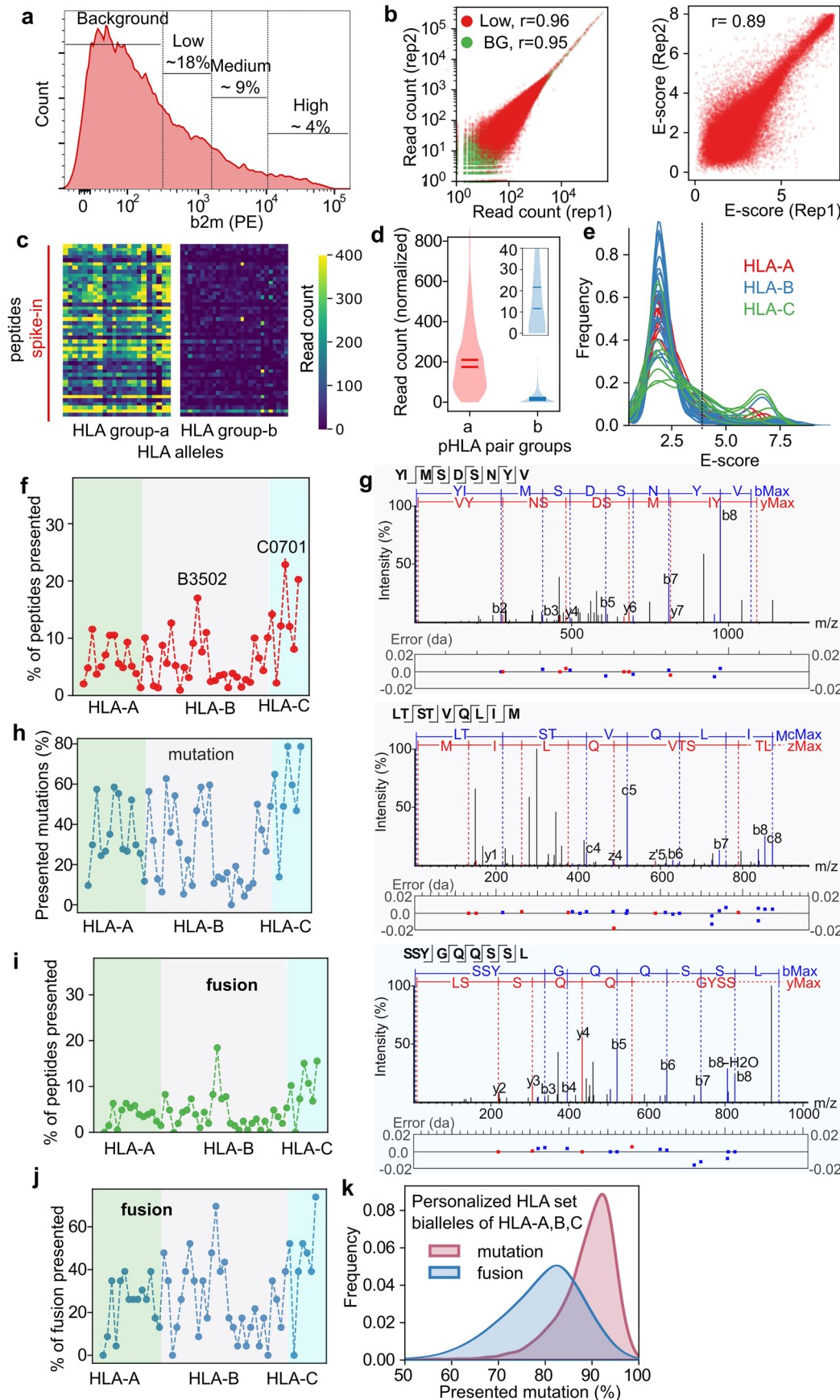

**Extended Data Fig. 5 | See next page for caption.**

**Extended Data Fig. 5 | Combinatorial ESCAPE-seq benchmark and MS validation of selected cancer neoantigens. a**, Representative histogram of B2M staining of HEK-KO cells transduced with the combinatorial virus pool. 4 fractions (termed Background, Low, Medium and High) were sorted based on staining signals on log-scale. **b**, The correlation of reads (**left**) and E-score (**right**) of combinatorial ESCAPE-seq. **c**, Heatmap showing the normalized read count for spike-in peptide HLA pairs. Here the input pool only consisted of spike-in peptide paired with HLA group-a. All reads from paired group-b were recombination events from various steps involved in the screen, such as template switch in virus pool, chimeric reads from PCR amplification and library generation, or during sequencing. **d**, The violin plot showed the quantification of read counts in (**c**). In each violin plot, the top and lower bars represent the mean and median.

inset: zoom in plot. **e**, Normalized distribution of E-score for each HLA-I allele, colored in HLA-A, B, C subtypes. **f**, Percentage of presented peptides varies across different HLA alleles. **g**, Intensity traces show the detection of peptides (YIMSDSNYV, LTSTVQLIM and SSYGQQSSL) with Mass spectroscopy (MS). **h**, Plot of the percentage of driver mutations from (Fig. 5e) that were presented by each HLAs in x-axis. **i**, Presented peptide percentage for oncogenic fusion peptides. **j**, Plot of presented fusion rate from (Fig. 5f) across HLA-I. **k**, Histogram of percentage of driver mutations (red line) or gene fusion mutations (blue line) can be presented by diploid human cells, where we repeatedly sampled 2 alleles from HLA-A, B, C alleles 1000 times to calculate the mutation coverage to generate the distribution.

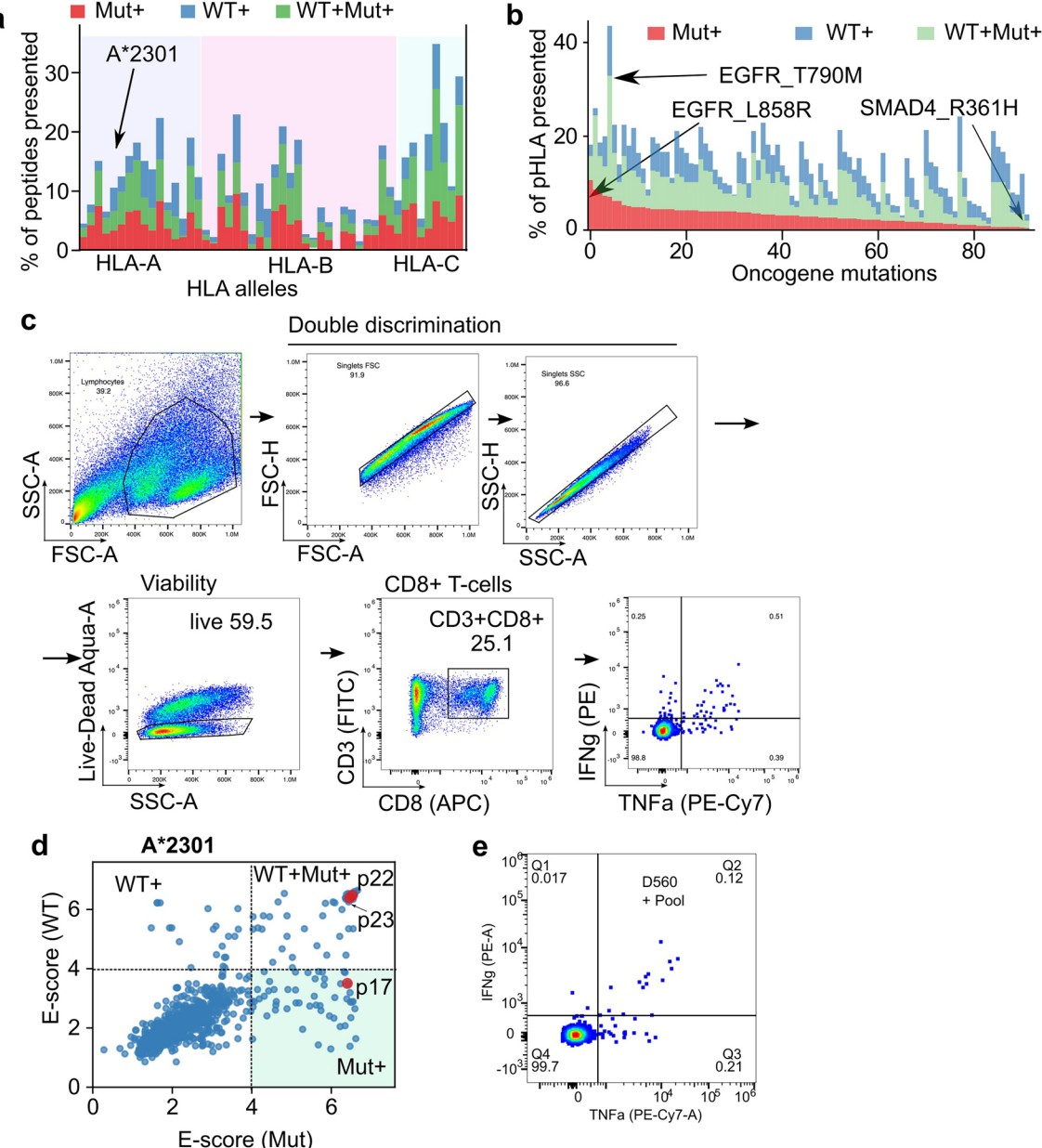

**Extended Data Fig. 6 | Characterization of WT vs Mut antigen presentation across HLA alleles and validation with T-cell stimulation assay. a,** Bar plot showing the percentage of presented peptides in the WT + , Mut + , and WT+Mut+ (both) groups, as depicted in Fig. 6a-b, across all HLA alleles examined in this study. The arrow highlights the bar for HLA-A*23:01. **b,** Bar plot showing the percentage of presented peptides in the WT + , Mut + , and WT+Mut+ (both) groups, as depicted in Fig. 6c, across different oncogene mutations, ranked by the percentage of peptides in the Mut+ group. **c,** Gating strategy for the T-cell

stimulation assay. Cells were gated sequentially by FSC/SSC, single cells, live cells, and CD8 + T cells. Double-positive cells for IFN-γ and TNF were quantified and used in the bar plots shown in Fig. 6g and Fig. 6i. **d,** Scatter plot of E-scores for mutant peptides (x-axis) versus the E-scores of their corresponding wild-type (WT) peptides (y-axis) for the HLA-A*2301 allele. Dashed lines indicate E-score thresholds that categorize peptides into four groups. Peptides from the EGFR T790M mutation (p17, p22, and p23) are labeled. **e,** Flow cytometry plot corresponding to Fig. 6h, showing stimulation by a peptide pool.

# Reporting Summary

## Statistics

For all statistical analyses, confirm that the following items are present in the figure legend, table legend, main text, or Methods section.

| n/a | Confirmed | |
|---|---|---|
| ☐ | ☒ | The exact sample size (*n*) for each experimental group/condition, given as a discrete number and unit of measurement |
| ☐ | ☒ | A statement on whether measurements were taken from distinct samples or whether the same sample was measured repeatedly |
| ☐ | ☒ | The statistical test(s) used AND whether they are one- or two-sided<br>*Only common tests should be described solely by name; describe more complex techniques in the Methods section.* |
| ☒ | ☐ | A description of all covariates tested |
| ☒ | ☐ | A description of any assumptions or corrections, such as tests of normality and adjustment for multiple comparisons |
| ☐ | ☒ | A full description of the statistical parameters including central tendency (e.g. means) or other basic estimates (e.g. regression coefficient) AND variation (e.g. standard deviation) or associated estimates of uncertainty (e.g. confidence intervals) |
| ☒ | ☐ | For null hypothesis testing, the test statistic (e.g. *F*, *t*, *r*) with confidence intervals, effect sizes, degrees of freedom and *P* value noted<br>*Give P values as exact values whenever suitable.* |
| ☒ | ☐ | For Bayesian analysis, information on the choice of priors and Markov chain Monte Carlo settings |
| ☒ | ☐ | For hierarchical and complex designs, identification of the appropriate level for tests and full reporting of outcomes |
| ☐ | ☒ | Estimates of effect sizes (e.g. Cohen's *d*, Pearson's *r*), indicating how they were calculated |

*Our web collection on statistics for biologists contains articles on many of the points above.*

## Software and code

Policy information about availability of computer code

| Data collection | Data collection was described in the Method section. In particular, the flow data was acquired with commercial Attune software (Thermofisher). Sequencing data was acquired with commercial Illumina Sequencer (Nextseq 550 or Novaseq). FACS data was collected using BD Aria. |
|---|---|
| Data analysis | Flow data were analyzed with commercial software Flowjo (v10.10). Sequence data were analyzed with custom python script (py2.7). Some very common python packages were used for analysis, including Biopython (v1.7), Pandas(0.24), and Matplotlib (v2.2). Codes scripts are included in Supplementary data files. |

For manuscripts utilizing custom algorithms or software that are central to the research but not yet described in published literature, software must be made available to editors and reviewers. We strongly encourage code deposition in a community repository (e.g. GitHub). See the Nature Portfolio guidelines for submitting code & software for further information.

## Data

Policy information about availability of data

All manuscripts must include a data availability statement. This statement should provide the following information, where applicable:
- Accession codes, unique identifiers, or web links for publicly available datasets
- A description of any restrictions on data availability
- For clinical datasets or third party data, please ensure that the statement adheres to our policy

> All sequencing files were deposited to SRA. Accession number was provided. Other data, materials and analysis scripts are available upon request. Details were described in Data availability statement

## Research involving human participants, their data, or biological material

Policy information about studies with human participants or human data. See also policy information about sex, gender (identity/presentation), and sexual orientation and race, ethnicity and racism.

| | |
|---|---|
| Reporting on sex and gender | n/a |
| Reporting on race, ethnicity, or other socially relevant groupings | n/a |
| Population characteristics | n/a |
| Recruitment | n/a |
| Ethics oversight | n/a |

Note that full information on the approval of the study protocol must also be provided in the manuscript.

# Field-specific reporting

Please select the one below that is the best fit for your research. If you are not sure, read the appropriate sections before making your selection.

☒ Life sciences  ☐ Behavioural & social sciences  ☐ Ecological, evolutionary & environmental sciences

For a reference copy of the document with all sections, see nature.com/documents/nr-reporting-summary-flat.pdf

# Life sciences study design

All studies must disclose on these points even when the disclosure is negative.

| | |
|---|---|
| Sample size | For individual validation, we perform at least 2 biological replicates to check the variation, and do more replicates when see a larger variation. In the pooled screening, we used cell number of >1000 times of the input library size in order to cover the distribution of input variants, on the range of 5million to 100million cells per screening, and replicated twice. We included Statistics and reproducibility statement. |
| Data exclusions | in the screening, each candidate (a peptide-HLA pair), if its read is low (e.g. <40), is excluded for downstream analysis. This is due to the fact that when read count is low, there is more error in all measurements, thus the result will not be reliable. So it is better to exclude those data point. |
| Replication | We performed at least 2 biological replicates in all our experiments to check the variation. We did more replicates when observing a larger variation (e.g. Coefficient of Variation> 10%). all replications are highly correlated and reproducible. |
| Randomization | We only used cells in our assay here instead of animals. We take cells randomly but with a fix number depending on experimental purpose. |
| Blinding | Blinding maybe is not applicable to our screening case here. But the investigator did not have any prior knowledge about the hits (i.e. peptide-HLA pairs) until the data were analyzed at the end. So in this persepctive, investigators were blinding to samples/results. |

# Reporting for specific materials, systems and methods

We require information from authors about some types of materials, experimental systems and methods used in many studies. Here, indicate whether each material, system or method listed is relevant to your study. If you are not sure if a list item applies to your research, read the appropriate section before selecting a response.

## Materials & experimental systems

| n/a | Involved in the study |
|---|---|
| ☐ | ☒ Antibodies |
| ☐ | ☒ Eukaryotic cell lines |
| ☒ | ☐ Palaeontology and archaeology |
| ☒ | ☐ Animals and other organisms |
| ☒ | ☐ Clinical data |
| ☒ | ☐ Dual use research of concern |
| ☒ | ☐ Plants |

## Methods

| n/a | Involved in the study |
|---|---|
| ☒ | ☐ ChIP-seq |
| ☐ | ☒ Flow cytometry |
| ☒ | ☐ MRI-based neuroimaging |

# Antibodies

| | |
|---|---|
| Antibodies used | PE anti-b2m (clone 2M2); Pe-Cy7 anti human HLA-A2 (clone BB7.2),CD3 (clone SK7, FITC), CD4 (clone RPA-T4, BV785) and CD8a (clone RPA-T8, APC), IFN-g (clone B27, PE) and TNF-a (clone Mab11, PE/Cy7) from Biolegend. anti-CD28 (BD Biosciences) and anti-CD49d (BD Biosciences). Usage and dilution follows manufacturer's protocol. for screening, we use 2ul (0.2mg/ml) of b2m antibody per 1M cells. |
| Validation | We follow manufacturer's protocol and previous published amount to stain the cells. specifically antibodies are from Biolegend and BD with their validation information at https://www.biolegend.com/fr-fr/bio-bits/highly-specific-validated-antibodies and https://www.bdbiosciences.com/en-us/products/reagents/flow-cytometry-reagents/research-reagents/quality-and-reproducibility |

# Eukaryotic cell lines

Policy information about cell lines and Sex and Gender in Research

| | |
|---|---|
| Cell line source(s) | HEK293T cells from ATCC, with its endogenous HLA knocked-out, PBMC donor samples purchased from https://immunospot.com/ |
| Authentication | HEK293T cells were purchased from and validated by ATCC. |
| Mycoplasma contamination | Mycoplasma test was done with Lonza MycoAlert kit and/or InvivoGene MycoStrips. tested negative. |
| Commonly misidentified lines (See ICLAC register) | no commonly misidentified cell lines were used in the study |

# Plants

| | |
|---|---|
| Seed stocks | n/a |
| Novel plant genotypes | n/a |
| Authentication | n/a |

# Flow Cytometry

## Plots

Confirm that:

☒ The axis labels state the marker and fluorochrome used (e.g. CD4-FITC).

☒ The axis scales are clearly visible. Include numbers along axes only for bottom left plot of group (a 'group' is an analysis of identical markers).

☒ All plots are contour plots with outliers or pseudocolor plots.

☒ A numerical value for number of cells or percentage (with statistics) is provided.

## Methodology

| | |
|---|---|
| Sample preparation | HEK293T with endogenous HLA knock-out cells were used to stain with specific antibodies. As described in the Method, cells |

| | |
|---|---|
| Sample preparation | were trypsinized, incubated after wash before staining of antibody at 4c for 30min; then wash and run flow cytometry. |
| Instrument | We used Attune flowcytometry (Thermofisher) for flow analysis only; and used BD FACSAria (BD biosciences) for cell sorting. |
| Software | For Attune, the software coming along with the instrument were used for data collection. BD Diva software was used for FACS. Afterward, Flowjo was used for analysis |
| Cell population abundance | In our screening, we aimed to have live cell count at least 1000x of input diversity; this strategy was shown by previous publications to be able to reliably capture the distribution of input variants well for screening experiments. Then we sorted the cells in purity mode. For example, with 75000 variants in our oncogene pHLA pool, we sorted over 100M live cells per replicates. The last gated cell population were sorted into 4 bins with different staining intensity. |
| Gating strategy | The flow plots were first gated with SSC vs FSC to get live cell population; next gated on single cells to remove doublets or aggregate with FSC-H vs FSC-A plot. Afterward, cells with successfully transduced pHLA variant was gated based on eGFP marker; and finally gated with the PE-anti b2m staining and sort the cell population into 4 bins. To distinguish positive and negative, a negative control either wihout eGFP or without staining, or with antibody staining but no surface expression, were used. Since our staining or eGFP signals were very bright, the positive and negative were separately quite far between each other and easy to tell. |

☒ Tick this box to confirm that a figure exemplifying the gating strategy is provided in the Supplementary Information.

