## [Peer Review File · Nature Genetics]

Massively parallel immunopeptidome by DNA sequencing provides insights into cancer antigen presentation

Corresponding Author: Professor Howard Chang

Version 0:

Decision Letter:

19th Apr 2024

Dear Howard,

Your Technical Report, entitled "Massively parallel immunopeptidome by DNA sequencing reveals landscape of cancer antigen presentation", has now been seen by 3 referees. I apologize for the long review process.

You will see from the reviewers' comments copied below that while they find your work of considerable potential interest, they have raised quite substantial concerns that must be comprehensively addressed. In light of these comments, we cannot accept the manuscript for publication, but would be interested in considering a revised version that addresses these serious concerns.

Reviewer #1 is broadly supportive and has some helpful suggestions for improvement (e.g. looking at class II HLA). Reviewer #2 is positive overall about the technical quality of the work but highlights that the degree of novelty is partially limited by Chour et al. (<https://www.nature.com/articles/s42003-023-04899-8>; not cited in the manuscript). Reviewer #3 doesn't flag major technical flaws but believes that the novelty is partially limited by Bruno et al. (<https://www.nature.com/articles/s41587-022-01566-x>; ref. 15), which presented some validation work. They'd want to see more validation work being carried out, similar to ref. 15.

We hope you will find the referees' comments useful as you decide how to proceed. If you wish to submit a substantially revised manuscript, please bear in mind that we will be reluctant to approach the referees again in the absence of major revisions.

If you choose to revise your manuscript taking into account all reviewer comments, please highlight all changes in the manuscript text file. At this stage we will need you to upload a copy of the manuscript in MS Word .docx or similar editable format.

We are committed to providing a fair and constructive peer-review process. Do not hesitate to contact me if there are specific requests from the reviewers that you believe are technically impossible or unlikely to yield a meaningful outcome.

*2) If you have not done so already please begin to revise your manuscript so that it conforms to our Technical Report format instructions, available [here](http://www.nature.com/ng/authors/article_types/index.html). Refer also to any guidelines provided in this letter.

*3) Include a revised version of any required Reporting Summary: <https://www.nature.com/documents/nr-reporting-summary.pdf>

Please be aware of our [guidelines](https://www.nature.com/nature-research/editorial-policies/image-integrity) on digital image standards.

Link Redacted

If you wish to submit a suitably revised manuscript we would hope to receive it within 6 months. If you cannot send it within this time, please let us know. We will be happy to consider your revision so long as nothing similar has been accepted for publication at Nature Genetics or published elsewhere. Should your manuscript be substantially delayed without notifying us in advance and your article is eventually published, the received date would be that of the revised, not the original, version.

Thank you for the opportunity to review your work.

Sincerely,

Tiago

Tiago Faial, PhD
Chief Editor
Nature Genetics
<https://orcid.org/0000-0003-0864-1200>

Reviewers' Comments:

Reviewer #1:

Remarks to the Author:

Massively parallel immunopeptidome by DNA sequencing reveals landscape of cancer antigen presentation

I read this paper with great interest, and found the study a really impressive one. I think the experimental strategy presented by the author is compelling, and seems to at least recapitulate MHC binding predictions from netMHC. In general I enjoyed the paper, and am supportive of its eventual publication. My major concerns included the lack of confirmation of key antigens. I also am curious to know how well their strategy works on class II HLA, which is now not included at all in their study.

(1) One key point to me is that overall it does appear that netMHC and ESCAPE-seq perform well. It appears that in COVID antigens there is some discrepancy (Supplementary Figure 3g, 3h). Similarly for neoantigens (Fig 4c). I think it is important to understand whether netMHC is underperforming ESCAPE-seq, or vice versa. I think it is also important to understand the situations where the predictions are discrepant. Is it indeed the case that ESCAPE-seq predicts binding better somehow? As the authors point out IEDB is used to train netMHC, so perhaps it is more compelling that ESCAPE-seq obtains comparable performance. It isn't clear in the data which is IEDB independent, that ESCAPE-seq is getting the right answers more often. I think it is important for the authors to experimentally validate some of their predictions to clearly show the benefit of their strategy in scoring new antigens. I think this would demonstrate the value of their experimental strategy. To me this is critical – since without this sort of evidence, netMHC is the easier choice.

(2) I notice that class II HLA is in fact not addressed at all. Does their strategy work for class II HLA? It is possible that their method is class I specific, in which case this should be clearly stated.

(3) For Figure 2d, the authors present a heatmap clustering HLA alleles, can the authors please label the individual HLA alleles. I see there is a large HLA-B block, A block, and C block. It would be interesting to see if the individual allele names to see if alleles with similar 2 digit designations group together. According to the UMAP some of the alleles cluster with

alleles from a different gene. For example A4301 appears to cluster with HLAB genes. Is that similar to the heat map?

(4) The paper's title is: "Massively parallel immunopeptidome by DNA sequencing reveals landscape of cancer antigen presentation". I think this is somewhat misleading. I think the paper presents a powerful method to do high throughput characterization of class I antigen binding presentation. I am unsure about the value that this paper adds to the cancer field. They apply their method to cancer antigens, but none of those results are validated or taken further in any meaningful way. Authors should alter the title and the abstract to reflect the methodological aspect, which is the main contribution here.

(5) While the methods section is extensive, the authors should make their detailed protocols public on protocols.io or some such place so that their methods can be reproducibly implemented by others.

Minor Comments:

(1) For the some of the reproducibility plots (e.g. Supp Figure 3) it would be useful to add test-retest correlation values to quantify the reproducibility.

(2) For figures where authors use barplots (Supp Fig 3g, Supp Figure 3h, Figure 2g/h) individual data points should actually be shown, so as not to obscure the underlying data.

(3) Figure 5f. Do the authors have a clear strategy to identify those antigens with discrepant Escores for wild type and mutant? They should color those antigens differently on the plot to make it clear.

Reviewer #2:

Remarks to the Author:

A. Summary of the key results:

The authors have developed a DNA sequencing-based method, called ESCAPE-seq, for peptide-HLA combinations screening with pMHC single-chain trimers (pMHC SCT), which consisted of amino acid antigen peptide, B2M and MHC alleles. They investigated thousands of peptides from SARS-CoV-2 variants, tumor hotspot mutations, and fusion variants across 50 diverse HLA class I alleles with ESCAPE-seq, which greatly underscores its significant applicability and potential in cancer immunotherapy.

B. Originality and significance: if not novel, please include reference

The overall design of ESCAPE-seq approach is elegant. However, William Chour et al. have already adapted single-chain trimer technologies for pMHC library generation across multiple Class I HLA alleles (Communications Biology, 2023 6, 528). They also constructed SCT libraries for SARS-CoV-2 variants, tumor-associated antigens and common viral strains (CMV, EBV, influenza, and rotavirus), although the scale and throughput are not as high as ESCAPE-seq. So, in that sense the paper is not hugely novel. The potential novelty is that the authors develop an algorithm to assign ESCAPE-scores to each peptide to evaluate the peptide binding affinity to cell surface staining and presentation quantitatively, which makes it possible to directly compare the results between ESCAPE-seq and IEDB or NetMHC.

C. Data & methodology: validity of approach, quality of data, quality of presentation

The methods are clearly explained with high quality figures and well written text with the following exceptions.

1) In Supplementary Figure 1b, what do the two bars of the same color represent? The authors should claim it clearly in figure legend if they are duplicate data or other things.

2) It appears there might be a typo error in the method section related to the E-score calculation formula. Should the formula be corrected to $Score = counts_{bg} * w_{bg} + counts_{low} * w_{low} + counts_{med} * w_{med} + counts_{high} * w_{high}$?

D. Suggested improvements: experiments, data for possible revision

1) On line 146 (page 5), the authors mentioned they compared pHLA surface display with or without G2C mutation across various peptide affinities. Please clarify the exact location of the cysteine mutation (Y84C) within HLA-A0201 in the SCT? Additionally, please confirm whether this mutation was consistently used in all subsequent experiments? And it seems several other groups have investigated the impact of Y84C mutations on STC, please cite them appropriately, such as Front Immunol 2023 May 3;14:1170462 and Communications Biology, 2023 6, 528.

2) As the authors indicated in the manuscript, the cysteine mutation enhances the sensitivity of B2M signal detection. Given this enhancement, is there a risk that peptides with weak in vivo presentation might exhibit false positive signals on cell surface due to this mutation?

3) On line 175 (page 6), the authors mentioned they sorted out cells into four intensity-based bins. Details on how these bins were configured, and whether this configuration was consistent across all HLA alleles, would be insightful. If variations exist, please explain their impact on the E-score analysis for different peptides or HLA alleles.

4) On line 222 (page 8), the authors compared the AUC of the ROC between ESCAPE-seq and NetMHC, and the result showed a significant difference, about 45% vs 90%. Please include additional methods for this comparative analysis and provide a more robust validation of the hypothesis that NetMHC may not perform very well in predicting new peptides.

5) On line 289 (page 10), the authors mentioned a few neoantigen peptides could be presented by 20 or more HLA alleles. Could you please validate these findings through alternative methods, such as mass spectrometry or others, which could help distinguish between real presentations and potential false positives.

6) The focus on point mutation and fusion junction with ESCAPE-seq is valuable. However, other types of tumor

neoantigens, such as frameshift indels and non-frameshift indels, are also important. Please investigate the applicability of ESCAPE-seq for these two types of tumor-neoantigens.

E. References: appropriate credit to previous work?

Most of the references and discussion of previous work are appropriate. However, SCT related publications, especially published in the past 5 years, were not well cited.

F. Clarity and context: lucidity of abstract/summary, appropriateness of abstract, introduction and conclusions

The abstract, context and conclusions are very clearly explained at a level suitable for a general scientific audience.

Reviewer #3:

Remarks to the Author:

In this study by Shi et al., the authors developed a sequencing-based approach to test and identify peptides that bind to multiple allelic variants of HLA-A, -B, and -C, called Enhanced Single Chain Antigen Presentation sequencing (ESCAPE-seq). Building on the foundation that the surface expression of the peptide-MHC single-chain trimer (SCT) is positively correlated with the peptide-HLA binding affinity, the authors generated single-chain peptide-HLA-I (pHLA) library, lentiviral transfected HLA/TAP-KO HEK293 to allow expression of pHLA complex, and utilized surface B2m abundance as a readout for quantitative screening of positive peptides. Using the known peptides from an existing database (NetMHC), it was shown that ESCAPE-seq could identify high-affinity peptides using their scoring system based on surface B2m staining by FACS. This offered an experimental-based screening method to evaluate antigen presentation beyond the pure computational prediction. They then used this approach to target antigen presentation of viral antigens from SARS Cov-2 and cancer neoantigens from mutated oncoproteins. Important discoveries from this approach was their ability to generate and test pHLA for multiple HLA genes and alleles in the same library, which led to the discovery of many novel HLA-C presented peptides, an under-studied HLA in previous reports. However, the strategy is generally not novel, as the authors state and reference a study that used a very analogous strategy (PMID: 36593401), and this study lacks key biological validation of the identified peptide predicted by ESCAPE-seq (as was done in PMID: 36593401). This study requires validation of the key new findings, i.e. uniquely identified HLA-C peptides presented, by assessing peptide presentation from the endogenous proteins directly, or revealing that these peptides can instigate specific T cell responses.

Based on the data presented in this manuscript, I have the following principle concerns:

1. Throughout different screens in this paper, positive peptides identified by ESCAPE-seq compared well to the current state-of-the-art computational approach (NetMHC4). However, whether the predicted positive peptides are true high-affinity binders or immunogenic was not experimentally validated. It is necessary to show by a direct experimental measurement that these novel peptides in their screens are naturally processed and presented. The standard for this would be parallel Mass Spectrometry-based analysis of the presented peptidome (as performed in PMID: 36593401). This is most important for validating the new peptides presented on HLA-C, but could be done to compare the peptidome across all peptides and HLAs from the screen. This could be done in HEK293 that only express HLA-C and the full-length proteins with TAP left intact. This is critical, as though the assay works very reproducibly and has some similarities to in silico prediction algorithms, the novel peptides are most interesting, and the use of a peptide linked to B2m could enhance the generation of peptide affinities for MHC that would not occur naturally. Furthermore, the generation of these peptides by genetic means does not mean that they are naturally generated by proteasomal cleavage. Thus, there is potential for false positives, that is peptides that could bind to MHC I, but never actually do in nature. Even the use of currently available MS data to validate ESCAPE-seq would go a long way in showing the hits are real. To take it even a step further, looking for T cells reactive to such peptide/MHCs could serve to validate their platform. This could be done by using peptide-loaded HLA tetramers and screening patients with SARS Cov-2 or PBMCs from cancer patients with identified mutations. Although this is clearly a higher bar to achieve, it would definitively show that peptides identified can generate specific T cell responses against them.

2. Only 9mer peptides were included in the ESCAPE-seq library. It is unclear about the utility of this technique on a more complex system, such as broader coverage of pHLA-I at all lengths and/or translatability to identify peptides associated with HLA-II. Could authors elaborate more on how the system would capture epitopes in a more biologically relevant scenario, ranging from 8-11 amino acids? I suspect it will be important to incorporate the natural difference in peptide lengths into calculating preference and abundance on MHC. For instance, in comparing WT versus mutant protein presentation, would this have been impacted by not only comparing peptides of 9aa, but also ranging from 8aa to 11aa? How would including a pHLA library that contains multiple peptide lengths affect the E-score algorithm?

3. The novelty of the study is partly an issue as the study PMID: 36593401 uses essentially an overlapping strategy to identify MHCI peptides, though goes further in validating the accuracy of predicted peptides by MS analysis. Nevertheless, I do believe this study has potentially revealed some important new findings related to public neoantigen presentation across multiple MHC alleles, a critical role for HLA-C in antigen presentation of peptides from virus and cancer, and a refined technique for validating more HLA allelic variants in a single screen. I believe with further validation of the technique and further proof the existence of such peptides would be of importance to the field of antigen presentation identification.

Additional comments:

4. I believe the introduction should have discussed the role of MS approaches for peptide identification, and not only mention this in the discussion.

5. Further discussion of why HLA-C comes up some prominently with ESCAPE-seq and not by previous methods. Beyond levels of expression, it seems this is an important discovery, which if validated to be true, deserves further discussion as to why HLA-C is such a good peptide presenter.
6. Fig 1a-b. Does the pHLA SCT complex remain intact after being transfected in HEK293? Both b2m and HLA were measured from the complex, but there is lacking direct measurement of peptides that still stay intact with the HLA molecule.
7. Fig 2b. How was the background histogram determined? Please include isotype control staining for b2m antibody.
8. Fig 3c. Do most of the "low E-score" peptides not share the same homologous with "high E-score" peptides? Are there any consensus motifs in "low E-score" peptides?
9. Fig S3G-H. Text states "significant decrease" but there are no statistical tests in the plot.
10. Fig 5d-e. Have the identified "public antigens" been also reported to be presented on multiple alleles in other literature? Is there any existing neoantigen database that could support the findings?
11. Fig 5f. The authors stated immunogenic neoantigens are likely to reside in the Mut+ quadrant and pointed out a few previously reported neoantigens. However, there is no data in this study to show the comparison of immunogenicity in peptides from Mut+ vs. other groups.
12. Fig S1b. Do the 2 red/blue bars represent 2 replicates? Please specify in the figure or figure legend.
13. Fig S2b & S3b. Please show correlation coefficients in all correlation curves.

Version 1:

Decision Letter:

Our ref: NG-TR64666R

28th Mar 2025

Dear Howard,

Thank you for submitting your revised manuscript entitled "Massively parallel immunopeptidome by DNA sequencing reveals landscape of cancer antigen presentation" (NG-TR64666R). It has now been seen by the original referees and their comments are below. The reviewers find that the paper has improved in revision, and therefore we'll be happy in principle to publish it in Nature Genetics, pending minor revisions to satisfy the referees' final requests and to comply with our editorial and formatting guidelines.

The current version of your manuscript is in a PDF format. Please email us (genetics@us.nature.com) a copy of the file in an editable format (Microsoft Word) - we can not proceed with PDFs at this stage.

Once we receive it, we will then be performing detailed checks on your paper and will send you a checklist detailing our editorial and formatting requirements soon. Please do not upload the final materials and make any revisions until you receive this additional information from us.

Thank you again for your interest in Nature Genetics. Please do not hesitate to contact me if you have any questions.

Congratulations!

Sincerely,

Tiago

Tiago Faial, PhD
Chief Editor
Nature Genetics
<https://orcid.org/0000-0003-0864-1200>

Reviewer #1 (Remarks to the Author):

The authors have fully addressed all of my concerns.

Reviewer #2 (Remarks to the Author):

I have re-evaluated the revised manuscript entitled 'Massively parallel immunopeptidome by DNA sequencing reveals landscape of cancer antigen presentation' by Shi et al. The authors have effectively addressed all of my previous concerns, significantly improving the quality of the manuscript. The inclusion of new data has further strengthened the study, and the additional experiments and detailed descriptions now provide a clear and comprehensive presentation of the method. I believe the manuscript meets the standards of Nature Genetics and is suitable for publication.

Reviewer #3 (Remarks to the Author):

Overall, the revision has addressed the major concerns in the first review, including using complementary methods (published MS datasets and a T cell assay from healthy donor) to validate ESCAPE-seq, expanding HLA class I length coverage, and providing a more in-depth discussion on similar technologies. The provided new data has supported ESCAPE-seq's sensitivity and robustness in identifying peptides with high binding affinity to HLA-I molecules. They also refined their analysis of HLA-C peptides to find that a more stringent threshold is required to prevent false-positive predictions. Overall, my opinion is that the manuscript has gone far enough to validate their predictive results with ESCAPE-seq.

While I believe it would have been more powerful if they had experimentally validated that more NOVEL peptides (beyond single EGFR mutant peptide that has been widely published already) identified by ESCAPE-seq were antigens expressed in patient samples, this is beyond the scope of feasibility in cancer patients. This could have been done more readily in HLA-haplotyped COVID patient samples using tetramer staining to look for specific T cell responses against top novel candidates from ESCAPE-seq. This evaluation would provide insights into the false-positive rate of new antigen discovery and support ESCAPE-seq's potential in revealing novel targets, but considering the entire study, it is not essential.

Minor comments:

1. Please explain the ranking algorithm (y-axis) in Fig 3b & S3c-e and elaborate on the meaning of peptides with the rank score <1.
2. Please explain the CEFT condition in Fig 6g.
3. The terminology of "genome-wide association study" was not cohesively used, with the full term in line 72 but the abbreviation in line 454.
4. Differential thresholding of E-scores for individual HLA alleles is well-discussed in the rebuttal. I believe this is a piece of critical information that should also be included in the manuscript to provide a comprehensive view of ESCAPE-seq.

made.

Overview

We are delighted that the reviewers have enthusiastically assessed this work and appreciate all the thoughtful comments that have improved the paper. The main additions in this revision are as follows:

1. **Comprehensive validation of ESCAPE-seq by mass spectrometry:** (1) We performed large-scale validation using monoallelic HLA mass spectrometry (MS) data across 30 HLA alleles, demonstrating that one ESCAPE-seq experiment captures MS-detected peptides from 30 MS experiments with AUC-ROC>0.8 for over 90% HLA alleles. (2) We performed direct MS validation of novel HLA-C0304 presented cancer neoantigens identified from ESCAPE-seq including a “public” neoantigen EGFR T790M. The results are shown in **Figures 4 and 5, Supplementary Fig. 4 and 5.**
2. **Immunogenicity of high priority cancer neoantigens identified by ESCAPE-seq:** ESCAPE-seq nominated immunogenic neoantigens derived from recurrent cancer point mutations by systemically comparing E-score between mutant epitopes and their wild-type counterparts. We identified endogenous T cell responses to HLA-restricted epitope from cancer point mutations and fusion oncoproteins in human donors (revised **Figure 6** and **Supplementary Figure 6**). These results confirm that these prioritized neoantigens are indeed processed, presented, and capable of eliciting immune response in vivo.
3. **Clarified novelty and advantages of ESCAPE-seq over prior studies:** Compared to prior single chain pMHC methods, ESCAPE-seq enables >500-fold more comprehensive screening scalability across diverse HLA alleles and peptide combinations, ultimately reaching population-wide coverage. Compared to the method of Bruno et al, ESCAPE-seq offers 100-fold greater signal to background, can uniquely profile HLA-C alleles, for the first time enables combinatorial screening of thousands of peptides across 50 HLA alleles in a single experiment, and comprehensive profiling of antigen types including viral epitopes, cancer point mutations, fusion oncoproteins, indels and deletions, representing the broadest HLA and antigen coverage to date in a single platform.

We address each of the reviewer comments below.

Page 1-11 for response to Reviewer 1

Page 12-21 for response to Reviewer 2

Page 21 onward for response to Reviewer 3

Reviewer #1:

I read this paper with great interest, and found the study a really impressive one. I think the experimental strategy presented by the author is compelling, and seems to at least recapitulate MHC binding predictions from netMHC. In general I enjoyed the paper, and am supportive of its eventual publication. My major concerns included the lack of confirmation of key antigens. I also am curious to know how well their strategy works on class II HLA, which is now not included at all in their study.

(1) One key point to me is that overall it does appear that netMHC and ESCAPE-seq perform well. It appears that in COVID antigens there is some discrepancy (**Supplementary Figure 3g, 3h**). Similarly for neoantigens (**Fig 4c**). I think it is important to understand whether netMHC is

underperforming ESCAPE-seq, or vice versa. I think it is also important to understand the situations where the predictions are discrepant. Is it indeed the case that ESCAPE-seq predicts binding better somehow? As the authors point out IEDB is used to train netMHC, so perhaps it is more compelling that ESCAPE-seq obtains comparable performance. It isn't clear in the data which is IEDB independent, that ESCAPE-seq is getting the right answers more often. I think it is important for the authors to experimentally validate some of their predictions to clearly show the benefit of their strategy in scoring new antigens. I think this would demonstrate the value of their experimental strategy. To me this is critical – since without this sort of evidence, netMHC is the easier choice.

Thank you for your enthusiastic assessment and constructive suggestions which have helped us improve the paper.

We have conducted comprehensive analyses using three independent approaches to address how ESCAPE-seq performs relative to NetMHC predictions:

1. First, we carefully delineated which datasets are NetMHC training-dependent versus training-independent. In **Figure 2**, we show that ESCAPE-seq and NetMHC perform similarly well using the IEDB peptide binding affinity dataset. This is expected since IEDB data were used to train NetMHC.

Figure legend, Figure 2, g, Barplots of AUC-ROC for all 4 HLA-I alleles to compare the performance of ESCAPE with NetMHC. **h,** Barplots of AUC-PRC for all 4 HLA-I alleles to compare the performance of ESCAPE with NetMHC.

2. To rigorously assess performance on novel antigens, we compiled a set of experimentally validated tumor neoantigens from the Cancer Epitopes Database (CEDAR) and curated literature as an IEDB-independent validation set. Our analysis in Figure 4g demonstrates that ESCAPE-seq successfully identified 100% (21/21, E-score>3.8) of these validated tumor neoantigens. In contrast, NetMHC identified only 19% (4/21) using binding affinity-based prediction (BA mode, affinity <500nM) or 57% (12/21) using elution-based prediction (EL mode, rank <2%). This suggests ESCAPE-seq has superior sensitivity for identifying true positive cancer neoantigens that have been experimentally validated. This result is shown in revised **Figure 4g**.

Figure legend, Figure 4g, Plot of E-score versus NetMHC prediction for the known antigen peptides in the combinatorial pool. NetMHC4 results in both binding affinity mode (Ba, upper) or elution model (El, lower) were shown. The greyed box highlights peptides for which NetMHC failed to make a prediction.

- For additional validation using orthogonal data, we leveraged a comprehensive monoallelic HLA-I mass spectrometry data (Sarkizova et al., *Nature Biotechnology*, 2019) containing top **986** MS-captured peptides for **30** HLA alleles as another IEDB-independent validation set. Basically, We performed combinatorial ESCAPE-seq screening of these peptides across all **30** HLA alleles, examining over 29,000 pHLA interactions. The quality control analysis showed high consistency in both sequencing reads and calculated E-scores between replicates. The results are shown in **Figure 4b-d** and **Figure S4a-e**.

Figure legend: a (Figure 4b in the manuscript), Schematic view of combinatorial ESCAPE-seq using HLA monoallelic MS data. **b (Figure S4a),** Representative correlation plot of read count between replicates for ESCAPE-seq on pool of peptides from MS dataset. Pearson correlation coefficients were shown in the plot. **c (Figure S4b),** scatter plot of calculated E-score between replicates from (b) above.

We further evaluated ESCAPE-seq's performance using receiver operating characteristic (ROC) curves, comparing true positive rates against false positive rates relative to MS data. The analysis revealed strong performance with AUC-ROC values between 0.8-1.0 for most HLA alleles. We further compared ROC between ESCAPE-seq and NetMHC prediction. This analysis revealed comparable performance between ESCAPE-seq and NetMHC across many alleles. Notably, ESCAPE-seq demonstrated superior performance (higher ROC-AUC metrics) for several alleles including HLA-C04:01, HLA-B13:01, and

HLA-A*34:01 (marked with arrows in figure below), suggesting particular strength in identifying peptides presented by these less well-characterized alleles. The results are shown in **Figure 4d** and **Figure S4e**.

Figure legend: left panel (**Figure 4d** in the manuscript), AUC-ROC metrics comparing ESCAPE-seq and MS results across 30 HLA alleles. Peptides were assigned a value of 1 if detected in MS and 0 if not when calculating area under curve (AUC) of ROC curve. Right panel (**Figure S4e**), Representative ROC curves across all 30 alleles and comparison of AUC-ROC between ESCAPE-seq with NetMHC predictions in both EL (elution) and BA (binding affinity) modes. Arrows indicate specific HLA alleles where ESCAPE-seq achieved higher AUC-ROC compared to NetMHC predictions.

Together, these analyses demonstrate that while ESCAPE-seq matches NetMHC's performance on well-characterized training data, ESCAPE-seq shows better performance on novel antigens and less well-characterized HLA alleles.

(2) I notice that class II HLA is in fact not addressed at all. Does their strategy work for class II HLA? It is possible that their method is class I specific, in which case this should be clearly stated.

We thank the reviewer for this insightful comment about HLA class II applications. Our current study focuses on HLA class I, and we have modified the abstract accordingly to state: "We present ESCAPE-seq...a massively parallel platform for comprehensive screening of peptide-HLA class I combinations."

ESCAPE-seq leverages the cell's endogenous quality control system that ensures only stable pHLA complexes reach the cell surface. While HLA class I presentation occurs in virtually all nucleated cells including our HEK293T platform, HLA class II presentation requires specialized cellular machinery found only in professional antigen-presenting cells (APCs). The current

implementation using HEK293T cells therefore lacks the necessary quality control and presentation machinery for HLA class II.

However, we envision that the core principle of ESCAPE-seq - using cellular quality control to select stable pMHC complexes - could be adapted for HLA class II applications through two potential approaches:

1. Engineering HEK293T cells to express the essential HLA class II quality control machinery and chaperone proteins
2. Adapting the platform to work directly in professional APCs that naturally possess the complete HLA class II presentation pathway

While implementing these adaptations would require substantial optimization and is beyond the scope of the current manuscript, we have added discussion of these future directions to highlight the broader potential of the ESCAPE-seq approach.

(3) For Figure 2d, the authors present a heatmap clustering HLA alleles, can the authors please label the individual HLA alleles. I see there is a large HLA-B block, A block, and C block. It would be interesting to see if the individual allele names to see if alleles with similar 2 digit designations group together. According to the UMAP some of the alleles cluster with alleles from a different gene. For example A4301 appears to cluster with HLAB genes. Is that similar to the heat map?

The hierarchical clustering heatmap in revised **Figure 5a** has been updated to include individual HLA allele labels, allowing direct visualization of clustering patterns.

There is indeed difference between hierarchical clustering and UMAP (e.g. HLA-A3401 cluster with HLA-B alleles in UMAP but not in hierarchical clustering). Such difference might be due to fundamental differences in the distance calculation.

1. Hierarchical clustering uses pairwise distances to build a tree structure, forcing a strict hierarchy of relationships. It typically employs Euclidean or correlation-based distance metrics and cannot capture non-linear relationships between alleles.
2. UMAP preserves both local and global structure while allowing for non-linear relationships. It projects high-dimensional data into 2D space while attempting to maintain relative distances between points.

These complementary visualization methods provide different perspectives on HLA allele relationships: hierarchical clustering highlights broad family relationships and direct similarities, while UMAP better captures subtle, non-linear relationships in peptide binding preferences that might cross traditional HLA class boundaries.

(4) The paper's title is: "Massively parallel immunopeptidome by DNA sequencing reveals landscape of cancer antigen presentation". I think this is somewhat misleading. I think the paper presents a powerful method to do high throughput characterization of class I antigen binding presentation. I am unsure about the value that this paper adds to the cancer field. They apply their method to cancer antigens, but none of those results are validated or taken further in any meaningful way. Authors should alter the title and the abstract to reflect the methodological aspect, which is the main contribution here.

We thank the reviewer for this thoughtful feedback about the manuscript's focus and title. We fully agree that ESCAPE-seq represents a powerful methodological advance, and the bulk of ESCAPE-seq data in this work is on cancer neoantigens. Therefore, we thought the title should reflect this fact to help readers identify this resource. Moreover, our revised manuscript demonstrates substantial contributions to cancer immunology through several key findings and new validation experiments:

1. We have added orthogonal validation experiments for cancer driver mutation-derived neoantigens and fusion oncoprotein breakpoint-derived neoantigens, including HLA-C0304 MS of high E-score tumor neoantigen and immunogenicity validation using peptide stimulated T cell reactivity assay. These experiments provide a resource of high-priority cancer neoantigens with immediate clinical relevance.
 - (a) Specifically, we engineered new monoallelic HLA-C*03:04 cell line and expressed the ER targeting epitopes discovered by ESCAPE-seq (we followed the same procedure as PMID: 36593401). We performed HLA-eluted peptide mass spectrometry analysis of cells expressing high E-score antigens, and detected neoantigens as shown below. The result is shown in revised **Figure S5g**.

Figure legend: Figure S5g, Intensity traces show the detection of peptides (YIMSDSNYV, LTSTVQLIM and SSYGQQSSL) with mass spectrometry.

Among them, SSYGQQSSL is breakpoint derived neoantigen from EWSR1 and FLI1 fusion gene, which is a chromosomal translocation that occurs in over 85% of Ewing sarcoma cases (PMID: 19417137). YIMSDSNYV is an epitope from Flt3 D835Y point mutation, which is the most frequently observed kinase domain mutation in AML patients, constituting ~50% of all FLT3 AL missense mutations (PMID: 28077790), while LTSTVQLIM is derived from oncogene EGFR T790M that is present in about half of lung cancer patients who have acquired resistance to EGFR-TKI treatment. It's also found in about 60% of patients with NSCLC (PMID: 19096299). Considering >7% world population has HLA-C0304 allele, many patients may benefit from targeting such neoantigens.

- (b) We characterized the immunogenicity of ESCAPE-seq identified neoantigens using HLA-typed healthy donor samples as an additional validation approach. Given that fusion oncoproteins often generate novel junctional sequences, we hypothesized these breakpoints could serve as immunogenic epitopes. We selected 13 fusion breakpoint-derived epitopes identified by ESCAPE-seq predicted to bind either HLA-A*03:01 or A*24:02 and tested them using PBMCs from an HLA-A*03:01+A*24:02+ healthy donor. Using a well-established protocol, we differentiated APCs from donor PBMCs, stimulated them with peptide pools in the presence of cytokines, and assessed T cell responses after 7 days of expansion. Upon restimulation with

individual peptides, we measured intracellular IFN γ and TNF α production by flow cytometry. This analysis revealed that the NPM1-ALK fusion-derived epitope was immunogenic, evidenced by a significant increase in IFN γ +TNF α + polyfunctional T cells following peptide stimulation. This result is shown in revised **Figure 6d,h-i**.

Figure legend: a (Figure 6d in the manuscript), Schematic view of peptide stimulation assay to detect antigens-specific T cell reactivity. **b (Figure 6h)**, Flow cytometry plot of IFN γ and TNF α producing CD8+ T cells. **c (Figure 6i)**, Bar plot showing the percentage of IFN γ +TNF α + polyfunctional CD8+ T cells upon individual peptide or pooled peptides stimulation.

- Our systematic comparison of mutated neoantigens with their wild-type counterparts revealed two categories of HLA presented neoantigens: (1) mutations that enable HLA binding while their wild-type sequences do not bind (Mut+ epitopes labeled in shaded area, (2) both mutant neoantigen and wild-type counterparts can present on HLA (Mut+Wt+). We found that previously reported immunogenic neoantigens are all Mut+ epitopes, likely due to the absence of central tolerance of Mut+ antigens as their wildtype counterparts do not present for selection. This finding provides a valuable prioritization strategy for neoantigen selection in therapeutic development. This result is shown in revised **Figure 6a-c**.

Figure legend: a (Figure 6a in the manuscript), Schematic view of Mut+ only peptides by comparing E-score of mutant vs wild-type counterpart peptides. **b** (Figure 6c), Scatter plots of E-score of peptides with mutation (Mut, x-axis) vs. the E-score of its corresponding wild-type peptides (WT, y-axis) from EGFR_T790M mutation (left), KRAS_G12D (middle), and FLT3 D835Y (right panel). The red dots highlighted the immunogenic neoantigens reported in the literature.

To validate this principle, we conducted new immunogenicity studies focusing on EGFR T790M neoantigen epitopes presented by HLA-A23:01 (an HLA allele enriched in African American populations). We determined the immunogenicity of one Mut+ peptide (p17: LTSTVQLIM) and two Mut+WT+ peptides (p22: QLIMQLMPF and p23: LIMQLMPFG) using peptide stimulation of HLA-A*23:01+ healthy donor T cells, only this Mut+ epitope induced robust T cell responses, as measured by polyfunctional cytokine production (IFN γ + TNF α +). This result is shown in revised Figure 6d-g

Figure legend: a (Figure 6d in the manuscript), Schematic view of peptide stimulation assay to detect antigens-specific T cell reactivity. **b** (Figure 6e), Bar plot showing E-score of EGFR T790M epitopes between mutant vs their wild-type counterpart. **c** (Figure 6f), Flow cytometry plot of IFN γ and TNF α producing CD8+ T cells. **d** (Figure 6g), Bar plot

showing the percentage of IFN γ +TNF α + polyfunctional CD8+ T cells upon individual peptide stimulation. CEFT peptide pools represent well-established immunogenic common viral antigens.

Together, these findings demonstrate that ESCAPE-seq not only provides a powerful screening platform but also yields actionable insights for cancer immunotherapy by:

- Identifying population-spanning neoantigens
- Establishing principles for prioritizing immunogenic neoantigens
- Experimentally validating these principles through T cell response studies

We believe these contributions substantiate the cancer focus reflected in our title.

(5) While the methods section is extensive, the authors should make their detailed protocols public on protocols.io or some such place so that their methods can be reproducibly implemented by others.

We thank the reviewer for this helpful suggestion to enhance the reproducibility of our method. We agree that detailed protocols are essential for broad adoption of ESCAPE-seq by the research community. We will publish comprehensive step-by-step protocols on protocols.io concurrent with the paper's publication.

Minor Comments:

(1) For the some of the reproducibility plots (e.g. Supp Figure 3) it would be useful to add test-retest correlation values to quantify the reproducibility.

Correlation coefficients have been added as suggested. For example (**Supplementary Figure 3a**):

(2) For figures where authors use barplots (Supp Fig 3g, Supp Figure 3h, Figure 2g/h) individual data points should actually be shown, so as not to obscure the underlying data.

Figures have been updated as suggested. For example, **Supplementary Figure 3g/h**:

(3) Figure 5f. Do the authors have a clear strategy to identify those antigens with discrepant E-scores for wild type and mutant? They should color those antigens differently on the plot to make it clear.

Thanks for your suggestion. We now included it in the Methods on calculating and comparing the E-score of cancer epitopes with E-score of the corresponding wild-type peptides. We also highlighted the region for neoantigens that can be presented (shaded in light green, revised Figure 6a-c), but its corresponding wild-type peptide cannot be presented (with low E-score).

Figure legend: a (Figure 6a in the manuscript), Schematic view of Mut+ only peptides by comparing E-score of mutant vs wild-type counterpart peptides. **b** (Figure 6b), Scatter plots of E-score of peptides with mutation (Mut, x-axis) vs. the E-score of its corresponding wild-type peptides (WT, y-axis) from EGFR_T790M mutation (left), KRAS_G12D (middle), and FLT3

D835Y (right panel). The red dots highlighted the immunogenic neoantigens reported in the literature.

Reviewer #2:

Remarks to the Author:

A. Summary of the key results:

The authors have developed a DNA sequencing-based method, called ESCAPE-seq, for peptide-HLA combinations screening with pMHC single-chain trimers (pMHC SCT), which consisted of amino acid antigen peptide, B2M and MHC alleles. They investigated thousands of peptides from SARS-CoV-2 variants, tumor hotspot mutations, and fusion variants across 50 diverse HLA class I alleles with ESCAPE-seq, which greatly underscores its significant applicability and potential in cancer immunotherapy.

Thank you for your positive assessment of the method and its strong applicability.

B. Originality and significance: if not novel, please include reference

The overall design of ESCAPE-seq approach is elegant. However, William Chour et al. have already adapted single-chain trimer technologies for pMHC library generation across multiple Class I HLA alleles (Communications Biology, 2023 6, 528). They also constructed SCT libraries for SARS-CoV-2 variants, tumor-associated antigens and common viral strains (CMV, EBV, influenza, and rotavirus), although the scale and throughput are not as high as ESCAPE-seq. So, in that sense the paper is not hugely novel. The potential novelty is that the authors develop an algorithm to assign ESCAPE-scores to each peptide to evaluate the peptide binding affinity to cell surface staining and presentation quantitatively, which makes it possible to directly compare the results between ESCAPE-seq and IEDB or NetMHC.

Thank you for the opportunity to clarify the novel contribution of ESCAPE-seq over the work of Chour et al, which we now cite. We fully agree with the advantages of scale and quantitative precision for ESCAPE-seq, which the reviewer astutely indicated. In addition, while both approaches utilize single-chain trimers (SCT), ESCAPE-seq represents a fundamentally different technology with distinct purposes and capabilities from Chour et al. The key differences are:

1. **Distinct Scientific Goals:** Chour et al. develop SCTs primarily as a foundation to generate soluble purified pMHC tetramers in vitro for detecting antigen-specific T cells. They use computational prediction to prioritize peptide sequence to design their pHLA SCT. Our ESCAPE-seq leverages cellular quality control machinery in living cells to directly identify stable pHLA complexes, enabling de novo antigen discovery without relying on prior computational predictions.
2. **Technical Implementation:** Our approach utilizes membrane-bound SCTs (containing natural HLA transmembrane domain) and cellular quality control machinery as a natural selection system. Chour et al. use cell-free secreted SCTs (deleted the transmembrane domain and added His-tag) requiring individual protein purification and multimerization steps.
3. **Throughput and Scalability:** ESCAPE-seq enables simultaneous screening of 75,000 pHLA combinations (1,500 peptides × 50 HLA alleles) through pooled library approaches. Chour et al. screened 162 pHLA combinations (18 peptides × 9 HLA templates) due to

limitations of individual processing steps. ESCAPE-seq represents a nearly 500-fold improvement in scalability.

Here is a detailed comparison table:

Feature	ESCAPE-seq (Our Method)	Chour et al. Method
Primary Purpose	De novo antigen discovery	Antigen-specific T cell detection
Screening Approach	Cell surface pHLA presentation	pHLA Tetramer generation
SCT Design	Membrane-bound	Secreted with His-tag
Library Generation	Pooled one-step cloning	Individual cloning
Cell Line Engineering	HLA-I KO cell line with pooled transduction of pHLAs	Individual transfection of each pHLA
Processing Steps	Direct cell sorting	Individual pHLA purification and multimerization
Throughput	75,000 pHLA combinations	162 pHLA combinations
Input Requirements	Peptide sequences only	Computationally predicted HLA binding peptides
Output	Quantitative binding scores (E-score)	Binary tetramer formation
Time/Cost Efficiency	High (pooled approach)	Lower (individual processing)
Application	Population-wide antigen discovery	Antigen-specific T cell detection

These fundamental differences in goals, design, and implementation make ESCAPE-seq a novel and complementary approach to existing methods in the field of antigen discovery and presentation.

C. Data & methodology: validity of approach, quality of data, quality of presentation
The methods are clearly explained with high quality figures and well written text with the following exceptions.

1) In Supplementary Figure 1b, what do the two bars of the same color represent? The authors should claim it clearly in figure legend if they are duplicate data or other things.

Thank you for bringing this issue to our attention. We apologize for any confusion in Supplementary Figure 1b. We have clarified in the revised **Figure S1b** and Figure legend: “Red bars: Two different high-affinity peptides (HP1 and HP2); Blue bars: Two different no-affinity peptides (NP1 and NP2).”

We have updated the figure labels and legend to clearly indicate these distinctions and have included the specific peptide sequences in **Supplementary Table 1** for reference.

Figure description: Figure S1b, Bar plots for (Fig. 1b) quantification, comparing gMFI of b2m staining of HEK293T cells expressing single chain trimer (SCT) of HLA-A*0201 with high affinity (HP) peptides vs. fusions with peptides that have no binding affinity (NP). Red bars: Two different high-affinity peptides (HP1 and HP2); Blue bars: Two different no-affinity peptides (NP1 and NP2).

2) It appears there might be a typo error in the method section related to the E-score calculation formula. Should the formula be corrected to $E_{score} = counts_{bg} * w_{bg} + counts_{low} * w_{low} + counts_{med} * w_{med} + counts_{high} * w_{high}$?

Thank you for pointing it out. We apologize for the typo in the formula. We have corrected it accordingly in the text (page 28 line 830).

D. Suggested improvements: experiments, data for possible revision
1) On line 146 (page 5), the authors mentioned they compared pHLA surface display with or without G2C mutation across various peptide affinities. Please clarify the exact location of the cysteine mutation (Y84C) within HLA-A0201 in the SCT? Additionally, please confirm whether this mutation was consistently used in all subsequent experiments? And it seems several other groups have investigated the impact of Y84C mutations on STC, please cite them appropriately, such as Front Immunol 2023 May 3:14:1170462 and Communications Biology, 2023 6, 528.

Thank you for this important point about the cysteine mutation. We appreciate the opportunity to clarify in the revised manuscript.

We confirm that the HLA-I mutation is Y84C (tyrosine to cysteine at position 84). This position and surrounding amino acids are highly conserved across HLA class I alleles (see motif analysis in revised Figure S1f), making this a consistent modification point across different HLA variants.

Figure description: Figure S1f, Motif analysis of amino acids across all HLA-A, B, and C alleles around the region with Y84. The arrow marks the location of conserved Y84 that is mutated to Y84C in SCT.

The Y84C mutation was consistently used in all subsequent experiments across all HLA alleles tested. The mutation creates a disulfide trap that helps stabilize the pHLA complex, as previously demonstrated.

We have added citations to acknowledge prior work investigating the impact of Y84C mutations on single-chain trimers. We have updated the methods section to explicitly state the position and nature of this mutation, and its consistent use throughout our studies.

2) As the authors indicated in the manuscript, the cysteine mutation enhances the sensitivity of B2M signal detection. Given this enhancement, is there a risk that peptides with weak in vivo presentation might exhibit false positive signals on cell surface due to this mutation?

Thank you for raising this important question about potential false positives due to the cysteine mutation. We have conducted several analyses to address this concern.

First, using HLA-A*02:01 as a model, we directly compared ESCAPE-seq performance with and without the Y84C mutation using IEDB peptides with known binding affinities. The cysteine mutant showed better correlation with IEDB affinity measurements (**Figure 2c**) compared to wild-type HLA allele (**Figure S2c**). This improved correlation suggests the cysteine mutation better recapitulates natural peptide-HLA binding properties.

Fig. S2c

Fig. 2c

Figure legend: **Fig. S2c** (Figure S2c in the manuscript), Scatter plot of IEDB affinities vs. ESCAPE-seq E-score for wild-type HLA-A*0201 allele. **Fig.2c** (Figure 2c), the same plot as S2c but for HLA-A*0201 harboring Y84C mutation. A line at 500nM of affinity and a cutoff line for E-score were drawn as indicated.

To systematically assess the false positive rate, we comprehensively evaluated potential false positives using >2,000 peptides with known affinities across four common HLA alleles. Both AUROC and AUC-PRC metrics showed high performance (≥ 0.9) for ESCAPE-seq (Figure 2g-h, red bars). While some peptides with lower measured affinities showed higher E-scores (Figure 2c, lower right quadrant), these cases were relatively rare and typically near the classification boundary. These borderline cases can be addressed through careful selection of E-score thresholds.

Figure legend: **Figure 2g.** Barplots of AUC-ROC for all 4 HLA-I alleles to compare the performance of ESCAPE with NetMHC. **h** (Figure 2h), Barplots of AUC-PRC for all 4 HLA-I alleles to compare the performance of ESCAPE with NetMHC.

We acknowledge the possibility of false positives due to enhanced stability from the cysteine mutation. However, our comprehensive benchmarking indicates that such effect is minimal and does not significantly impact the overall reliability of the method. The improved signal-to-noise ratio provided by the cysteine mutation ultimately enables more accurate identification of true binding peptides. These analyses support our decision to use the cysteine mutation design in all downstream experiments, as it provides optimal performance while maintaining low false positive rates.

3) On line 175 (page 6), the authors mentioned they sorted out cells into four intensity-based bins. Details on how these bins were configured, and whether this configuration was consistent across all HLA alleles, would be insightful. If variations exist, please explain their impact on the E-score analysis for different peptides or HLA alleles.

Thank you for the opportunity to clarify about our sorting strategy. We have now included comprehensive details about the binning procedure in the Methods section.

The four intensity-based bins are configured using evenly distributed intervals on a logarithmic scale based on PE-B2m intensity. Specifically for example, we set the center of the negative (background) peak at 100 fluorescence units, with subsequent bins centered at approximately 400 (low), 1600 (medium), and 6400 (high) units, with the high bin also including all cells with greater intensity.

We recognize that inherent experimental variability and instrument settings can cause slight variations in absolute intensity values between batches. To account for this technical variation, we implemented a two-step normalization process: (1) Normalization of read counts within each bin. (2) Normalization per peptide to generate the final E-score

This normalization approach ensures that small variations in bin boundaries between experiments do not substantially affect the final E-scores. Most importantly, the relative ranking of peptides based on E-scores remains consistent even with slight variations in bin boundaries. This is evident by the high correlation between biological replicates, where each sort of replicate would vary a slightly on gating when sorted on a different time and sorter. However, E-scores from replicates are typically highly correlated (Figure S2a-b, $r > 0.98$).

Figure legend: Figure S2. a, Representative correlation plot of read count between replicates for ESCAPE screen. HLA-A*0201 was used here as an example. 4 fractions (BG/background, low, medium and high) were highlighted in different colors. Pearson correlation coefficients were shown. b, Correlation of E-score between replicates of ESCAPE-seq using HLA alleles A*0201. Pearson correlation coefficients of each were shown in the plots.

4) On line 222 (page 8), the authors compared the AUC of the ROC between ESCAPE-seq and NetMHC, and the result showed a significant difference, about 45% vs 90%. Please include

additional methods for this comparative analysis and provide a more robust validation of the hypothesis that NetMHC may not perform very well in predicting new peptides.

T-statistics test is now included in the revised plot that documents the significance of the difference. Please note that we interpret the erroneous predictions of NetMHC as a result of insufficient training data for NetMHC for certain HLA alleles, rather than any defect in the algorithm. ESCAPE-seq is potentially complimentary to NetMHC by generating the missing immunopeptidome data.

5) On line 289 (page 10), the authors mentioned a few neoantigen peptides could be presented by 20 or more HLA alleles. Could you please validate these findings through alternative methods, such as mass spectrometry or others, which could help distinguish between real presentations and potential false positives.

We appreciate the reviewer's important question about validating broadly presented neoantigens. We have strengthened our analysis through multiple complementary approaches.

First, analysis of monoallelic HLA mass spectrometry data (PMID: 31844290) demonstrates that some peptides can naturally be presented across diverse HLA alleles (Figure below). This biological phenomenon has been previously documented. For example, the well-characterized KRAS G12V neoantigen (VVGAVGVGK) has been shown to be presented by multiple HLA-A alleles including A0301, A1101, A3001, A6801 and KRAS G12D (GADGVGKSAL) is presented by B0702, C0304, C0802, as documented through both binding assays and HLA mass spectrometry (PMID: 35474673).

Figure legend: Distribution of peptides detected in MS from multiple monoallelic MS experiments (sarkizova, et al Nat. Biotech 2020, PMID: 31844290)

To distinguish true broadly-presented antigens from potential false positives, we performed a systematic analysis of HLA-A*02 binding peptide motifs across different E-score ranges. We found that peptides with high E-scores ($E > 6$) showed strong matches to experimentally-determined HLA binding motifs from mass spectrometry data (See Reviewer Figure below). In contrast, peptides with medium ($3.8 < E \leq 6$) or low ($2.9 < E \leq 3.8$) E-scores demonstrated progressively weaker matches to these validated motifs, suggesting increased false positive rates at lower thresholds.

Figure legend (Figure SN-2 in the manuscript): E-score distribution for HLA-A*02:01 (**a**) and peptide motif patten (**b**) based on different E-score bins as illustrated in (**a**).

Using this more stringent high E-score cutoff ($E > 6$), we reanalyzed broadly-presented antigens. While this reduced the total number of broadly-presented candidates, we still identified select peptides presented by up to 20 or more HLA alleles. One notable example is the EGFR T790M LTSTVQLIM epitope, which was supported by several publications in the literature and we have further validated through multiple orthogonal approaches:

1. Recent HLA monoallelic mass spectrometry has directly confirmed presentation by HLA-C*07:01 using cells expressing the T790M minigene (PMID: 37857725).
2. In a neoantigen clinical trial, this epitope was shown to be presented by HLA-C*15:02, with specific T cell responses detected using pMHC tetramers (PMID: 34244308).
3. We engineered new monoallelic HLA-C*03:04 cell line and overexpress the epitope targeting ER (we followed the same procedure as Bruno et al. (PMID: 36593401). We performed HLA-eluted peptide mass spectrometry analysis of cells expressing high E-score antigens, including EGFR T790M LTSTVQLIM, confirmed direct presentation by this additional allele. The result is shown in revised **Figure S5g**.

4. Using peptide stimulation of HLA-A23:01+ healthy donor PBMC sample, we characterized EGFR T790M-specific T cell responses by quantifying the frequency of IFN γ + TNF α + polyfunctional T cells in response to EGFR T790M individual epitope peptide and peptide pools. We observed a significant T cell response upon stimulation of T790M LTSTVQLIM (mutation at the position 9), indicating both the presentation by HLA-A23:01 and its immunogenicity to induce T cell reactivity. The result is shown in revised **Figure 6d-g**.

Figure legend: **a** (Figure 6d in the manuscript), Schematic view of peptide stimulation assay to detect antigens-specific T cell reactivity. **b** (Figure 6e), Bar plot showing E-score of EGFR T790M epitopes between mutant vs their wild-type counterpart. **c** (Figure 6f), Flow cytometry plot of IFNγ and TNFα producing CD8+ T cells. **d** (Figure 6g), Bar plot showing the percentage of IFNγ+TNFα+ polyfunctional CD8+ T cells upon individual peptide stimulation. CEFT peptide pools represent well-established immunogenic common viral antigens.

Together, these orthogonal validations through mass spectrometry and T cell reactivity assays, combined with previous literature evidence, confirm that the EGFR T790M epitope can indeed be presented by multiple HLA alleles including C0701, C1502, C0304, and A2301. This demonstrates that while broadly-presented neoantigens may be less common than initially predicted using lower thresholds, they do exist and can be validated through multiple experimental approaches.

6) The focus on point mutation and fusion junction with ESCAPE-seq is valuable. However, other types of tumor neoantigens, such as frameshift indels and non-frameshift indels, are also important. Please investigate the applicability of ESCAPE-seq for these two types of tumor-neoantigens.

Thank you for this astute suggestion that highlights the versatility of ESCAPE-seq and the value of our data sets. We would like to clarify that ESCAPE-seq is fundamentally sequence-agnostic and can be applied to any peptide sequence regardless of the underlying mutation type. The method requires only the input peptide sequence to be screened, making it equally applicable to point mutations, fusion junction peptides, frameshift indels, non-frameshift indels, deletions, etc.

To demonstrate this versatility, we have expanded our ESCAPE-seq in the revised manuscript to include several additional types of neoantigens: Fusion oncoproteins with indel sequences: EML4-ALK fusion containing the indel sequence AKMSTREKNS, RUNX1-RUNX1T1 fusion with the indel sequence RNR, TCF3-PBX1 fusion incorporating the indel sequence YSV. Deletion variants: EGFR exon 19 deletion (E746-750), which removes five amino acids and is a clinically relevant mutation in non-small cell lung cancer (See **Supplementary Table 4**)

These data showcase that ESCAPE-seq can effectively screen any user-specified antigen sequence, regardless of the underlying genomic alteration that generated it. This flexibility

makes ESCAPE-seq a comprehensive platform for investigating the complete landscape of tumor neoantigens.

E. References: appropriate credit to previous work?

Most of the references and discussion of previous work are appropriate. However, SCT related publications, especially published in the past 5 years, were not well cited.

We thank the reviewer for highlighting the importance of single chain pMHC trimer (SCT) related publications. In the revised manuscript, we have expanded our discussion and citations of SCT literature, particularly from the past five years. We have added key references such as Chour et al (PMID: 37193826) and (PMID: 37207206).

F. Clarity and context: lucidity of abstract/summary, appropriateness of abstract, introduction and conclusions

The abstract, context and conclusions are very clearly explained at a level suitable for a general scientific audience.

Thank you!

Reviewer #3:

Remarks to the Author:

In this study by Shi et al., the authors developed a sequencing-based approach to test and identify peptides that bind to multiple allelic variants of HLA-A, -B, and -C, called Enhanced Single Chain Antigen Presentation sequencing (ESCAPE-seq). Building on the foundation that the surface expression of the peptide-MHC single-chain trimer (SCT) is positively correlated with the peptide-HLA binding affinity, the authors generated single-chain peptide-HLA-I (pHLA) library, lentiviral transfected HLA/TAP-KO HEK293 to allow expression of pHLA complex, and utilized surface B2m abundance as a readout for quantitative screening of positive peptides. Using the known peptides from an existing database (NetMHC), it was shown that ESCAPE-seq could identify high-affinity peptides using their scoring system based on surface B2m staining by FACS. This offered an experimental-based screening method to evaluate antigen presentation beyond the pure computational prediction. They then used this approach to target antigen presentation of viral antigens from SARS Cov-2 and cancer neoantigens from mutated oncoproteins. Important discoveries from this approach was their ability to generate and test pHLA for multiple HLA genes and alleles in the same library, which led to the discovery of many novel HLA-C presented peptides, an under-studied HLA in previous reports. However, the strategy is generally not novel, as the authors state and reference a study that used a very analogous strategy (PMID: 36593401), and this study lacks key biological validation of the identified peptide predicted by ESCAPE-seq (as was done in PMID: 36593401). This study requires validation of the key new findings, i.e. uniquely identified HLA-C peptides presented, by assessing peptide presentation from the endogenous proteins directly, or revealing that these peptides can instigate specific T cell responses.

Based on the data presented in this manuscript, I have the following principle concerns:

1. Throughout different screens in this paper, positive peptides identified by ESCAPE-seq compared well to the current state-of-the-art computational approach (NetMHC4). However, whether the predicted positive peptides are true high-affinity binders or immunogenic was not

experimentally validated. It is necessary to show by a direct experimental measurement that these novel peptides in their screens are naturally processed and presented. The standard for this would be parallel Mass Spectrometry-based analysis of the presented peptidome (as performed in PMID: 36593401). This is most important for validating the new peptides presented on HLA-C, but could be done to compare the peptidome across all peptides and HLAs from the screen. This could be done in HEK293 that only express HLA-C and the full-length proteins with TAP left intact. This is critical, as though the assay works very reproducibly and has some similarities to in silico prediction algorithms, the novel peptides are most interesting, and the use of a peptide linked to B2m could enhance the generation of peptide affinities for MHC that would not occur naturally. Furthermore, the generation of these peptides by genetic means does not mean that they are naturally generated by proteasomal cleavage. Thus, there is potential for false positives, that is peptides that could bind to MHC I, but never actually do in nature. Even the use of currently available MS data to validate ESCAPE-seq would go a long way in showing the hits are real. To take it even a step further, looking for T cells reactive to such peptide/MHCs could serve to validate their platform. This could be done by using peptide-loaded HLA tetramers and screening patients with SARS Cov-2 or PBMCs from cancer patients with identified mutations. Although this is clearly a higher bar to achieve, it would definitively show that peptides identified can generate specific T cell responses against them.

We appreciate the reviewer's critical feedback regarding experimental validation of ESCAPE-seq predictions. We have extensively addressed this concern through three complementary experimental approaches.

First, we conducted large-scale validation using existing monoallelic HLA mass spectrometry (MS) data. We leveraged a comprehensive monoallelic HLA MS dataset (Sarkizova et al NBT 2020) containing top **986** MS-captured peptides for **30** HLA alleles. We performed combinatorial ESCAPE-seq screening of these peptides across all 30 HLA alleles, examining over **29,000** pHLA interactions (revised **Figure 6b**). The quality control analysis showed high consistency in both sequencing reads and calculated E-scores between replicates (**Figure. S6b**).

To assess ESCAPE-seq's ability to identify MS-validated peptides, we calculated the percentage of MS-detected peptides that were also identified by ESCAPE-seq for each HLA allele (capture

percentage, see result in revised **Fig. 6c**). Remarkably, a single ESCAPE-seq experiment captured over 80% of MS-detected peptides across most HLA alleles, reaching 90% in some cases. This demonstrates that one ESCAPE-seq experiment can recapitulate the majority of findings from multiple MS experiments across 30 different HLA alleles. We further evaluated ESCAPE-seq's performance using receiver operating characteristic (ROC) curves, comparing true positive rates against false positive rates relative to MS data (see result in revised **Fig. 6d**). The analysis revealed strong performance with AUC-ROC values between 0.8-1.0 for most HLA alleles.

Figure legend: **Figure 6c**, The percentage of epitopes identified by MS that shows a positive E-score by ESCAPE-seq. **Figure 6d**, AUC-ROC metrics comparing ESCAPE-seq and MS results across 30 HLA alleles. Peptides were assigned a value of 1 if detected in MS and 0 if not.

Second, we performed direct MS validation of novel ESCAPE-seq predictions. Following the MS validation approach from the EpiScan paper (PMID: 36593401), we conducted MS analysis of HLA-eluted peptides from HLA-I KO HEK cells expressing HLA-C*03:04 and prioritized epitopes. We successfully validated three high-confidence epitopes with E-scores >7.2: EWSR-FLI fusion, EGFR T790M, and FLT3 D835Y (See results in revised **Figure S5g**). Notably, medium/low E-score epitopes were not detected by MS, suggesting the use of stringent E-score cutoffs for identifying high-confidence epitopes.

EWSR-FLI fusion: SSYGQQSSL

EGFR T790M: LTSTVQLIM:

FLT3 D835Y: YIMSDSNYV

Figure legend: Figure S6g, Intensity traces show the detection of peptides (YIMSDSNYV, LTSTVQLIM and SSYGQSSL) with Mass spectroscopy (MS).

Finally, we characterized the immunogenicity of ESCAPE-seq identified neoantigens using HLA-typed healthy donor samples as an additional validation approach. Given that fusion oncoproteins often generate novel junctional sequences, we hypothesized these breakpoints could serve as immunogenic epitopes. We selected 13 fusion breakpoint-derived epitopes identified by ESCAPE-seq predicted to bind either HLA-A03:01 or A24:02 and tested them using PBMCs from an HLA-A03:01+/A24:02+ healthy donor. Using a well-established protocol, we differentiated APCs from donor PBMCs, stimulated them with peptide pools in the presence of cytokines, and assessed T cell responses after 7 days of expansion. Upon restimulation with individual peptides, we measured intracellular IFN γ and TNF α production by flow cytometry. This analysis revealed that the NPM1-ALK fusion-derived epitope was immunogenic, evidenced by a significant increase in IFN γ +TNF α + polyfunctional T cells following peptide stimulation. The results are shown in revised **Figure. 6**.

Figure legend: **a** (Figure 6d in the manuscript), Schematic view of peptide stimulation assay to detect antigens-specific T cell reactivity. **b** (Figure 6h), Flow cytometry plot of IFNg and TNFa producing CD8+ T cells. **c** (Figure 6i), Bar plot showing the percentage of IFNg+TNFa+ polyfunctional CD8+ T cells upon individual peptide or pooled peptides stimulation.

We extended this analysis to EGFR T790M neoantigens, comparing three distinct cases: two epitopes (p22: QLIMQLMPF and p23: LIMQLMPFG) where both mutant and wild-type versions showed high E-scores (mut+wt+), and one epitope (p17: LTSTVQLIM) where only the mutant version showed a high E-score (mut+ only). We hypothesized that the mut+ only epitope would be more immunogenic due to the absence of central tolerance, as the wild-type version is not naturally presented. Consistent with our hypothesis, only the mut+ only neoantigen (p17 in Figure below) elicited a T cell response, while the mut+wt+ epitopes failed to generate responses despite their high presentation potential. These findings validated the neoantigens identified from ESCAPE-seq and nominated potentially immunogenic neoantigens from comparison of presentation ability between WT and Mut antigens. The results are shown in revised Fig. 6.

Figure legend: **a** (Figure 6d in the manuscript), Schematic view of peptide stimulation assay to detect antigens-specific T cell reactivity. **b** (Figure 6e), Bar plot showing E-score of EGFR T790M epitopes between mutant vs their wild-type counterpart. **c** (Figure 6f), Flow cytometry

plot of IFN γ and TNF α producing CD8 $^+$ T cells. **d (Figure 6g)**, Bar plot showing the percentage of IFN γ +TNF α + polyfunctional CD8 $^+$ T cells upon individual peptide stimulation. CEFT peptide pools represent well-established immunogenic common viral antigens.

Together, these validations collectively demonstrate ESCAPE-seq's particular strength in recapitulating MS-detected peptides across multiple HLA alleles, identifying novel HLA-C presented neoantigens confirmed by direct MS, and discovering immunogenic neoantigens capable of eliciting specific T cell responses.

2. Only 9mer peptides were included in the ESCAPE-seq library. It is unclear about the utility of this technique on a more complex system, such as broader coverage of pHLA-I at all lengths and/or translatability to identify peptides associated with HLA-II. Could authors elaborate more on how the system would capture epitopes in a more biologically relevant scenario, ranging from 8-11 amino acids? I suspect it will be important to incorporate the natural difference in peptide lengths into calculating preference and abundance on MHC. For instance, in comparing WT versus mutant protein presentation, would this have been impacted by not only comparing peptides of 9aa, but also ranging from 8aa to 11aa? How would including a pHLA library that contains multiple peptide lengths affect the E-score algorithm?

Thank you for this important question about peptide length diversity in ESCAPE-seq. ESCAPE-seq is not limited to 9-mer peptides and can analyze peptides of any length. While our initial manuscript focused on 9-mers, as they represent the dominant length for Class I HLA peptides in both IEDB and MS datasets, we have now performed additional ESCAPE-seq across multiple peptide lengths.

Following the reviewer's suggestion, we performed ESCAPE-seq analysis with peptides ranging from 8 to 12 amino acids, examining 10 gene loci (3 mutations, 3 wild-type counterpart genes, 3 fusions, 1 deletion) across 25 HLA alleles. For each point mutation/fusion/deletion, we designed approximately 45 peptides of varying lengths tiling across the mutation site/junction site, resulting in around 10,000 peptide-HLA interactions (See revised **Supplementary Figure 4**).

Figure legend: **a (Figure S4f in the manuscript)**, Schematic of ESCAPE-seq screening for peptides of 8–12 amino acids in length across 25 HLA alleles. **b (Figure S6g)**, Scatter plots of read counts between biological replicates, and **c (Figure S6h)**, scatter plots of calculated E-scores between biological replicates. Pearson correlation coefficients are shown in each plot.

Our analysis revealed that 10-mer peptides showed the highest presentation frequency across all examined alleles, at least for the selected peptides (see results in revised **Figure S4i-j**).

Figure legend: a (Figure S4i in the manuscript), Bar plot showing the number of presented peptides identified by ESCAPE-seq at different peptide lengths. **b (Figure S4j)**, E-score heatmap for peptides generated from the same gene locus with varying lengths per allele. Examples include peptides derived from **EGFR T790M** presented by HLA-B*52:01 and peptides tiling across **KRAS G12D** presented by HLA-A*68:01. The heatmap illustrates changes in E-scores when an epitope is extended by one amino acid at either the N- or C-terminus. In some cases, peptide presentability remains consistent when an additional amino acid is added to the N-terminus (left and right panels) or the C-terminus (right panel).

This unexpected finding might be explained by several factors:

1. The selected mutations in our dataset may contain biologically meaningful 10-mer epitopes (such as KRAS G12D, **Figure S4j**)
2. In our single-chain format, 10-mers may present two different 9-mer binding registries, potentially increasing presentation probability
3. N-terminal extensions might be accommodated through ERAP trimming, while C-terminal extensions may be tolerated due to flexible linker regions.

To understand how peptide length affects presentation, we use EGFR T790M epitope LTSTVQLIM as an example (validated for immunogenicity in A*23:01+ donors). We observed that EGFR T790M peptide length variant showed consistent E-scores between 9-mers and 10-mers for both wild-type and mutant sequences. However, extending to 11 or 12 amino acids resulted in substantially decreased E-scores, suggesting intolerance for longer extensions.

Figure legend: Bar plot showing comparison of HLA-A2301 E-score of mutant and wild-type counterparts between different peptide length.

Importantly, the E-score calculation is normalized per peptide, ensuring that including different peptide lengths does not affect the overall E-score distribution or calculation methodology. This normalization allows for reliable comparison across different peptide lengths while maintaining the integrity of our scoring system.

3. The novelty of the study is partly an issue as the study PMID: 36593401 uses essentially an overlapping strategy to identify MHC I peptides, though goes further in validating the accuracy of predicted peptides by MS analysis. Nevertheless, I do believe this study has potentially revealed some important new findings related to public neoantigen presentation across multiple MHC alleles, a critical role for HLA-C in antigen presentation of peptides from virus and cancer, and a refined technique for validating more HLA allelic variants in a single screen. I believe with further validation of the technique and further proof the existence of such peptides would be of importance to the field of antigen presentation identification.

Thank you for the positive assessment. While ESCAPE-seq and EpiScan share the fundamental principles of leveraging cellular quality control and deep sequencing, our platform introduces several key innovations that significantly accelerate the antigen discovery. The primary differences lie in both technical design and practical capabilities.

To directly compare platform performance, we analyzed SARS-CoV-2 spike protein epitope screening on HLA-A02:01 and HLA-B07:02 alleles from ESCAPE-seq and EpiScan (using supplementary table in EpiScan paper PMID: 36593401). Cross-referencing with 19 previously validated HLA-A02:01 epitopes and 4 HLA-B07:02 epitopes known to stimulate antigen-specific T cell responses (PMID: 34458873), ESCAPE-seq identified 17 out of 19 A02:01 epitopes and all 4 B07:02 epitopes. In contrast, EpiScan identified only 1 of the A02:01 epitopes and none of the B07:02 epitopes.

Figure legend: Pie charts to show the overlap of SAR-COV2 epitopes discovered by ESCAPE-seq (left panels) or by EpiScan (right panels, data from PMID: 36593401) with published known epitopes. Top two panels shows the epitopes for HLA-A*0201 allele. Bottom panels are for epitopes presented by HLA-B*0702.

This increased sensitivity stems from our single-chain peptide-HLA trimer design with cysteine mutation, compared to EpiScan's approach of expressing peptide and HLA separately. When we

experimentally compared surface display levels between our design and co-transfected HLA allele with ER-locating peptide (mimicking EpiScan's approach), our single-chain design showed ~100-fold higher pHLA surface expression. While this increased sensitivity might potentially increase false-positive rates, we can address this through stringent E-score cutoffs.

Figure legend: Figure 1f, Histogram of B2m staining for cell surface pHLA-A2 presentation. 3 conditions were measured, respectively B2m fused with HLA-A2 only, co-expression of B2m-HLA-A2 with a high affinity peptide pp65 preceded with a signal peptide, and lastly a normal SCT for pp65 peptide. Right side: the bargraph showing the quantification of signals. **** p<0.0001;

Key technical and practical advantages of ESCAPE-seq include:

1. **Increased sensitivity:** ESCAPE-seq's unique single-chain peptide-HLA trimer design with cysteine mutation ensures high sensitivity by guaranteeing peptide-HLA proximity and enhanced complex stability. This design shows significantly higher pHLA surface expression compared to conventional co-expression approaches, and higher SARS-CoV2 antigen discovery rate compared to EpiScan.
2. **Universal applicability:** ESCAPE-seq requires only a universal HLA-I/TAP double KO cell line, while EpiScan needs additional engineering (e.g., ERAP knockout for all alleles and signal peptide peptidase knockout for HLA-A*02:01 allele) and individual monoallelic HLA expression.
3. **Broader HLA coverage:** ESCAPE-seq has been applied to across 50 diverse HLA alleles (14 HLA-A, 29 HLA-B, 7 HLA-C), successfully identifying high-confidence binders for all tested alleles (validated by spiked-in positive binders of each allele in combinatorial screen). EpiScan has more limited testing (4 HLA-A, 4 HLA-B, 3 HLA-C alleles) notably failed with all HLA-C alleles.
4. **Combinatorial screening capability:** ESCAPE-seq for the first time enables simultaneous screening of multiple peptide-HLA combinations (1500 peptide X 50 HLA alleles = 75,000 combinations in one screen), a feature that cannot be achieved with EpiScan.
5. **Comprehensive antigen profiling:** We have screened diverse antigens including viral epitopes, tumor neoantigens from driver mutations, fusion oncoproteins, fusion-derived indels, and deletions across all 50 HLA alleles, whereas EpiScan only screened viral antigens across 11 HLA alleles where 3 HLA-C alleles failed.

These advances make ESCAPE-seq a more versatile and sensitive platform for antigen presentation identification, particularly valuable for discovering public neoantigens.

Additional comments:

4. I believe the introduction should have discussed the role of MS approaches for peptide identification, and not only mention this in the discussion.

We have revised the introduction to include a comprehensive discussion of the role of MS approaches, highlighting: (1) the historical significance of MS in defining the rules of antigen presentation and discovering naturally processed peptides; (2) the power of modern monoallelic MS approaches in systematically characterizing HLA allele-specific peptide repertoires; (3) the current limitations of MS approaches, such as sensitivity thresholds and the challenge of detecting low-abundance peptides, and (4) the complementary nature of MS with other peptide identification methods.

5. Further discussion of why HLA-C comes up some prominently with ESCAPE-seq and not by previous methods. Beyond levels of expression, it seems this is an important discovery, which if validated to be true, deserves further discussion as to why HLA-C is such a good peptide presenter.

Our ESCAPE-seq analysis revealed surprisingly high numbers of presented neoantigens for certain HLA-C alleles compared to HLA-A and -B alleles, prompting us to carefully investigate potential systematic bias. We conducted comprehensive validation using independent HLA monoallelic MS data and performed new ESCAPE-seq experiments across diverse HLA alleles, including 5 HLA-C alleles, using MS-identified peptides as ground truth. Comparison of ESCAPE-seq results with monoallelic MS data demonstrated consistent predictive performance (AUC-ROC 0.8-1.0) across HLA-A, -B, and -C alleles, indicating our method was not systematically biased toward HLA-C (**Figure 4d**).

Figure legend: **Figure 4d**, AUC-ROC metrics comparing ESCAPE-seq and MS results across 30 HLA alleles. Peptides were assigned a value of 1 if detected in MS and 0 if not when calculating area under curve (AUC) of ROC curve.

However, deeper analysis revealed that a small subset of HLA-C alleles display distinct E-score distributions. When examining the E-score histograms across all HLA alleles, three HLA-C alleles (highlighted in bold in Figure-a) exhibited notably broader background peaks compared to typical HLA-A and -B alleles. This broader distribution pattern (highlighted with an arrow) suggested these HLA-C alleles might require different E-score thresholds for accurate peptide presentation prediction.

Figure description: Figure SN-3, Examples of E-score distributions for multiple alleles. Arrows highlighted the curve locations with an extra intermediate peak or broader background peak.

To validate this hypothesis, we performed targeted MS validation of HLA-C*03:04 peptides across different E-score ranges. Only peptides with very high E-scores ($E \geq 7.2$) were identified by MS, supporting the need for more stringent thresholds for these HLA-C alleles.

EWSR-FLI fusion: SSYGQQSSL E-score: 7.29

EGFR T790M: LTSTVQLIM E-score: 7.34

FLT3 D835Y: YIMSDSNYV E-score: 7.33

Figure description: Figure S5g, Intensity traces show the detection of peptides (YIMSDSNYV, LTSTVQLIM and SSYGQSSL) with Mass spectroscopy (MS).

We propose that the broader E-score background distributions observed in certain HLA-C alleles reflect their distinct structural and biochemical properties. HLA-C molecules are known to have more promiscuous peptide binding capabilities and shallower peptide-binding grooves compared to HLA-A and -B. Additionally, our single-chain trimer design incorporating cysteine mutations may enhance this intrinsic binding promiscuity, leading to elevated background signals. Rather than indicating superior peptide presentation capacity, these patterns suggest a need for allele-specific thresholding. This finding highlights a key advantage of ESCAPE-seq: its ability to provide both quantitative ranking scores and allele-specific E-score distributions. Users can leverage these distributions to optimize detection thresholds based on their specific applications and the HLA alleles of interest. For applications requiring high specificity, more stringent thresholds can be applied to HLA-C alleles showing broader background distributions, while maintaining standard thresholds for other alleles.

6. Fig 1a-b. Does the pHLA SCT complex remain intact after being transfected in HEK293? Both b2m and HLA were measured from the complex, but there is lacking direct measurement of peptides that still stay intact with the HLA molecule.

To validate the structural integrity of the pHLA SCT complex on the HEK293 cell surface following transfection, we used two independent approaches.

First, we utilized the well-characterized NY-ESO-1 TCR system, which specifically recognizes the NY-ESO-1₁₅₇₋₁₆₅ SLLMWITQC epitope presented by HLA-A0201. We engineered a Jurkat NFAT-GFP reporter T cell line expressing NY-ESO-1 TCR, where GFP expression serves as a sensitive readout for antigen-specific TCR activation. Specifically, we transfected HEK293 cells with either the HLA-A2 SCT containing NY-ESO-1 epitope or an HLA-A2 SCT containing a CMV pp65

epitope as a negative control and coculture them with NYESO-1 TCR Jurkat reporter cells. Only HLA-A2:NY-ESO-1 SCT-expressing cells triggered NFAT reporter activation in Jurkat cells expressing the cognate NY-ESO-1-specific TCR, demonstrating proper peptide positioning for TCR recognition and antigen-specific TCR activation. The result is shown in revised **Figure S1c-d**.

Figure legend: Figure S1c, T-cell stimulation assay showing that NYESO pHLA-A0201-expressing cells induce NFAT-eGFP reporter expression in T cells co-expressing a cognate 1G4 TCR. In contrast, pp65 pHLA-A0201-expressing cells do not stimulate T cells.

Second, we leveraged a widely-used antibody that can specifically recognize and bind to OVA peptide SIINFEKL presented on mouse H-2Kb allele. We transfected H2Kb:SIINFEKL SCT in HEK293T cells and stained this antibody to assess cell surface display of this peptide-MHC. Our results showed concordant staining of this antibody with total surface H2-Kb in SCT-transfected HEK 293T cells, indicating stable peptide-MHC binding by this antibody and structural integrity on cell surface. The result is shown in revised **Figure S1d**.

Figure legend: Figure S1d, Direct staining of OVA-pH-2Kb-expressing cells using anti-SIINFEKL pH-2Kb and anti-H-2Kb antibodies.

Together, these results showed that only properly folded SCTs with high-affinity peptides can traffic to and maintain structural integrity at the cell surface.

7. Fig 2b. How was the background histogram determined? Please include isotype control staining for b2m antibody.

The background histogram is determined using HLA-I/TAP double KO HEK cells stained with B2M antibody. We have included the isotype control for B2M staining and updated the **Figure S1a** and relevant figure legend.

Figure legend: **Figure S1a**, Histogram of β 2m staining in wild-type (WT) HEK293T cells, HEK293T cells with HLA-A, -B, and -C knocked out using CRISPR-Cas9 RNP (HLA-KO), and HLA-KO HEK293T cells expressing NYESO pHLA-A0201 single-chain trimer (SCT). Bottom: isotype antibody staining of HLA-KO HEK293T cells expressing pHLA-A0201.

8. Fig 3c. Do most of the “low E-score” peptides not share the same homologous with “high E-score” peptides? Are there any consensus motifs in “low E-score” peptides?

We performed a systematic analysis of HLA-A2 binding peptide motifs across different E-score ranges. We found that peptides with high E-scores ($E > 6$) showed strong matches to experimentally-determined HLA binding motifs from mass spectrometry data. In contrast, peptides with medium ($3.8 < E \leq 6$) or low ($2.5 < E \leq 3.8$) E-scores demonstrated progressively weaker matches to these validated motifs, suggesting potentially increased false positive rates at lower thresholds.

Known HLA-A0201 motif:

Our motif analysis across different E-score ranges:

Figure legend: Figure SN-2, E-score distribution for HLA-A*02:01 (a) and peptide motif pattern (b) based on different E-score bins as illustrated in (a).

9. Fig S3G-H. Text states “significant decrease” but there are no statistical tests in the plot.

A t-test p-value was added to document significant decrease.

Figure S3g-h:

**** marked p-value < 0.0001

10. Fig 5d-e. Have the identified “public antigens” been also reported to be presented on multiple alleles in other literature? Is there any existing neoantigen database that could support the findings?

Indeed, several of our identified "public antigens" align with previous reports of neoantigen peptide presentation across multiple HLA alleles including the following examples

Publication	Neoantigen sequence	HLA alleles shown to present neoantigen
PMID:35474673	KRAS G12V (VVVGAVGVGK)	HLA-A0301, HLA-A1101, HLA-A6801
PMID:35474673	KRAS G12V (GAVGVGKSAL)	HLA-C0304, HLA-C1203
PMID 37758652	BRAF-V600E (KIGDFGLATEK)	HLA-A0301, HLA-A1101
PMID 37758652	EGFR-L858R (KQSSKALQR)	HLA-A0301, HLA-A1101

The current existing neoantigen database such as CEDAR (Cancer Epitope Database and Analysis Resource) also supports the findings showing that KRAS G12V neoantigen can be broadly presented by HLA-A0301 and HLA-A1101 alleles.

11. Fig 5f. The authors stated immunogenic neoantigens are likely to reside in the Mut+ quadrant

and pointed out a few previously reported neoantigens. However, there is no data in this study to show the comparison of immunogenicity in peptides from Mut+ vs. other groups.

We hypothesized that Mut+ neoantigens would demonstrate higher immunogenicity compared to Mut+WT+ peptides due to fundamental differences in central T cell tolerance. Since Mut+ peptides lack presentable wild-type counterparts on the same HLA allele, T cells recognizing these epitopes would not have been eliminated during thymic selection. In contrast, for Mut+WT+ peptides, high-affinity T cells recognizing these epitopes would likely have been eliminated during central tolerance due to the presentation of their wild-type counterparts, potentially limiting their immunogenic potential.

To experimentally validate this hypothesis, we conducted a direct comparison between EGFR T790M Mut+ and Mut+WT+ peptides using HLA-A*23:01 as our model system. We defined Mut+ peptides as those showing high presentation scores for mutant sequences (E-score > 6) but low scores for wild-type sequences (E-score < 3.8). Conversely, Mut+WT+ peptides were defined as those with high presentation scores (E-score > 6) for both mutant and wild-type sequences. We performed a well-established T cell reactivity assay following a systematic protocol. First, we induced APC from donor PBMCs and stimulated these APCs with either Mut+ or Mut+WT+ peptides. The cells were cultured for 7 days to allow antigen-specific T cell expansion, followed by peptide restimulation and intracellular cytokine staining. We then measured IFN γ and TNF α production by flow cytometry to assess T cell responses.

Using PBMCs from an HLA-A*23:01-positive healthy African American donor, we observed that the Mut+ peptide elicited a significant increase in IFN γ + TNF α + double-positive T cells compared to DMSO control. In contrast, the Mut+WT+ peptides failed to generate a significant T cell response above background. The difference in immunogenicity was consistent across multiple experimental replicates.

Figure legend: a (Figure 6d in the manuscript), Schematic view of peptide stimulation assay to detect antigens-specific T cell reactivity. **b (Figure 6e),** Bar plot showing E-score of EGFR T790M epitopes between mutant vs their wild-type counterpart. **c (Figure 6f),** Flow cytometry plot of IFN γ and TNF α producing CD8+ T cells. **d (Figure 6g),** Bar plot showing the percentage of IFN γ +TNF α + polyfunctional CD8+ T cells upon individual peptide stimulation. CEFT peptide pools represent well-established immunogenic common viral antigens.

These findings provide direct experimental evidence supporting our hypothesis that Mut+ neoantigens are more likely to be immunogenic compared to Mut+WT+ peptides, likely due to the absence of central tolerance against these novel epitopes. This observation has important implications for neoantigen prioritization in cancer immunotherapy approaches.

12. Fig S1b. Do the 2 red/blue bars represent 2 replicates? Please specify in the figure or figure legend.

Thank you for bringing this issue to our attention. We apologize for any confusion in **Supplementary Figure 1b**. We have clarified in the revised Figure S1b and Figure legend: “Red bars: Two different high-affinity peptides (HP1 and HP2); Blue bars: Two different no-affinity peptides (NP1 and NP2).”

We have updated the figure labels and legend to clearly indicate these distinctions and have included the specific peptide sequences in **Supplementary Table 1** for reference.

Figure legend: Bar plots for (Fig. 1b) quantification, comparing gMFI of b2m staining of HEK293T cells expressing single chain trimer (SCT) of HLA-A*0201 with high affinity (HP) peptides vs. fusions with peptides that have no binding affinity (NP). Red bars: Two different high-affinity peptides (HP1 and HP2); Blue bars: Two different no-affinity peptides (NP1 and NP2).

13. Fig S2b & S3b. Please show correlation coefficients in all correlation curves.

Correlation coefficients have been added as suggested.

Figure S2a-b:

Figure S3a-b:

Reviewer #1:

Remarks to the Author:

The authors have fully addressed all of my concerns.

Thank you so much for your comments!

Reviewer #2:

Remarks to the Author:

I have re-evaluated the revised manuscript entitled 'Massively parallel immunopeptidome by DNA sequencing reveals landscape of cancer antigen presentation' by Shi et al. The authors have effectively addressed all of my previous concerns, significantly improving the quality of the manuscript. The inclusion of new data has further strengthened the study, and the additional experiments and detailed descriptions now provide a clear and comprehensive presentation of the method. I believe the manuscript meets the standards of Nature Genetics and is suitable for publication.

Thank you so much for your comments!

Reviewer #3:

Remarks to the Author:

Overall, the revision has addressed the major concerns in the first review, including using complementary methods (published MS datasets and a T cell assay from healthy donor) to validate ESCAPSE-seq, expanding HLA class I length coverage, and providing a more in-depth discussion on similar technologies. The provided new data has supported ESCAPSE-seq's sensitivity and robustness in identifying peptides with high binding affinity to HLA-I molecules. They also refined their analysis of HLA-C peptides to find that a more stringent threshold is required to prevent false-positive predictions. Overall, my opinion is that the manuscript has gone far enough to validate their predictive results with ESCAPE-seq.

While I believe it would have been more powerful if they had experimentally validated that more NOVEL peptides (beyond single EGFR mutant peptide that has been widely published already) identified by ESCAPSE-seq were antigens expressed in patient samples, this is beyond the scope of feasibility in cancer patients. This could have been done more readily in HLA-haplotyped COVID patient samples using tetramer staining to look for specific T cell responses against top novel candidates from ESCAPSE-seq. This evaluation would provide insights into the false-positive rate of new antigen

discovery and support ESCAPSE-seq's potential in revealing novel targets, but considering the entire study, it is not essential.

Thank reviewers' comments. We have validated NOVEL peptides beyond EGFR mutant including FLT3 D835Y peptide YIMSDSNYV and EWSR1-FLI1 fusion peptide SSYGQQSSL which are all presented on HLA-C0304 allele from mass spectrometry validation in Figure S5g. We also validated NOVEL peptide HISGQHLVV from NPM1-ALK fusion based on T cell stimulation assay in Figure 6h-i.

Minor comments:

1. Please explain the ranking algorithm (y-axis) in Fig 3b & S3c-e and elaborate on the meaning of peptides with the rank score <1.

We added the explanation in the Figure 3 legend. The ranking score is from the widely-used NetMHCpan prediction tool, which use Artificial Neural Networks trained on experimental data in IEDB. The % of ranking score shows how a peptide's predicted binding affinity compares to a large set of random natural peptides.

2. Please explain the CEFT condition in Fig 6g.

CEFT is Positive Control Pool of 27 peptides selected from defined HLA class I-restricted T-cell epitopes (mostly are viral epitopes from common viruses) for T cell assays, purchased from JPT

3. The terminology of "genome-wide association study" was not cohesively used, with the full term in line 72 but the abbreviation in line 454.

Thank reviewer to point this out. We modify to "genome-wide association study (GWAS)" on line 72, and use abbreviation afterward.

4. Differential thresholding of E-scores for individual HLA alleles is well-discussed in the rebuttal. I believe this is a piece of critical information that should also be included in the manuscript to provide a comprehensive view of ESCAPSE-seq.

Thank reviewer's comments and suggestions. We now included it in the supplement materials.